# Improving Black-Box Generative Attacks via Generator Semantic Consistency

**Jongoh Jeong**[1], **Hunmin Yang**[1,2], **Jaeseok Jeong**[1], **and Kuk-Jin Yoon**[1],
[1]Visual Intelligence Lab., Korea Advanced Institute of Science and Technology (KAIST)
[2]Agency for Defense Development (ADD)
{jeong2, hmyang, jason.jeong, kjyoon}@kaist.ac.kr ○ Project Page

## Abstract

Transfer attacks optimize on a surrogate and deploy to a black-box target. While iterative optimization attacks in this paradigm are limited by their efficiency and scalability due to multistep gradient updates per input, generative attacks alleviate these by producing adversarial examples in a single forward pass at test time. However, current generative attacks still adhere to optimizing surrogate losses (e.g., feature divergence) and overlook the generator's internal dynamics, underexploring how the generator's internal representations shape transferable perturbations. To address this, we enforce semantic consistency by aligning the early generator's intermediate features to an exponential moving average (EMA) teacher, stabilizing object-aligned representations and improving black-box transfer without inference-time overhead. To ground the mechanism, we quantify semantic stability as the standard deviation of foreground IoU between cluster-derived activation masks and foreground masks across generator blocks, and observe reduced semantic drift under our method. For more reliable evaluation, we also introduce Accidental Correction Rate (ACR) to separate inadvertent corrections from intended misclassifications, complementing the inherent blind spots in traditional Attack Success Rate (ASR), Fooling Rate (FR), and Accuracy metrics. Across architectures, domains, and tasks, our approach can be seamlessly integrated into existing generative attacks with consistent improvements in black-box transfer, while maintaining test-time efficiency. Code at https://github.com/andyj1/scga.

## 1 Introduction

Deep neural networks have driven advances in computer vision, natural language processing, and medical diagnosis by learning rich hierarchical representations. At the same time, they remain vulnerable to small human-imperceptible perturbations known as adversarial examples (AE) Szegedy et al. (2013), which can induce confident misclassification and raise safety concerns in real deployments. The risk is amplified in black-box settings, where an attacker has no access to the parameters or architecture of a model. In these scenarios, transfer-based attacks craft perturbations on a surrogate and deploy them against unseen targets, enabling a single perturbation strategy to threaten diverse safety-critical systems such as self-driving and biometrics.

Early white-box iterative attacks (for example, FGSM and its multistep variants Zhang et al. (2021); Dong et al. (2018); Xie et al. (2019); Dong et al. (2019)) rely on direct gradient access. Transfer-based attacks extend this idea by seeking perturbations that generalize across models, often using iterative optimization in the white-box regime Madry et al. (2017); Carlini et al. (2019); Zhang et al. (2021). While effective, they require per-example iterative optimization, whereas generative attacks amortize this cost by producing perturbations in a single forward pass.

Generative transfer attacks train a feedforward perturbation generator against a surrogate and then produce adversarial noise with one forward pass at the test time Xiao et al. (2018); Wang et al. (2018); Baluja & Fischer (2017; 2018); Poursaeed et al. (2018); Naseer et al. (2019); Nakka & Salzmann (2021); Zhang et al. (2022); Aich et al. (2022); Yang et al. (2024a;b); Nakka & Alahi (2025). This design yields fast inference and strong scalability. However, current generative attacks are centered around optimizing the surrogate-level objectives and treat the generator merely as a

tool to generate adversarial examples given the adversarial objective, overlooking the progressive AE synthesis process in which perturbations are incrementally formed block by block within the generator. This oversight leaves potential for improving transferability, as the intermediate blocks of the generator are where semantic structure, such as the contour of the object and the coarse shape, is preserved or degraded during synthesis Zhang et al. (2022). As a result, perturbations may disperse onto object-irrelevant regions that are relatively less victim model -agnostic, weakening adversarial transferability. This critically raises the following questions:

> Q1. *At what stage of perturbation synthesis do semantic cues deteriorate?*
> Q2. *Which generator blocks most influence transferability?*

To investigate the perturbation synthesis in detail, we partition the six intermediate blocks in the generator into three split blocks–early, mid, and late–and find that the early blocks better preserve object-aligned structure than later ones. We substantiate this claim with a diagnostic analysis of the stability of object-aligned perturbation semantics within the generator intermediate blocks As in Fig. 1, lower cross-block variability, and thus higher consistency of object semantics, is associated with higher transferability of AEs.

Guided by this observation, we propose a semantically consistent generative attack (SCGA) that explicitly targets semantic consistency during perturbation synthesis within the generator. Concretely, we use a Mean Teacher pathway in which an Exponential Moving Average (EMA)-updated teacher provides temporally smoothed reference features, and a self-feature consistency loss aligns the student's early generator block activations with these references while keeping the adversarial objective on the surrogate features unchanged, as shown in Fig. 2. This guidance operates only during training without additional test-time cost, and integrates with existing generative attacks.

Finally, we broaden the evaluation beyond misclassification-based metrics (ASR, FR) and a correction-based metric (Accuracy) to include our proposed Accidental Correction Rate (ACR). For reliable evaluations, ACR complements these conventional metrics by identifying cases that are inherently likely to be overlooked, such as unintended corrections of initially wrong benign predictions. In a comprehensive evaluation setting, we demonstrate that the internal dynamics within the generator play a critical role in enhancing adversarial transferability between domains, models, and even tasks. We summarize our main contributions as follows:

- **Generator–internal evidence for perturbation semantics.** To investigate perturbation semantics within the generator, we partition the generator into early/mid/late blocks and quantify object-aligned semantics per block. Our analysis reveals that methods with lower variability in the foreground IoU across the intermediate blocks exhibit higher adversarial transfer. (§2.2)

- **Generator–level semantic consistency guidance.** By enforcing training-only semantic consistency at the generator's *early* intermediates, we achieve improved adversarial transfer while keeping the adversarial objective on the surrogate unchanged. The guidance can be seamlessly integrated into existing generative attacks without altering the test pipeline at no additional inference cost. (§3)

- **Comprehensive evaluation with an added reliability measure.** We conduct a comprehensive transferability evaluation spanning classification (CLS) across architectures, domains, and dense prediction tasks (SS, OD). We also complement conventional Accuracy, ASR, and FR metrics by introducing a novel ACR metric to assess the attack reliability, measured by inadvertent corrections from intended misclassifications. (§4.2)

## 2 BACKGROUND AND MOTIVATION

### 2.1 PRELIMINARIES

Given a pre-trained victim model $\mathcal{F}^t(\cdot)$ evaluated on a test distribution $\mathcal{D}_{\text{test}}$, the objective is to synthesize human-imperceptible perturbations that transfer across models, domains, and tasks, using only a source domain $\mathcal{D}_{\text{src}}$ and its pre-trained models as substitutes. Generative attack framework employs a generator $\mathcal{G}_\theta(\cdot)$ that maps a benign input $x$ to an unconstrained adversarial candidate $\tilde{x}^{\text{adv}}$, followed by a projector $\mathcal{P}(\cdot)$ that enforces the $\ell_\infty$-budget, i.e., $\|\mathcal{P}(\tilde{x}^{\text{adv}}) - x\|_\infty \leq \epsilon$. Training of $\mathcal{G}_\theta(\cdot)$ is supervised in white-box fashion by a surrogate model $\mathcal{F}^s(\cdot)$ trained on $\mathcal{D}_{\text{src}}$, enabling gradient-based updates via backpropagation. The adversarial loss leverages surrogate logits or intermediate

features of $\mathcal{F}^s(\cdot)$, e.g. at layer $k$, to capture model-shared characteristics known to enhance black-box transferability Naseer et al. (2019); Nakka & Salzmann (2021); Zhang et al. (2022); Aich et al. (2022); Yang et al. (2024a;b). Formally, $\mathcal{G}_\theta(\cdot)$ is optimized to generate AEs that maximize evaluation metrics against victim models $\mathcal{F}^t(\cdot)$ and/or relative to ground-truth labels $y$ with:

$$\texttt{Metric}\Big(x, x_{adv}, \mathcal{F}^t(\cdot), y\Big), \quad \text{with } \|x_{adv} - x\|_\infty \leq \epsilon, \qquad \text{(See §4.2 for metric details.)} \qquad (1)$$

where $\epsilon$ denotes the maximum perturbation budget that guarantees a minimal change in $x$. Here, Metric refers to ASR, FR, Acc., and ACR for classification (CLS); mIoU and mAP50 for semantic segmentation (SS) and object detection (OD), respectively.

## 2.2 PERTURBATION SEMANTICS IN GENERATOR-INTERNAL DYNAMICS

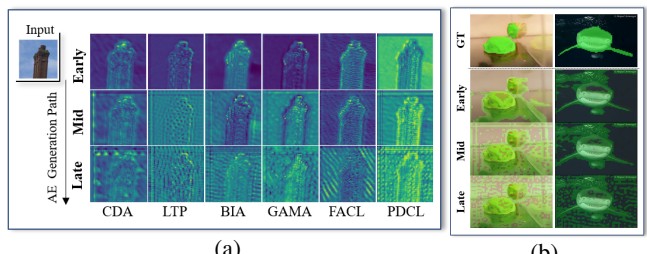

(a)                          (b)                          (c)

Figure 1: Our observation on the semantic variability within the perturbation generator. (a) Generator intermediate feature maps for each block partition, (b) predicted masks from intermediate feature clusters on ImageNet-S Gao et al. (2022) from the baseline Zhang et al. (2022), and (c) quantified variability in foreground IoU.

We observe that intermediate features progressively lose semantic recognizability across residual blocks. Figure 1 shows that early maps preserve object contours, while mid and late maps blur them. Using k-means clustering to separate the foreground and background, we also find that stronger attacks preserve the coarse shape earlier and more consistently in stages. To better quantify how much semantic information is retained throughout the intermediates, we define semantic variability as the cross-block standard deviation of foreground IoU between clustered activation masks and foreground masks along perturbation trajectories, where advanced attacks achieve lower variability, suggesting more stable overlap with foreground. These findings are consistent with the well-established premise that the majority of noise being synthesized in the intermediate stage Zhang et al. (2022); Naseer et al. (2019); Nakka & Salzmann (2021); Zhang et al. (2022); Aich et al. (2022); Yang et al. (2024a;b). Based on this evidence, we apply a lightweight EMA teacher to early blocks, leaving inference unchanged, so that subsequent blocks concentrate perturbations on salient regions and black-box transfer improves. Further analysis is provided in *Supp.* §D.

Crucially, these findings motivate our design to enforce semantic consistency in the intermediate stages of the generator, using an EMA teacher applied in the early blocks to curb semantic drift while leaving the inference path unchanged. By anchoring perturbations to early, semantically consistent features, the later blocks naturally concentrate the generated perturbations in salient object regions, thereby improving black-box transfer between models while preserving internal semantics.

## 3 SEMANTICALLY CONSISTENT GENERATIVE ATTACK

Our semantically consistent generative attack, as described in Alg. 1, augments a standard generative adversarial attack with two key components: a Mean Teacher-based feature smoothing and a self-feature consistency loss that enforces semantic preservation across the intermediate layers of the generator. We base our approach on the baseline work BIA Zhang et al. (2022) as all subsequent works GAMA Aich et al. (2022), FACL Yang et al. (2024a), PDCL Yang et al. (2024b) base their losses on its feature similarity-based adversarial loss, and thus it is adequate to serve as a solid baseline. See *Supp.* §C for the method distinction.

**Role of Mean Teacher.** The Mean Teacher (MT) framework Tarvainen & Valpola (2017); Deng et al. (2021); Li et al. (2022); Zhao et al. (2022); Cao et al. (2023); Döbler et al. (2023) has consistently demonstrated robustness in tasks characterized by significant domain shifts between

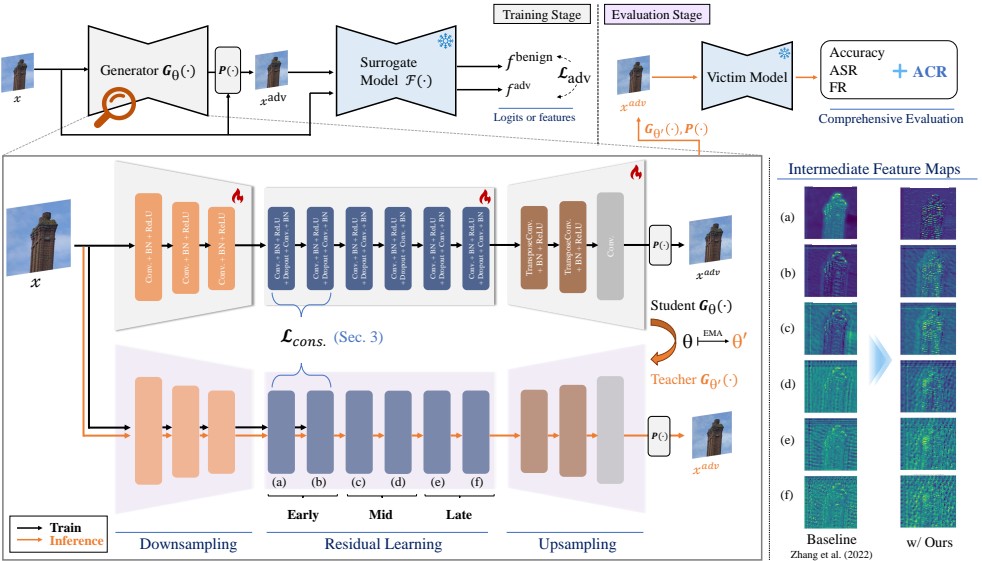

Figure 2: **Overview of our proposed SCGA framework.** Given a benign input image, a perturbation generator produces an adversarial output under the supervision of a Mean Teacher (MT) structure. The student and teacher share the generator architecture, with the teacher updated via EMA. Semantic consistency is enforced by aligning their intermediate features, selectively applied to the *early* blocks to effectively preserve structural information from the benign input across the residual blocks. The adversarial example is then evaluated against victim models according to the four evaluation metrics. This MT-based design further promotes semantic alignment, combining consistency and integrity, thereby enhancing adversarial transferability across diverse victims.

training and testing. Its core mechanism of updating the teacher's parameters with EMA of the student's parameters provides a form of temporal ensemble that naturally suppresses instance-specific noise. Intuitively, this EMA update smooths out high-frequency perturbation artifacts, enriching the semantic consistency and stability of the teacher's intermediate feature maps. As a result, these smoothed features serve as a reliable reference for the student, helping to preserve object contours and shapes throughout adversarial synthesis. To integrate MT, we maintain two generators: a *student* $G_\theta(\cdot)$ that is trained via gradient descent, and a *teacher* $G_{\theta'}(\cdot)$. We set these mean teacher features as a reference for our self-feature consistency matching. We update $\theta'$, per training step $t$, as follows:

$$\theta'_t \leftarrow \eta\, \theta'_{t-1} \;+\; (1-\eta)\, \theta_t, \tag{2}$$

where $\eta \in [0, 1]$ is a smoothing coefficient hyperparameter.

**Self-feature consistency.** Object-salient intermediate representations have been shown to be critical for adversarial transfer in black-box settings Wu et al. (2020); Byun et al. (2022); Kim et al. (2022); Zhang et al. (2022), and recent work has explored manipulating input or surrogate-level features to this end Huang et al. (2019); Li et al. (2023); Nakka & Alahi (2025). In our generative framework, however, a naïve generator progressively loses semantic integrity in its intermediate layers (Fig. 1), scattering perturbations away from object-salient regions. To preserve these crucial object cues, we introduce a self-feature consistency mechanism grounded in the MT paradigm Grill et al. (2020); Caron et al. (2021); Lee et al. (2023). Concretely, we treat the EMA teacher as the source of temporally smoothed, semantically rich features. At each training iteration, we extract early block activations from both the student and the teacher and enforce semantic consistency via a hinge-based feature consistency loss as follows:

$$\mathcal{L}_{\text{cons.}} = \sum_{\ell=1}^{L_{\text{early}}} \mathcal{W}_{\text{cons.}} \cdot \left[ \tau - \frac{\langle \mathbf{g}_s^\ell, \mathbf{g}_t^\ell \rangle}{\|\mathbf{g}_s^\ell\|_2 \, \|\mathbf{g}_t^\ell\|_2} \right]_+, \tag{3}$$

where $[\cdot]_+ := \max(0, \cdot)$ and $\tau$ is the similarity threshold. This loss anchors the student's edges and shape prior to the smoothed semantics of the teacher, ensuring that subsequent perturbations focus on object-centric regions. $\mathcal{W}_{\text{cons.}} \in \mathbb{R}^{|L|}$ denotes the softmax output of a learnable parameter

---

**Algorithm 1:** Pseudo-code of SCGA

---

**Data:** Training dataset $\mathcal{D}_{src}$
**Input:** Generator $\mathcal{G}_\theta(\cdot)$, a surrogate model trained on source data $\mathcal{F}^s(\cdot)$, projector $\mathcal{P}(\cdot)$, perturbation budget $\epsilon$
**Output:** Optimized teacher perturbation generator $\mathcal{G}_{\theta'}(\cdot)$

---

1    Initialize generators: student $\mathcal{G}_\theta(\cdot) \leftarrow$ random init., teacher $\mathcal{G}_{\theta'}(\cdot) \leftarrow \mathcal{G}_\theta(\cdot)$
2    **repeat**
3        Randomly sample a mini-batch $x_i$ from $\mathcal{D}_{\text{src}}$
4        Acquire student generator intermediate features:     $\mathbf{g}_i \leftarrow \mathcal{G}_\theta^{\text{enc}}(x_i)$
5        Acquire teacher generator intermediate features:     $\mathbf{g}'_i \leftarrow \mathcal{G}_{\theta'}^{\text{enc}}(x_i)$
6        Generate unbounded adversarial examples from student generator intermediate features:
        $\tilde{x}_i^{\text{adv}} \leftarrow \mathcal{G}_\theta^{\text{dec}}(\mathbf{g}_i)$
7        Bound (project) $\tilde{x}_i^{\text{adv}}$ using $\mathcal{P}$ within the perturbation budget such that $||\mathcal{P}(\tilde{x}_i^{\text{adv}}) - x_i||_\infty \leq \epsilon$ to obtain $x_i^{\text{adv}}$
8        Forward pass $x_i$ and $x_i^{\text{adv}}$ through the surrogate model, $\mathcal{F}^s(\cdot)$ at layer $k$, to acquire $f_i^{\text{benign}}, f_i^{\text{adv}}$
9        Compute loss using $f_i^{\text{benign}}, f_i^{\text{adv}}, \mathbf{g}_i, \mathbf{g}'_i$:     $\mathcal{L} = \mathcal{L}_{\text{adv}} + \lambda_{\text{cons.}} \cdot \mathcal{L}_{\text{cons.}}$        `// Eq.5`
10       Update student generator parameters via backpropagation
11       EMA update teacher weights with student weights:     $\theta \mapsto \theta'$        `// Eq. 2`
12 **until** $\mathcal{G}_\theta(\cdot)$ *converges*

---

for intermediate block-wise loss weighting. When combined with the adversarial objective, these semantically consistent perturbations that are highly transferable and tightly aligned with the core structure of the image. For fair comparisons with state-of-the-art methods, we adopt adversarial loss in the surrogate feature space as practiced in the baseline, e.g. BIA Zhang et al. (2022):

$$\mathcal{L}_{\text{adv}} = \cos(\mathcal{F}_k(x), \mathcal{F}_k(x^{adv})), \tag{4}$$

where $\cos(\cdot, \cdot)$ denotes cosine similarity.

**Final loss objective.** Putting the proposed and baseline losses together on the `MT` framework, we formulate the final loss objective with $\lambda_{\text{cons.}}$ as a weight term for $\mathcal{L}_{\text{cons.}}$, as follows:

$$\mathcal{L} = \mathcal{L}_{\text{adv}} + \lambda_{\text{cons.}} \cdot \mathcal{L}_{\text{cons.}}. \tag{5}$$

## 4 EXPERIMENTS

We refer to *Supp.* §E.3 for training implementations and computational complexity. For evaluation (*Supp.* §E.1–2), we conduct cross-setting tests under two black-box protocols. In the cross-model setting, perturbations are crafted on surrogate models trained with the same data distribution (i.e., ImageNet-1K Russakovsky et al. (2015)) and then tested on unseen target model architectures. In the cross-domain/task settings, adversarial examples are to generalize across domain/task shifts without access to any target-distribution samples.

### 4.1 LIMITATIONS IN EXISTING EVALUATION PROTOCOL

Although developing an effective attack mechanism is crucial, it must be validated by fair and comprehensive evaluations. The current evaluation protocols adopted by previous works GAP Poursaeed et al. (2018), CDA Naseer et al. (2019), LTP Nakka & Salzmann (2021), BIA Zhang et al. (2022), GAMA Aich et al. (2022), FACL Yang et al. (2024a),

Table 1: **Examples of real-world impacts on predictions** with different evaluation metrics and attack reliability concerns.

| Real-world examples: | | | | | |
|---|---|---|---|---|---|
| Scenario # | **GT Label** | **Benign pred.** | **Adv. pred.** | **Impact** | **Captured by** |
| 1 | cat | cat ✓ | cat ✓ | Correct → Correct | Acc. only |
| 2 | cat | cat ✓ | dog ✗ | Correct → Incorrect | ASR, FR |
| 3 | van | truck ✗ | bus ✗ | Incorrect → Other incorrect | FR only |
| 4 | pelagic cormorant | albatross ✗ | pelagic cormorant ✓ | Incorrect → Correct | ACR, FR. Acc. |
| **Reliable attack example:** | | | | | |
| **Cross-Setting** | | **GT Label** | **Benign pred.** | **Intended Attack** | **Unreliable Attack** |
| ImageNet → FGVC Aircraft | | F-22 Raptor | F-22 Raptor ✗ | F-18 Hornet ✗ | F-22 Raptor ✓ |

PDCL Yang et al. (2024b) exhibit three key limitations. (L1) Most studies report only one primary metric (either ASR, FR, or Acc.), offering only a one-dimensional view of attack robustness and neglecting other aspects such as unintended corrections in predictions. (L2) Data sets and sample

sizes are often arbitrarily or limited to a single scale, preventing a fair comparison between attacks and undermining statistical significance. (L3) Evaluations in previous work commonly target a narrow set of victim architectures (e.g., mostly CNN-based), lacking the diversity of modern model families, including vision transformers (ViT) and state-space models (SSM), and thus overstating robustness. Although conventional work frames the success of attacks as *fooling* the target classifier, we contend that evaluation facets should be expanded for a reliable assessment of attacks.

To address these shortcomings, we introduce, in §4.2 (L1), *Accidental Correction Rate* (ACR) as a complementary metric that captures the proportion of AEs that *inadvertently* restore correct predictions, enriching the evaluation of attack efficacy alongside conventional measures (i.e. ASR, FR, Acc.) as demonstrated with practical examples in Table 1. ACR measures a nuanced model behavior often missed by ASR and FR, which is crucial for a complete understanding of robustness in safety-critical systems where any unreliable response to perturbation may pose a risk. We also evaluate AEs on the *entire* validation set in §4.3 (L2, L3), instead of arbitrary subsets, and cover a *wide range* of victim models for the classification task. We provide further details in *Supp.* §E.

## 4.2 EVALUATION METRICS

We tested the effectiveness and transferability of adversarial attacks across model architectures and domain shifts using four key metrics. For notational convenience here, let $f(x)$ denote the predicted label for input $x$, $f(x + \delta)$ the prediction after applying adversarial perturbation $\delta$, and $y$ the ground-truth label. The evaluation set is indicated by $\mathcal{D}$, with $\mathcal{C} = \{x \in \mathcal{D} \mid f(x) = y\}$ representing correctly classified samples, and $\mathcal{I} = \{x \in \mathcal{D} \mid f(x) \neq y\}$ denoting misclassified samples under clean inference. We formally define our evaluation metrics (%) as follows:

$$\text{Acc.} = |\{x \in \mathcal{D} \mid f(x + \delta) = y\}| / |\mathcal{D}|, \qquad \text{ASR} = |\{x \in \mathcal{C} \mid f(x) = y \wedge f(x + \delta) \neq y\}| / |\mathcal{C}|,$$
$$\text{FR} = |\{x \in \mathcal{D} \mid f(x) \neq f(x + \delta)\}| / |\mathcal{D}|, \quad \text{ACR} = |\{x \in \mathcal{I} \mid f(x) \neq y \wedge f(x + \delta) = y\}| / |\mathcal{I}|,$$
$$(6)$$

where *Top-1 Accuracy* Zhang et al. (2022); Yang et al. (2024a;b) measures the overall proportion of correctly classified samples under clean or adversarial conditions. It serves as a global performance indicator to assess degraded performance after the attack, orthogonal to FR, ASR, and ACR. *Attack Success Rate (ASR)* Poursaeed et al. (2018); Naseer et al. (2019) is a subset of FR, which measures the proportion of samples originally correctly classified that are misclassified by adversarial attack. It directly reflects the targeted misclassification. *Fooling Rate (FR)* Nakka & Salzmann (2021); Nakka & Alahi (2025) quantifies the proportion of adversarial examples that cause a change in the model's prediction, regardless of correctness. It reflects how often the attack disrupts the original decision and is used as a transferability measure. *ACR*, also a subset of FR, is a novel metric that quantifies how often misclassified samples are "accidentally" corrected by adversarial perturbations. This unintended side effect provides insight into the nuanced model uncertainty and behavior at the decision boundaries. For SS and OD, we use the standard mIoU and mAP50 metrics, respectively.

## 4.3 EXPERIMENTAL RESULTS

We demonstrate enhanced cross-model attacks in Table 2, wherein augmenting each baseline generative attack with our method yields consistent improvements across various architectures. Although these results confirm the orthogonality and efficacy of our framework, we observe that CLIP-based approaches with objectives similar to ours, e.g. PDCL Yang et al. (2024b), yield only marginal improvements when combined with our method. We conjecture that optimizing for divergence in CLIP's high-dimensional semantic embedding space may override or dilute the local structural consistency enforced by our early block semantic consistency, thus attenuating incremental gains from preserving fine-grained object contours and textures (see *Supp* §D for detailed explanation).

Table 3 presents the black-box cross-domain transferability results. In both cross-domain and task, the transferability enhancements become more pronounced. Incorporating MT smoothing and early block consistency steadily enhances the attack performance across unseen domains, architectures, and tasks, demonstrating the broad applicability beyond the source data distribution and task.

With measurable gains in attack accuracy, we visually verify whether our method actually induces the generator to pay more attention to the object-salient regions in Fig. 3. Through Grad-CAM Selvaraju et al. (2017) comparisons against the baseline, ours either reinforces confusion or flips the correctly

Table 2: **Quantitative cross-model transferability results**. We report the improvements (Δ %p) of our method relative to each baseline, with better results marked in a darker color. 'Avg.' corresponds to black-box average.

| Cross-model | | CNN | | | | | | | | | | | Transformer | | | | | | Mixer | | Mamba | | Avg. |
|---|---|---|---|---|---|---|---|---|---|---|---|---|---|---|---|---|---|---|---|---|---|---|---|
| Method | Metric | (a) | (b) | (c) | (d) | (e) | (f) | (g) | (h) | (i) | (j) | (k) | (l) | (m) | (n) | (o) | (p) | (q) | (r) | (s) | (t) | (v) | |
| Benign | Acc. (%)↓ | 74.60 | 77.33 | 74.22 | 75.74 | 76.19 | 77.95 | 66.50 | 55.91 | 79.12 | 81.49 | 75.42 | 80.67 | 79.28 | 81.19 | 80.48 | 79.10 | | 69.90 | 66.53 | 66.53 | 73.21 | 73.77 |
| CDA w/ Ours | Acc. (Δ%p)↓ | -15.93 | -8.39 | -12.93 | -12.70 | -8.41 | -11.21 | -5.09 | -6.48 | -10.17 | -35.91 | -19.74 | -0.12 | -0.14 | +0.72 | +0.03 | +0.06 | +0.09 | +0.53 | +0.73 | +0.06 | +0.29 | **-6.89** |
| | ASR (Δ%p)↑ | +20.13 | +10.35 | +16.52 | +15.96 | +10.37 | +13.75 | +6.92 | +10.35 | +12.19 | +42.13 | +24.37 | +0.09 | +0.10 | -0.95 | -0.05 | -0.05 | -0.23 | -0.67 | -1.04 | -0.13 | -0.51 | **+8.55** |
| | FR (Δ%p)↑ | +17.39 | +9.29 | +14.24 | +13.80 | +8.96 | +11.91 | +5.87 | +7.86 | +11.49 | +38.92 | +21.57 | +0.09 | +0.03 | -0.99 | -0.05 | -0.15 | -0.37 | -0.74 | -0.94 | -0.20 | -0.60 | **+7.49** |
| | ACR (Δ%p)↓ | -3.58 | -1.72 | -2.59 | -2.52 | -2.19 | -2.25 | -1.45 | -1.58 | -2.52 | -7.04 | -4.57 | -0.24 | -0.31 | -0.28 | -0.05 | +0.12 | -0.12 | +0.20 | +0.08 | -0.11 | -0.29 | **-1.57** |
| LTP w/ Ours | Acc. (Δ%p)↓ | -8.71 | -9.52 | -8.45 | -10.24 | -4.62 | -10.00 | -5.59 | -9.60 | -9.57 | -5.57 | -5.93 | -1.23 | -1.77 | -7.17 | -1.74 | -3.57 | -5.86 | -5.87 | -9.05 | -3.13 | -5.93 | **-6.34** |
| | ASR (Δ%p)↑ | +11.11 | +11.92 | +10.87 | +12.92 | +5.90 | +12.27 | +8.04 | +15.83 | +11.89 | +6.55 | +7.50 | +1.66 | +2.37 | +8.63 | +2.38 | +4.71 | +9.70 | +8.22 | +13.03 | +4.44 | +7.97 | **+8.47** |
| | FR (Δ%p)↑ | +9.53 | +10.58 | +9.35 | +11.30 | +5.32 | +10.70 | +6.93 | +12.25 | +11.65 | +6.15 | +6.54 | +2.19 | +2.82 | +8.71 | +2.79 | +5.10 | +8.95 | +7.63 | +11.10 | +3.87 | +7.27 | **+7.65** |
| | ACR (Δ%p)↓ | -1.65 | -1.31 | -1.48 | -1.85 | -0.53 | -2.01 | -0.72 | -1.71 | -0.76 | -1.01 | -0.77 | +0.52 | +0.53 | -0.79 | +0.91 | +0.84 | -0.56 | -0.44 | -1.04 | -0.53 | -0.36 | **-0.70** |
| BIA w/ Ours | Acc. (Δ%p)↓ | -2.23 | -1.99 | -1.29 | +0.01 | -3.72 | -1.59 | -3.29 | -2.85 | -0.39 | +3.12 | -0.63 | -0.80 | -0.32 | -0.78 | -0.56 | -1.18 | -1.08 | -1.48 | -0.45 | -0.38 | +0.03 | **-1.04** |
| | ASR (Δ%p)↑ | +2.83 | +2.46 | +1.64 | -0.05 | +4.55 | +1.96 | +4.68 | +4.80 | +0.56 | -3.32 | +1.21 | +0.92 | +0.33 | +0.89 | +0.59 | +1.40 | +1.91 | +2.05 | +0.88 | +0.56 | +0.03 | **+1.47** |
| | FR (Δ%p)↑ | +2.57 | +2.28 | +1.48 | -0.06 | +4.20 | +1.73 | +3.78 | +3.69 | +0.56 | -3.10 | +0.69 | +1.01 | +0.44 | +1.00 | +0.57 | +1.44 | +1.62 | +1.82 | +0.81 | +0.35 | +0.04 | **+1.28** |
| | ACR (Δ%p)↓ | -0.45 | -0.36 | -0.28 | -0.09 | -1.09 | -0.27 | -0.53 | -0.38 | +0.24 | +0.33 | -0.37 | -0.33 | -0.27 | -0.29 | -0.45 | -0.30 | +0.06 | -0.12 | +0.40 | -0.03 | +0.19 | **-0.21** |
| GAMA w/ Ours | Acc. (Δ%p)↓ | -2.54 | -2.46 | -2.65 | -2.15 | -2.49 | -2.19 | -0.24 | -0.17 | -0.97 | -2.49 | -2.94 | +0.07 | +0.03 | -0.24 | -0.01 | -0.15 | -0.51 | +0.07 | -0.48 | -0.59 | -0.41 | **-1.12** |
| | ASR (Δ%p)↑ | +3.22 | +3.14 | +3.40 | +2.82 | +3.14 | +2.73 | +0.34 | +0.30 | +1.22 | +2.92 | +3.67 | -0.04 | -0.05 | +0.30 | +0.13 | +0.12 | +0.89 | -0.17 | +0.83 | +0.75 | +0.52 | **+1.44** |
| | FR (Δ%p)↑ | +2.81 | +2.87 | +2.91 | +2.56 | +2.76 | +2.53 | +0.24 | +0.21 | +1.20 | +2.67 | +3.23 | +0.05 | +0.03 | +0.27 | +0.20 | +0.05 | +0.73 | -0.21 | +0.73 | +0.59 | +0.44 | **+1.28** |
| | ACR (Δ%p)↓ | -0.58 | -0.14 | -0.51 | -0.08 | -0.43 | -0.31 | -0.03 | 0.00 | -0.02 | -0.49 | -0.56 | +0.21 | -0.01 | -0.02 | +0.53 | -0.27 | +0.02 | -0.16 | +0.20 | -0.29 | -0.09 | **-0.14** |
| FACL w/ Ours | Acc. (Δ%p)↓ | +0.10 | -0.59 | -3.35 | -1.97 | -4.92 | -0.60 | -3.29 | -0.69 | -2.01 | -1.91 | -2.64 | +0.11 | -0.33 | +0.21 | -0.51 | +0.56 | -0.18 | -0.50 | +0.45 | -0.30 | -0.17 | **-1.07** |
| | ASR (Δ%p)↑ | -0.20 | +0.74 | +4.30 | +2.46 | +6.15 | +0.75 | +4.68 | +1.25 | +2.40 | +2.23 | +3.15 | -0.10 | +0.41 | -0.24 | +0.53 | -0.69 | +0.14 | +0.68 | -0.72 | +0.34 | +0.15 | **+1.35** |
| | FR (Δ%p)↑ | -0.20 | +0.64 | +3.75 | +2.27 | +5.37 | +0.74 | +3.97 | +0.96 | +2.24 | +2.05 | +2.78 | -0.02 | +0.47 | -0.19 | +0.47 | -0.67 | +0.08 | +0.72 | -0.64 | +0.25 | +0.14 | **+1.20** |
| | ACR (Δ%p)↓ | -0.23 | -0.09 | -0.61 | -0.46 | -0.97 | -0.08 | -0.54 | 0.00 | -0.52 | -0.41 | -0.96 | +0.16 | -0.02 | +0.05 | -0.46 | +0.09 | -0.23 | -0.09 | -0.09 | -0.24 | -0.24 | **-0.28** |
| PDCL w/ Ours | Acc. (Δ%p)↓ | +0.55 | -0.29 | +1.01 | -0.40 | -0.31 | -0.98 | -1.13 | -0.06 | -1.09 | -0.72 | +0.79 | -0.07 | +0.06 | -0.11 | +0.18 | -0.14 | +0.06 | +0.52 | +0.09 | +0.37 | +0.11 | **-0.07** |
| | ASR (Δ%p)↑ | -0.73 | +0.31 | -1.26 | +0.56 | +0.46 | +1.19 | +1.64 | +0.07 | +1.30 | +0.83 | -0.96 | +0.08 | -0.14 | +0.04 | -0.19 | +0.05 | -0.12 | -0.64 | -0.06 | -0.43 | -0.09 | **+0.09** |
| | FR (Δ%p)↑ | -0.68 | +0.27 | -1.08 | +0.36 | +0.45 | +1.09 | +1.42 | +0.13 | +1.23 | +0.81 | -0.88 | +0.22 | -0.10 | +0.09 | -0.03 | +0.09 | -0.22 | -0.44 | +0.05 | -0.33 | -0.10 | **+0.11** |
| | ACR (Δ%p)↓ | +0.03 | -0.18 | +0.29 | +0.09 | +0.18 | -0.22 | -0.12 | -0.05 | -0.33 | -0.22 | +0.21 | -0.06 | -0.23 | -0.42 | +0.08 | -0.47 | -0.04 | +0.22 | +0.15 | +0.27 | +0.16 | **-0.03** |

Table 3: **Quantitative cross-domain/task transferability results**. We report the average improvement (Δ %p) with ours added from each baseline for each domain. Better results in green **boldface**.

| | Cross-domain | | | | | | | | | | | | | Cross-task | | | | | |
|---|---|---|---|---|---|---|---|---|---|---|---|---|---|---|---|---|---|---|---|
| | **CUB-200-2011** | | | | **Stanford Cars** | | | | **FGVC Aircraft** | | | | Avg. | **SemSeg (SS)** | | Avg. | **ObjDet (OD)** | | Avg. |
| Method \ Metric | Acc↓ | ASR↑ | FR↑ | ACR↓ | Acc↓ | ASR↑ | FR↑ | ACR↓ | Acc↓ | ASR↑ | FR↑ | ACR↓ | Acc | DeepLabV3+ | SegFormer | mIoU↓ | Faster R-CNN | DETR | mAP50↓ |
| Benign | 86.91 | N/A | N/A | N/A | 93.56 | N/A | N/A | N/A | 92.07 | N/A | N/A | N/A | 90.85 | 76.21 | 71.89 | 74.05 | 61.01 | 62.36 | 61.69 |
| CDA | 67.73 | 21.48 | 14.16 | 26.66 | 77.68 | 21.88 | 15.38 | 24.07 | 64.42 | 27.51 | 14.55 | 31.13 | 69.94 | 25.63 | 20.16 | 22.90 | 32.78 | 26.29 | 29.54 |
| w/ Ours (Δ%p) | **-16.92** | **+21.48** | **+20.63** | **-3.94** | **-5.86** | **+2.38** | **+2.35** | **-0.24** | **-22.58** | **+27.74** | **+26.44** | **-6.00** | **-15.12** | **-0.47** | **+0.10** | **-0.18** | **-0.80** | **-0.61** | **-0.71** |
| LTP | 48.74 | 45.32 | 8.75 | 49.31 | 57.98 | 39.02 | 13.03 | 40.85 | 43.01 | 54.15 | 8.35 | 56.48 | 49.91 | 23.71 | 26.97 | 25.34 | 29.39 | 22.41 | 25.90 |
| w/ Ours (Δ%p) | **-10.43** | **+11.72** | **+10.87** | **-0.90** | **-10.62** | **+10.98** | **+10.68** | **-2.16** | **-6.41** | **+6.66** | **+6.35** | **-1.41** | **-9.15** | **-1.44** | **-0.86** | **-0.23** | **-2.54** | **-0.23** | **-1.39** |
| BIA | 47.92 | 46.13 | 8.54 | 50.26 | 59.89 | 37.22 | 13.96 | 38.97 | 45.38 | 51.52 | 9.24 | 54.06 | 51.07 | 23.89 | 25.60 | 24.75 | 28.43 | 21.01 | 24.72 |
| w/ Ours (Δ%p) | **-0.02** | **+0.08** | **+0.09** | **+0.49** | **-6.89** | **+6.89** | **+6.70** | **-2.00** | **-4.98** | **+5.25** | **+4.89** | **-1.24** | **-3.96** | **-1.84** | **-0.85** | **-1.35** | **-0.09** | **-0.29** | **-0.20** |
| GAMA | 48.72 | 45.41 | 9.51 | 49.67 | 54.59 | 42.58 | 11.94 | 44.28 | 42.37 | 54.46 | 7.49 | 56.77 | 48.56 | 23.67 | 25.95 | 24.81 | 28.01 | 20.71 | 24.36 |
| w/ Ours (Δ%p) | **-2.41** | **+2.59** | **+2.30** | **-0.67** | **-2.33** | **+2.24** | **+2.10** | **-0.60** | **-2.68** | **+3.06** | **+2.87** | **+0.14** | **-2.47** | **-0.43** | **-1.58** | **-1.01** | **-0.41** | **+0.08** | **-0.16** |
| FACL | 40.85 | 54.36 | 7.21 | 58.01 | 51.23 | 48.23 | 12.9 | 49.71 | 40.08 | 59.35 | 7.34 | 61.39 | 44.05 | 23.75 | 26.40 | 25.08 | 27.94 | 20.91 | 24.43 |
| w/ Ours (Δ%p) | **+3.12** | **-3.79** | **-3.58** | **+0.70** | **-7.26** | **+2.34** | **+4.72** | **-4.49** | **-2.68** | **+0.60** | **+0.66** | **-0.24** | **-2.27** | **-0.37** | **-1.39** | **-0.88** | **-0.30** | **-0.62** | **-0.46** |
| PDCL | 42.36 | 52.32 | 7.48 | 55.93 | 50.41 | 46.85 | 12.31 | 48.46 | 38.96 | 58.23 | 6.86 | 60.34 | 43.91 | 24.42 | 26.05 | 25.24 | 28.48 | 21.38 | 24.93 |
| w/ Ours (Δ%p) | **-0.46** | **+0.61** | **+0.66** | **+0.40** | **-0.71** | **+0.75** | **+0.69** | **-0.32** | **-1.38** | **+1.52** | **+1.42** | **+0.14** | **-0.85** | **-1.91** | **-0.17** | **-1.04** | **-0.82** | **-0.65** | **-0.73** |

attending regions (similar to those of benign). Across unseen tasks, we also observe fewer pixels and instances with the correct classifications. We attribute this cross-task generalization to our label-agnostic training pipeline and further validate that our method can be integrated with alternative generator architectures beyond ResNet in *Supp.* §B.

Against robust training (i.e. adversarially trained IncV3 Kurakin et al. (2018),ViT Dosovitskiy et al. (2021), ConvNeXt Singh et al. (2023), and input pre-processing JPEG Guo et al. (2017) BDR Xu et al. (2018), R&P Xie et al. (2018)) techniques, our methods demonstrate superior attacks compared to the baseline as shown in Table 4, reinforcing our hypothesis that enforcing semantic consistency in early generator blocks not only boosts transferability in standard black-box settings but also produces perturbations capable of further enhancing attacks against defense mechanisms. By anchoring structural cues in the early stages, our self-feature consistency loss yields more potent and robust attacks against adversarially trained models and input pre-processing defenses alike.

Table 4: **Superior attack success with our method against robustly trained models** including Adversarially trained (AT) models and robust input pre-processing methods. Better averaged results in green **boldface**.

| Method | Metric | Adv.IncV3 | Adv.ViT | Adv.ConvNeXt | JPEG | BDR | R&P | Avg. |
|---|---|---|---|---|---|---|---|---|
| Benign | Acc. (%)↓ | 76.33 | 48.82 | 58.44 | 74.68 | 74.68 | 76.58 | 68.26 |
| Baseline Zhang et al. (2022) | Acc. (%)↓ | 68.54 | 45.64 | 53.88 | 63.49 | 47.82 | 44.78 | 54.03 |
| | ASR (%)↑ | 14.95 | 11.72 | 10.26 | 20.24 | 40.76 | 44.59 | 23.75 |
| | FR (%)↑ | 24.02 | 25.48 | 19.40 | 28.09 | 48.06 | 51.60 | 32.78 |
| | ACR (%)↓ | 15.30 | 4.96 | 3.46 | 11.45 | 11.30 | 10.56 | 9.51 |
| w/ Ours | Acc. (%)↓ | 67.92 | 45.33 | 53.62 | 60.83 | 44.07 | 39.01 | **51.80** |
| | ASR (%)↑ | 15.75 | 11.95 | 10.65 | 23.74 | 45.37 | 51.63 | **26.52** |
| | FR (%)↑ | 24.83 | 25.31 | 19.60 | 31.61 | 52.22 | 57.86 | **35.28** |
| | ACR (%)↓ | 15.23 | 4.57 | 3.38 | 11.48 | 10.29 | 9.08 | **9.01** |

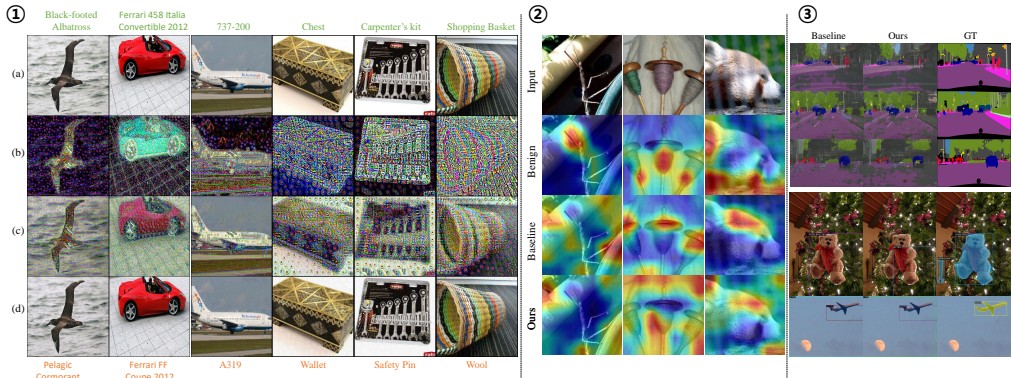

Figure 3: **Qualitative results.** Our semantically consistent generative attack successfully guides the generator to focus perturbations particularly on the semantically salient regions, effectively fooling the victim classifier. ①: (a) benign input image, (b) generated perturbation (normalized for visual purposes only), (c) unbounded adversarial image, and (d) bounded adversarial image across CUB-200-2011 Wah et al. (2011), Stanford Cars Krause et al. (2013), FGVC Aircraft Maji et al. (2013), and ImageNet-1K Russakovsky et al. (2015). The label on top (green) and bottom (orange) denotes the correct label and prediction after the attack, respectively. ②: We highlight that our method induces Grad-CAM Selvaraju et al. (2017) to focus on *drastically different regions* in our adversarial examples compared to both the benign image and the adversarial examples crafted by the baseline Zhang et al. (2022). Moreover, our approach *noticeably spreads and reduces the high activation regions* observed in the benign and baseline cases, enhancing the transferability of our adversarial perturbations. ③: Cross-task prediction results (SS on top, OD on bottom). Our approach further disrupts the victim models by triggering higher false positive rates and wrong class label predictions. See *Supp.* §E.4 for additional visualizations.

**Interplay with the baselines.** The pattern of gains across baselines in Table 3 is largely determined by the level at which each method probes the surrogate (logits, frequency domain, or intermediate features). By enforcing early-block semantic anchoring, our generator produces locally structured, object-aware perturbations. These perturbations move energy away from degenerate high-frequency noise and toward low- and mid-frequency components that align with objects and boundaries. This structural regularization couples most strongly with CNN-centric objectives. When combined with CDA, whose relativistic loss is defined directly on CNN logits, and with frequency- or CNN-prior-based baselines such as FACL and PDCL, our semantics-enhanced perturbations yield the largest improvements on CNN victims. ViT victims, whose global attention patterns and feature geometry differ more from the CNN surrogate, tend to show smaller or more localized changes.

In contrast, mid-layer feature-based attacks such as LTP, BIA, and GAMA rely on intermediate surrogate features that transfer more readily across architectures. These methods benefit more uniformly. Our generator-side semantics act as a complementary regularizer that sharpens feature-space separability on both CNN and ViT targets, with broadly positive or neutral effects. On image classification, the additional gains when combining with PDCL are modest. This behavior is consistent with a saturation regime in which the strong CLIP-space objective already induces powerful global semantic shifts and dominates

Table 5: **Ablation study on the targeted generator intermediate block and the proposed components.** Our self-feature consistency strategy on the early intermediate block outperforms matching other block features (a), and the generator trained with all of our components together performs best (b).

| Cross-Task | Task | Metric | Block: Early | Mid | Late | All | | $\mathcal{L}_{adv}$ ✓
PT ✗
MT ✓
$\mathcal{L}_{cons}$ ✓ | ✓ ✗ ✗ ✗ | ✓ ✓ ✗ ✓ | ✓ ✓ ✗ ✗ | ✓ ✗ ✗ ✗ |
|---|---|---|---|---|---|---|---|---|---|---|---|---|
| Model | CLS | Acc. (%)↓ | **44.13** | 45.76 | 45.79 | 51.13 | | **44.13** | 48.23 | 45.11 | 46.49 | 45.17 |
| | | ASR (%)↑ | **44.02** | 41.85 | 41.87 | 41.67 | | **44.02** | 33.37 | 42.80 | 41.02 | 42.55 |
| | | FR (%)↑ | **50.66** | 48.71 | 48.75 | 48.57 | | **50.66** | 44.08 | 49.47 | 46.99 | 49.38 |
| | | ACR (%)↓ | **8.32** | 8.59 | 8.66 | 8.60 | | **8.32** | 8.71 | 8.43 | 8.68 | 8.53 |
| Domain | CLS | Acc. (%)↓ | **47.10** | 50.95 | 49.03 | 51.13 | | **47.10** | 48.46 | 49.57 | 51.63 | 51.07 |
| | | ASR (%)↑ | **49.02** | 44.91 | 47.02 | 44.72 | (b) | **49.02** | 47.60 | 46.35 | 44.17 | 44.96 |
| | | FR (%)↑ | **51.66** | 47.67 | 49.75 | 47.50 | | **51.66** | 50.30 | 49.10 | 47.02 | 47.76 |
| | | ACR (%)↓ | **9.66** | 10.36 | 10.57 | 10.36 | | **9.66** | 9.99 | 9.89 | 10.73 | 10.58 |
| Task | SS | mIoU↓ | 23.40 | 24.10 | **22.82** | 23.92 | | 23.40 | 23.96 | 24.83 | 23.73 | 24.75 |
| | OD | mAP50↓ | **24.52** | 24.53 | 24.69 | 24.52 | | 24.52 | 24.73 | 24.55 | **24.41** | 24.72 |

the joint gradient. Even in this setting, our anchor still rebalances the perturbation spectrum. For localization-oriented downstream tasks such as detection and segmentation, the same local structural consistency produces noticeably larger cross-task improvements. This behavior suggests that Ours refines the global CLIP-driven semantic direction rather than competing with it.

**Ablation studies.** We conducted ablation studies on the intermediate block and our proposed components in Table 5. Across all cross-settings, we observe the highest gains with self-feature

consistency applied to the *early* block compared to those at other and all locations, insinuating the early block matching triggers generator features to place stricter constraint such that perturbations are progressively focused on or around the object.

We also observe performance gains with each component: $\mathcal{L}_{\text{adv}}$, MT, and $\mathcal{L}_{\text{cons.}}$, wherein our consistency of self-features on the intermediate features of the generator serves to widen the transferability gap even further. We attribute this improvement to explicit semantic alignment in the early blocks which complements the effect of implicit smoothing with MT. We also compare against the plain student-copy teacher, as indicated by plain teacher (PT), with and without $\mathcal{L}_{\text{cons.}}$, which underperforms our MT configuration. These results validate our hypothesis that anchoring perturbation synthesis on the early intermediate blocks consistently preserves the object semantics the most, and thus guides later blocks to concentrate noise on object-centric regions, maximizing transferability.

## 4.4 GENERATOR INTERMEDIATE BLOCK-LEVEL ANALYSIS

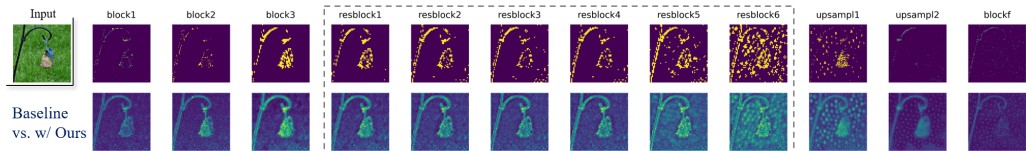

Figure 4: Visualization of generator intermediate block-level differences with the baseline Zhang et al. (2022): raw feature differences on bottom, and thresholded on top (normalized for illustration purposes only). With our generator-internal semantic consistency mechanism, we progressively guide adversarial perturbation to focus on the salient object regions initially and gradually disperse to surrounding background regions. See *Supp.* Fig.S10 for other baseline comparisons.

**Feature difference.** Following Zhang et al. (2022); Yang et al. (2024a) but generalizing the procedure to all generator layers as follows, for each layer $l$:

$$\text{Diff}(\mathbf{g}_{\text{baseline}}^{l,pooled}, \mathbf{g}_{\text{ours}}^{l,pooled}) = \begin{cases} 1, & \mathbf{g}_{\text{ours}}^{l,pooled} - \mathbf{g}_{\text{baseline}}^{l,pooled} > \tau_{\text{diff}}, \\ 0, & else, \end{cases} \quad \text{with} \quad \mathbf{g}_{(\cdot)}^{l,pooled} = \frac{1}{C}\left|\sum_C G_{\theta_{(\cdot)}}^l(\mathbf{x})\right|,$$

we visualized generator block-wise feature difference maps between each baseline with and without our method. At each block, we computed the difference by applying cross-channel average pooling to the activation tensor and then thresholding the resulting map to qualitatively emphasize the added perturbations. As shown in Fig. 4, the thresholded mask (row 1) and the raw feature difference map (row 2) jointly illustrate that, specifically within the targeted resblocks layers in our design, the adversarial signal concentrates on object-salient regions extracted by the preceding downsampling stages. Gradually into the later blocks, the generator learns to craft perturbation not only on object-salient regions but also regions closer to the background, generating more transferable noise. Compared to each baseline alone, our approach more strongly induces perturbations to better align with the semantic characteristics primarily in the intermediate residual blocks.

**Spectral energy comparisons.** To validate the early-block semantic anchoring hypothesis, we conducted a frequency-domain energy analysis of intermediate feature activations in Table 6, exploiting the link between spectral content and visual structure: low-frequency (LF) components encode coarse shapes and layouts, whereas high frequencies (HF) capture fine texture. By tracking the normalized low-band energy in every block before and after our method, we obtained a quantitative measure of how strongly each block preserves the coarse structure. Anchoring on the early blocks, rather than mid or late, consistently raises low-frequency energy and suppresses superfluous

Table 6: Spectral energy by band (Baseline→w/ Ours).

| | Band | Early | Mid | Late |
|---|---|---|---|---|
| CDA | Low (↑) | 0.82→**0.91** | 0.75→**0.97** | 0.77→**0.96** |
| →w/ Ours | High (↓) | 0.18→**0.09** | 0.25→**0.03** | 0.23→**0.04** |
| LTP | Low (↑) | 0.73→0.72 | 0.78→**0.79** | 0.95→0.75 |
| →w/ Ours | High (↓) | 0.27→0.28 | 0.22→**0.21** | 0.05→0.25 |
| BIA | Low (↑) | 0.56→0.56 | 0.53→**0.54** | 0.53→**0.58** |
| →w/ Ours | High (↓) | 0.44→0.44 | 0.47→**0.45** | 0.47→**0.42** |
| GAMA | Low (↑) | 0.57→**0.79** | 0.54→**0.60** | 0.56→**0.59** |
| →w/ Ours | High (↓) | 0.43→**0.21** | 0.46→**0.40** | 0.44→**0.41** |
| FACL | Low (↑) | 0.57→**0.73** | 0.52→**0.61** | 0.54→**0.59** |
| →w/ Ours | High (↓) | 0.43→**0.27** | 0.48→**0.39** | 0.46→**0.45** |
| PDCL | Low (↑) | 0.54→**0.62** | 0.51→**0.59** | 0.58→**0.59** |
| →w/ Ours | High (↓) | 0.46→**0.38** | 0.49→**0.41** | 0.42→**0.41** |

high-frequency noise downstream, confirming that our method targeting semantic consistency in the early intermediates more effectively propagates the same semantic scaffold through later blocks, yielding higher adversarial transferability.

The pattern reveals how anchoring affects generator's frequency bias. For band-wise relatively balanced models such as GAMA, the early-block anchor sharply increases low-frequency energy ($0.57 \to 0.79 \uparrow$), giving later blocks a clearer structural blueprint. When a baseline already over-emphasizes low frequencies, as in LTP whose late-block LF reaches 0.95, our method lowers that value to $0.75 \downarrow$, restoring HF detail. This spectral analysis thus reveals that anchoring on the early intermediate features results in perturbations that remain coarse semantic structures aligned and intact within the generator, thereby enhancing transfer effectively across unseen domains and architectures.

**Hyperparameter sensitivity.** We vary the EMA coefficient ($\eta$) and the consistency weight $\lambda_{\text{cons.}}$ and report the cross-setting transfer performance in Table 7. We observe a trade-off between optimizing classification and cross-task scores for both hyperparameters, as no single combination uniformly outperforms the rest. However, maintaining relatively high values for both tends to yield better performance, indicating that each module sufficiently contributes to the overall self-consistency mechanism. Based on this observation, we select $\lambda_{\text{cons.}} = 0.7$ and $\eta = 0.999$ as our default configuration, which provides the best overall balance across all cross-setting scenarios.

We define the "early", "mid", and "late" stages of the generator intermediates by grouping two consecutive residual blocks based on the observation that perturbations undergo the most noticeable qualitative changes over every two blocks. As illustrated in Fig. 2, the first two blocks (rows (a)–(b)) still closely track the benign image: coarse object shape, foreground–background separation, and large-scale texture are clearly preserved. The next two blocks (rows (c)–(d)) begin to introduce more pronounced distortions and fine-grained variations, while the final two blocks (rows (e)–(f)) predominantly add high-frequency details and noise-like patterns that are no longer easily interpretable as object-level structure. This makes the first two blocks a natural choice for enforcing semantic consistency: they are structurally well-formed and dominantly encode benign scene semantics, before most of the perturbation mass emerges in later stages.

Applying the temporal self-consistency loss only to block 1 or only to block 2 yields some benefits, but using both early blocks jointly provides a better balance across domains, models, and tasks. This pattern aligns with our intuition: anchoring both early blocks preserves coarse semantics at the onset of perturbation generation, which in turn biases later blocks to place perturbations along near-object regions rather than injecting unconstrained noise. As a consequence, the resulting perturbations align more closely with shared, object-level structure across architectures and datasets, thereby enhancing model- and data-agnostic black-box transfer. See Supp.§E for ablations of other components.

Table 7: **Hyperparameter ($\lambda_{\text{cons}}$, $\eta$) sensitivity and early-block selection. (Domain (Acc.), Model (Acc.), SS (mIoU), OD (mAP50)).**

| $\lambda_{\text{cons}}$ | 0.1 | 0.3 | 0.5 | 0.7 | 0.9 |
|---|---|---|---|---|---|
| Domain | 49.55 | 48.29 | 48.49 | 47.10 | 50.08 |
| Model | 44.84 | 44.80 | 44.68 | 44.13 | 45.89 |
| Task (SS) | 22.59 | 23.63 | 23.82 | 23.40 | 22.79 |
| Task (OD) | 23.96 | 24.36 | 24.78 | 24.52 | 24.19 |

| $\eta$ | 0.9 | 0.95 | 0.98 | 0.99 | 0.995 | 0.999 |
|---|---|---|---|---|---|---|
| Domain | 48.72 | 50.47 | 51.17 | 47.97 | 48.09 | 47.10 |
| Model | 45.04 | 45.56 | 45.84 | 44.79 | 44.70 | 44.13 |
| Task (SS) | 24.89 | 23.92 | 24.23 | 23.39 | 24.34 | 23.40 |
| Task (OD) | 24.83 | 24.51 | 24.73 | 24.26 | 24.48 | 24.52 |

| Block | 1 | 2 | 1 & 2 |
|---|---|---|---|
| Domain | 49.13 | 49.88 | 47.10 |
| Model | 47.62 | 48.82 | 44.13 |
| Task (SS) | 23.78 | 22.57 | 23.40 |
| Task (OD) | 24.47 | 24.15 | 24.52 |

## 5 CONCLUSION

In this paper, we introduce a semantically consistent generative attack leveraging the Mean Teacher and early-block semantic consistency to preserve the object semantics during perturbation generation, thus guiding it towards object-salient regions to markedly improve black-box transferability as in Fig. 5. With comprehensive evaluations across various models, domains, and tasks, we demonstrate object salient regions play a crucial role within the generator. See *Supp.* §E.6 for limitations and broader societal impact.

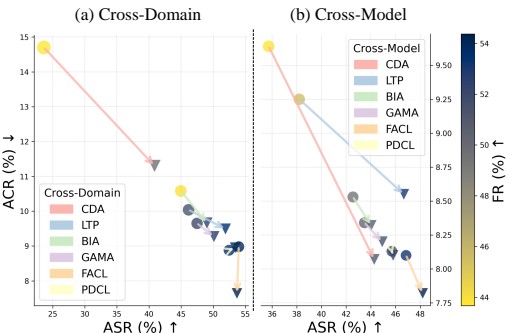

Figure 5: Our semantically consistent generative attack effectively exploits the generator intermediates to craft adversarial examples to enhance transferability from the baselines ($\bullet \to \blacktriangledown$) across domains (a) and models (b).

ACKNOWLEDGMENTS

This work was supported by the Technology Innovation Program (2410013617,20024355, Development of autonomous driving connectivity technology based on sensor-infrastructure cooperation) funded By the Ministry of Trade, Industry & Energy (MOTIE, Korea) and the Institute of Information & communications Technology Planning & Evaluation (IITP) grant funded by the Korea government (MSIT) (No. RS-2024-00457882, AI Research Hub Project).

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
