In this supplementary material, we provide comprehensive insights and detailed resources that complement our main manuscript. First, we provide an in-depth review of related work in Sec. A and additional materials in Sec. B. We then highlight the distinctions of our method that clearly distinguish it from the concurrent works in Sec. C, describe the reason for enhanced adversarial transferability in Sec. D, and provide additional experimental details and results in Sec. E aimed at supplementing our manuscript for a better understanding of our approach.

# A    RELATED WORK

## A.1    TRANSFER-BASED ADVERSARIAL ATTACKS

Transfer-based attacks exploit the empirical finding that adversarial perturbations crafted on one model often remain effective against others, even when architectures or training data differ. Early methods relied on iterative gradient-based strategies—momentum-integrated attacks (DI Xie et al. (2019), TI Dong et al. (2019)), input-diversity techniques (SI Lin et al. (2019), Admix Wang et al. (2021)), and strong baselines such as BIM Kurakin et al. (2016), PGD Madry et al. (2017), and C&W Carlini et al. (2019). These approaches enhance transferability via gradient smoothing, input transformations, and ensemble gradients, but incur heavy per-example optimization costs and often struggle against architectures that diverge significantly from the surrogate.

More recent work has introduced generative frameworks that train feed-forward generators to synthesize perturbations in a single pass. GAP Poursaeed et al. (2018), CDA Naseer et al. (2019), and LTP Nakka & Salzmann (2021) demonstrated orders-of-magnitude speedups with comparable transfer rates. Subsequent advances BIA Zhang et al. (2022b), GAMA Aich et al. (2022), FACL-Attack Yang et al. (2024a), PDCL-Attack Yang et al. (2024b), and NAT Nakka & Alahi (2025) have further improved robustness by integrating logit or mid-level layer feature divergence, frequency-domain constraints, text prompt-driven, and neuron-targeted losses. In a similar lineage, targeted attacks Li et al. (2020a); Wang et al. (2023); Zhao et al. (2023b); Fang et al. (2024b); Peng et al. (2025) aim to steer the classifer into mispredicting as the target class, fooling the decision boundary towards that targeted class. Rather than focusing solely on end-to-end optimization or domain-level constraints, we analyze the generator's intermediate feature hierarchy and preserve semantic fidelity in its early blocks to steer perturbations onto object-centric regions, thereby enhancing cross-model transfer effectiveness.

## A.2    GENERATIVE MODEL-BASED ATTACKS

Generative attacks recast adversarial synthesis as a learning problem, training an image-to-image network (e.g. GAN or encoder–decoder) to produce perturbations in one pass. GAP Poursaeed et al. (2018) pioneered a framework in which the generator outputs adversarial noise that is then added to the input. CDA Wang et al. (2022) extends this by training a transformation network that directly outputs adversarial examples. Subsequent works incorporate perceptual losses based on surrogate logits Nakka & Salzmann (2021) and mid-level surrogate features Zhang et al. (2022b); Nakka & Alahi (2025). Building on the feature-similarity loss of Zhang et al. Zhang et al. (2022b), more recent approaches leverage foundation models such as CLIP Radford et al. (2021) Aich et al. (2022); Yang et al. (2024b) and apply frequency-domain manipulations to surrogate features Yang et al. (2024a), further boosting transferability. Another concurrent work, dSVA Wu et al. (2025), innovates surrogate level manipulation by exploiting a dual self-supervised ViT ensemble features, which shows a different attack behavior as the previous works targeting a CNN surrogate. While prior frameworks prioritize perturbation realism or frequency characteristics, we explicitly target the generator's internal semantics by combining Mean Teacher-based smoothing with self-feature consistency on early blocks, preserving object contours and textures and concentrating adversarial perturbations in the most transferable regions.

## A.3 SELF-KNOWLEDGE DISTILLATION

Self-knowledge distillation (Self-KD) aims to train a model to refine its own representations without an external teacher. Pioneering works in this field, Born-Again Networks Furlanello et al. (2018) and Deep Mutual Learning Zhang et al. (2018), demonstrated that iterative self- and peer-distillation can improve generalization and robustness. Recent works Li et al. (2024); Yun et al. (2020) incorporate self-kd by aligning logits or intermediate features within its own network, or progressively updating the network Kim et al. (2021). In this paradigm of using a student-teacher framework, the Mean Teacher framework Tarvainen & Valpola (2017), originally developed for semi-supervised learning, aims to maintain a teacher as the exponential moving average of the student's weights, implicitly enforcing temporal consistency in predictions or feature maps. This EMA-based smoothing has been shown to reduce overfitting, stabilize training, and enhance domain invariance—properties that are directly relevant to generating perturbations that transfer across black-box models. Departing from classification-centric distillation, we integrate the Mean Teacher paradigm into a generative attack pipeline, using EMA to smooth intermediate features and enforcing hinge-based feature consistency on early blocks to preserve semantic integrity critical for cross-setting transferability.

## B ADDITIONAL RELATED WORK

**Iterative optimization-based attacks.** For years, iterative gradient-based attacks have become a cornerstone of adversarial research. Methods such as Projected Gradient Descent (PGD) Madry et al. (2017) extend the Fast Gradient Sign Method by applying multiple small, $\ell_\infty$-bounded steps; Momentum Iterative FGSM (MI-FGSM) Dong et al. (2018) further stabilizes updates via accumulated momentum; Diverse Input FGSM (DI-FGSM) Xie et al. (2019) injects random resizing and padding at each iteration; and Translation-Invariant FGSM (TI-FGSM) Dong et al. (2019) averages gradients over shifted inputs to enhance spatial robustness. More advanced variants even incorporate feature-space objectives to target intermediate representations Zhang et al. (2021).

**Generative model-based attacks.** In parallel, more efficient generative model-based attacks train a feed-forward image-to-image transformation network to synthesize perturbations in a single pass: Universal Adversarial Perturbations (UAP) Moosavi-Dezfooli et al. (2017) learn a single image-agnostic noise vector, Generative Adversarial Perturbations (GAP) Poursaeed et al. (2018) use a GAN framework to produce highly transferable noise maps (added to the input images), and AdvGAN Xiao et al. (2018) leverages GANs for image-dependent attacks that balance stealth and speed. Together, these two paradigms offer complementary trade-offs between precision, transferability, and inference efficiency.

In this vein, recent generative model-based untargeted attack methods Naseer et al. (2019); Nakka & Salzmann (2021); Zhang et al. (2022b); Aich et al. (2022); Yang et al. (2024a;b); Nakka & Alahi (2025) have further added techniques to enhance the transferability of the crafted adversarial examples by incorporating surrogate model's output logit-level and mid-level feature-level separation, frequency domain manipulation, vision-language model guidance, and heuristic selection of one effective neuron-level generator among a pool of multiple generators. However, none of these works have dealt with directly manipulating the generative feature space to improve the transferability of AEs. To address this, we uncover the correlation between generative features and adversarial transferability of the output AEs.

**Note:** Our semantically consistent generative attack does not redesign attack pipelines in a label-free or label-required setting. Rather, it operates orthogonally regardless of label availability: by regulating generator features, it can be plugged into any generative framework because its own objective is independent of label availability. This contrasts with prior work that mainly changes the adversarial loss, for example by moving from logit- to feature-based objectives.

**U-Net-based generator.** Along with ResNet He et al. (2016), U-Net Ronneberger et al. (2015) is another effective network architecture comprising a symmetric encoder–decoder with skip connections, fusing low- and high-level features to preserve fine-grained details, which are ideal when perturbations must tightly follow object boundaries. By contrast, a ResNet generator stacks residual blocks with identity shortcuts, thus building deep hierarchical representations that emphasize global context. Although U-Net decoders add computational overhead, they can produce sharper, pixel-accurate

noise, while ResNet backbones scale more efficiently and excel at generating broadly distributed perturbations. Ultimately, the choice of generator architecture hinges on the desired trade-off between pixel-level fidelity, attack transferability, and inference speed.

In the context of generative adversarial attack, GAP Poursaeed et al. (2018) first demonstrated that U-Net can serve as a perturbation generator with a lower inference time cost than that with ResNet. However, the authors of Poursaeed et al. (2018) also stated that ResNet, in general, outperforms U-Net in attack transferability. In this work, we demonstrate that our method of anchoring perturbation generation on early-intermediate features can also be applied to a different generator architecture than ResNet, namely U-Net.

Specifically, we note the differences between the U-Net and ResNet generators in detail. Due to the symmetric encoder-decoder design in the U-Net, there is only one feature block at the `bottleneck` that we can employ as the intermediate block feature, as opposed to six in ResNet. Given this architecture, we applied our semantic consistency mechanism on this `bottleneck` feature for the cross-task experiments in Table S1, where we observe consistent improvements in attack transferability with the addition of each of our components. While Given this observation, we remark that while U-Net can still be leveraged as a generative model for transfer-based attacks, further research on boosting U-Net-based attack transferability may be necessary.

Table S1: **Quantitative cross-task transferability results**. We report the average improvement ($\Delta$%p) for our components applied to different generator architectures and evaluated against semantic segmentation (mIoU $\downarrow$) and object detection (mAP50 $\downarrow$) models. MT denotes mean teacher, and better results in **boldface**.

| Cross-task | | Generator Arch. | | | | | |
| --- | --- | --- | --- | --- | --- | --- | --- |
| | | | U-Net | | | ResNet | |
| Method | Benign | Baseline | +MT | +MT+$\mathcal{L}_{cons.}$ | Baseline | +MT | +MT+$\mathcal{L}_{cons.}$ |
| SS DeepLabV3+ Chen et al. (2018) | 76.21 | 24.22 | +0.76 | **-2.92** | 23.89 | -0.79 | **-1.84** |
| SS SegFormer Xie et al. (2021) | 71.89 | 29.34 | -3.31 | **-3.81** | 25.60 | -0.78 | **-0.85** |
| Avg. | 74.05 | 26.78 | -1.27 | **-3.36** | 24.75 | -0.79 | **-1.35** |
| OD FRCNN Girshick (2015) | 61.01 | 27.51 | -0.05 | **-0.12** | 28.43 | +0.03 | **-0.09** |
| OD DETR Carion et al. (2020) | 62.36 | 23.92 | -3.18 | **-3.54** | 21.01 | -0.02 | **-0.29** |
| Avg. | 61.69 | 25.72 | -1.62 | **-1.83** | 24.72 | +0.01 | **-0.20** |

**Diffusion-based generative attacks.** Diffusion attacks generate imperceptible perturbations via iterative denoising Chen et al. (2024). Later studies extend this idea to conditional or Stable Diffusion for black-box transfer Liu et al. (2024; 2023b); Lei et al. (2025), treat diffusion mimicry as a defense target Xue & Chen (2024), and push attacks to prompts, retrieval, or fully synthesized images Ma et al. (2024); Huang & Shen (2025); Dai et al. (2024). All rely on high sampling steps, often with extra classifier or CLIP guidance, so inference is markedly slower and heavier than single-shot GAN or gradient generators. Our framework avoids that cost with a fast, lightweight alternative suited to real-time or large-scale threats. While diffusion-based approaches are certainly relevant in the literature and offer strong performance, yet still incur high inference-time costs due to iterative sampling, our approach alleviates these costs and ensures faster and more practical deployment.

## C  DISTINCTIONS OF OUR METHOD

**Purpose.** *Generator-centric regularization vs. surrogate-centric ILP.* Prior ILP works typically optimize pixels against a feature map from a *static surrogate model*. We instead ask how to regularize the synthesis process *within a learnable generator* to enforce semantic stability. This shifts the focus from (external) surrogate optimization (followed by iterative ILP updates) to *internal* regularization inside the generator. We summarize the distinctions of SCGA (Sec. §3 in the main paper) below.

- **Attack framework.** Prior surrogate–centric ILPs update pixels iteratively under a surrogate classifier, typically requiring many small steps and sometimes fine-tuning existing adversarial examples. Our approach is generator–centric: a single forward pass through a learnable perturbation generator uses the full perturbation budget at once, without any fine-tuning of existing AEs. This yields a qualitatively different attack pipeline with essentially zero additional inference cost.
- **Source of guidance features.** Conventional ILPs extract guidance from fixed intermediate layers of a frozen surrogate. We instead draw guidance from the intermediate representations *inside* the learnable perturbation generator, thus internalizing what the model should preserve or alter. Semantic information is therefore obtained from the generator's own dynamics rather than from an external model.
- **Role of intermediate features (optimization objective).** In surrogate–centric ILPs, the surrogate's mid-level features directly define the objective, commonly by maximizing the feature distance between a benign image and its adversarial counterpart. In our method, the generator's intermediate

features are regularized to stabilize core semantics via an EMA teacher, providing self-guided regularization during noise synthesis. Whereas ILPs feed both benign and adversarial images to the surrogate, our generator consumes only the benign image, producing internal features with different embedded semantics.

- **Driving factor for transferability.** Classic ILPs largely seek stronger disturbance in surrogate mid-layer features to improve transfer. Our method maintains salient object semantics throughout the noise-generation path, emphasizing semantic consistency in the generator's early blocks. This distinction shifts the driver of transfer from surrogate divergence to internally consistent noise generation.
- **Plug-in compatibility.** Traditional ILP attacks are usually self-contained and not designed for modular composition. Our regularizer is a drop-in module that plugs into existing generators (e.g., BIA, GAMA) without changing their inference routine. It thus offers a general-purpose axis of improvement for generative attacks.

The seminal *Intermediate-Level Attack (ILA)* Huang et al. (2019) begins with a baseline perturbation and amplifies its change at a single mid-layer of a frozen classifier, boosting cross-model transfer. *ILA++* Li et al. (2020b) maximizes the scalar projection onto a learned discrepancy vector, making the amplification direction data-adaptive. *ILPD* Li et al. (2023) folds amplification into one stage and adds a decay schedule to damp spurious directions. *TAP* Zhou et al. (2018) enlarges clean–adversarial feature distances while imposing a smoothness prior on the noise. All rely on a *fixed surrogate classifier* to define the intermediate layer and therefore lose potency when the victim architecture or modality changes.

*Generator-based ILPs.* A parallel line supervises a generator with surrogate features. *LTP* splits a chosen surrogate layer into class-consistent and class-inconsistent channels and steers the generator toward the latter. *BIA* manipulates early surrogate features to weaken low-level cues across domains. Recent works impose high-level semantic or contrastive losses (often via CLIP) on the *output image* but leave the generator's internal layers largely unconstrained.

*Our focus on generator internals.* We align early-block feature maps to an EMA-smoothed teacher and distill onto the student, preserving coarse structure *before* any surrogate-level adversarial loss is applied. This internal alignment is agnostic to the choice of surrogate or adversarial loss, and remains effective when transferring to unseen architectures. Empirically, adding our self-feature consistency to representative generator attacks further *lowers average accuracy across all four cross-setting protocols*, indicating complementarity rather than redundancy.

**Structure.** Our method stands distinct from the focus of existing generative attacks in that we delve into the generator feature space, rather than the surrogate model space, as categorized in Table S2. Previous works have targeted various stages of the generative attack pipeline Poursaeed et al. (2018); Naseer et al. (2019); Nakka & Salzmann (2021); Zhang et al. (2022b), including input data augmentation, pixel-level perturbation, surrogate model's logit- and feature-level manipulations. Nonetheless, no work has yet explicitly manipulated the internal features of the generative model to enhance transferability. In this work, we investigate how internal feature representations within generative models can be harnessed to enhance the transferability of AEs.

Table S2: **Our method distinction.** Comparison of transfer-based generative adversarial attacks, highlighted by the method's targeted stage in the training pipeline (in order from left to right), and GT label requirement.

| Attack | Input data aug. | Generator feature-level | Perturbed image-level | Surrogate mid-level layer feature-level | Surrogate output logit-level | GT label required? |
|---|---|---|---|---|---|---|
| GAP Poursaeed et al. (2018) | - | - | - | - | ✓ | ✓/✗ |
| CDA Wang et al. (2022) | - | - | - | - | ✓ | ✓/✗ |
| LTP Nakka & Salzmann (2021) | - | - | - | ✓ | - | - |
| BIA Zhang et al. (2022b) | - | - | ✓ | ✓ | - | - |
| GAMA Aich et al. (2022) | - | - | - | ✓ | - | ✓ |
| FACL Yang et al. (2024a) | ✓ | - | - | ✓ | - | - |
| PDCL Yang et al. (2024b) | - | - | - | ✓ | - | ✓ |
| Our focus | - | ✓ | - | - | - | - |

**Distinction from ensemble-based approach.** We assert that our proposed method is intended as a complementary add-on to the single perturbation generator already employed by existing generative attacks, rather than as a separate generator. As such throughout our experiments, we meticulously focus on the effect of attaching our method onto existing perturbation generators and show that our complementary add-on exhibits attack-beneficial effects. Though we do agree that an ensemble of different generators may be an interesting direction of work, the scope in our work focuses more on a complementary add-on for existing adversarial generative perturbation works.

**Full algorithm.** For a full picture of training and inference stage of our algorithm, we provide in Alg. 1 outlining the procedure in both stages.

---

**Algorithm 1:** Full pseudo-code of SCGA

**Data:** Training dataset $\mathcal{D}_{src}$

**Input:** Generator $\mathcal{G}_\theta(\cdot)$, a surrogate model trained on source data $\mathcal{F}^s(\cdot)$, projector $\mathcal{P}(\cdot)$, perturbation budget $\epsilon$

**Output:** Optimized teacher perturbation generator $\mathcal{G}_{\theta'}(\cdot)$

1   `Training:`
2   Initialize generators:
3   student $\mathcal{G}_\theta(\cdot) \leftarrow$ random init., teacher $\mathcal{G}_{\theta'}(\cdot) \leftarrow \mathcal{G}_\theta(\cdot)$
4   **repeat**
5      Randomly sample a mini-batch $x_i$ from $\mathcal{D}_{\text{src}}$
6      Acquire student generator intermediate features:    $\mathbf{g}_i \leftarrow \mathcal{G}_\theta^{\text{enc}}(x_i)$
7      Acquire teacher generator intermediate features:    $\mathbf{g}'_i \leftarrow \mathcal{G}_{\theta'}^{\text{enc}}(x_i)$
8      Generate unbounded adversarial examples from student generator intermediate features:
      $\tilde{x}_i^{\text{adv}} \leftarrow \mathcal{G}_\theta^{\text{dec}}(\mathbf{g}_i)$
9      Bound (project) $\tilde{x}_i^{\text{adv}}$ using $\mathcal{P}$ within the perturbation budget such that $||\mathcal{P}(\tilde{x}_i^{\text{adv}}) - x_i||_\infty \le \epsilon$ to obtain
      $x_i^{\text{adv}}$
10     Forward pass $x_i$ and $x_i^{\text{adv}}$ through the surrogate model, $\mathcal{F}^s(\cdot)$ at layer $k$, to acquire $f_i^{\text{benign}}, f_i^{\text{adv}}$
11     Compute loss using $f_i^{\text{benign}}, f_i^{\text{adv}}, \mathbf{g}_i, \mathbf{g}'_i$:    $\mathcal{L} = \mathcal{L}_{\text{adv}} + \lambda_{\text{cons.}} \cdot \mathcal{L}_{\text{cons.}}$      `// Eq.??`
12     Update student generator parameters via backpropagation
13     EMA update teacher weights with student weights:    $\theta \mapsto \theta'$       `// Eq. ??`
14 **until** $\mathcal{G}_\theta(\cdot)$ *converges*
15 `Inference:`
16 Acquire an input image sample, $x_{\text{test}}$
17 Forward pass $x_{\text{test}}$ through the trained teacher, $\mathcal{G}_{\theta'}(\cdot)$, to obtain an unbounded adversarial example, $\tilde{x}_{\text{test}}^{\text{adv}}$
18 Bound (project) $\tilde{x}_{\text{test}}^{\text{adv}}$ using $\mathcal{P}$ within $\epsilon_{\text{test}}$ to obtain $x_{\text{test}}^{\text{adv}}$
19 Forward pass $x_{\text{test}}, x_{\text{test}}^{adv}$ through pre-trained victim model $\mathcal{F}^t(\cdot)$ to obtain $p_{\text{test}}^{\text{out}}, p_{\text{test}}^{\text{out,adv}}$, respectively
20 Compute metric scores by comparing the $\arg\max\{p_{\text{test}}^{\text{out,adv}}\}$ against $\arg\max\{p_{\text{test}}^{\text{out}}\}$ or GT label

---

## D   REASONS FOR TRANSFERABILITY

As our work is motivated by the semantic variability across the intermediate blocks within the generator, we look further into how our early-block semantic anchoring drives the observed phenomenon in Sec. §2.2 of the main paper. Our empirical analysis of intermediate feature activation maps from existing ResNet-based generators reveals that coarse, object-salient regions consistently emerge in the early residual blocks, and appear even more pronounced in models with higher black-box transferability. This insight suggests that these early-block features play a pivotal role in shaping perturbations. To capitalize on this, we anchor our adversarial noise generation to the clean image's semantic structure at these early stages. Lacking explicit semantic priors to retain the semantic integrity of the benign input images, we introduce a Mean Teacher mechanism: by maintaining an exponential moving average (EMA) of the student generator's weights, the teacher generator yields temporally smoothed features that are *largely free of adversarial noise.* We then fully leverage the Mean Teacher framework by further imposing a self-feature consistency loss between the student generator's and the teacher's early-intermediate block activations, *filtering out spurious noise while preserving the coarse object shapes and boundaries* present in the teacher generator features. This strict semantic-consistency constraint focuses perturbation power on object-salient regions, thereby enhancing transferability without sacrificing efficiency.

In the figure below (Fig. S1), we directly compare the feature activation maps and the added adversarial noise per block (absolute difference of the input and output of each block) of Ours against the baseline Zhang et al. (2022b). We particularly focus on the intermediate residual blocks ("Residual Learning"), as most of the adversarial noise is generated in these blocks Zhang et al. (2022b), and the preceding ("Downsampling" layers) and succeeding blocks ("Upsampling" layers) serve to simply adjust the spatial resolution of the feature maps.

We preserve semantic integrity in the early blocks because these layers capture the coarse structure of the object, such as boundaries and shapes. By aligning the student model's early block activations to a teacher reference, we remove incidental details and initial noise. This alignment compresses feature magnitudes and lowers the measured semantic quality in those early blocks. However, that simpler representation allows the generator to focus on stronger and more widespread noise in the later layers.

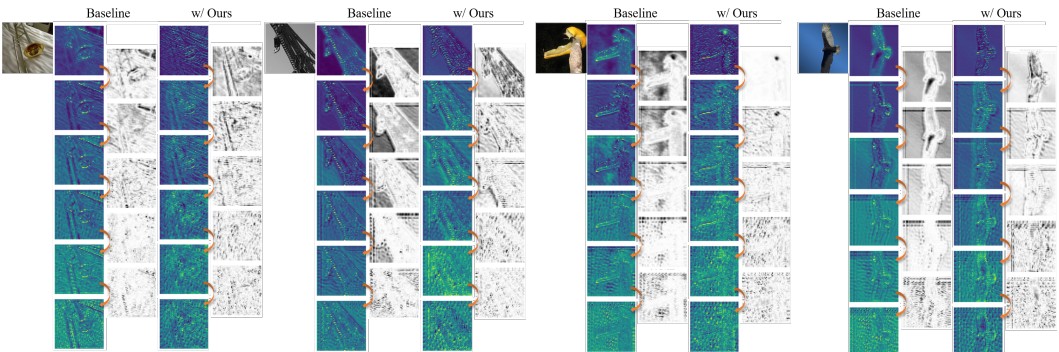

Figure S1: Comparison of the input image (column 1), the feature activation maps, and the added adversarial noise between two residual blocks (absolute difference between the feature maps of two residual blocks) of Ours (columns 4, 5) against the baseline Zhang et al. (2022b) (columns 2, 3). Although the baseline's feature maps more vividly emphasize object boundaries and contours, this focus actually prevents perturbations from appearing in those highlighted regions. By contrast, our method produces relatively less pronounced early features than those of the baseline, yet focuses more perturbation power on object-salient regions towards the later blocks than the baseline, thereby allowing adversarial noise to be dispersed directly on and around those salient regions, as opposed to the baseline. For the feature activation maps and the added noise, the brighter and darker (respectively), the higher the value.

**Deliberate compression of early features.** We enforce semantic consistency in the early blocks to focus the generator on true object outlines. Aligning student features to a smoothed teacher strips away incidental detail and any initial noise, which reduces the average magnitude of feature activations, and thus appears to degrade early-block semantics. That lean representation then lets the network concentrate its available capacity deeper in the intermediate blocks, where it produces stronger and more widely dispersed perturbations. When comparing the absolute difference between two feature maps in Fig. S1, we see that the baseline avoids object-salient regions and restricts noise to the peripheral regions, whereas our approach applies noise across the entire image, including the object itself. Although the early semantics seem more degraded relative to the baseline, this deliberate compression of early features enables a broader attack on diverse features. Although the baseline's early blocks may exhibit stronger semantic activations, our method's slightly muted early features enable a broader and more effective perturbation distribution towards the later intermediate blocks, thus achieving higher transferability than the baseline. Crucially, we observe that *the baseline's finely detailed early-block semantics add little benefit; retaining only the coarse semantic outline is sufficient to guide highly transferable perturbations.*

**Comparison of feature activation maps by block.** Compared to the baseline and to ablations that apply semantic consistency in mid layers, late layers, or across all layers, our approach of enforcing semantic consistency in the early block produces the most effective and transferable perturbations (Fig. S2). When we inspect feature activation maps, we see that preserving the coarse semantic structure in the earliest layers anchors the noise to object-salient regions that tend to be shared across different models. This focus prevents the generator from wasting capacity on irrelevant details and guides it to concentrate its attack on universally important features. By contrast, semantic consistency applied later or across all intermediate blocks causes noise to be rather sparse or dispersed over

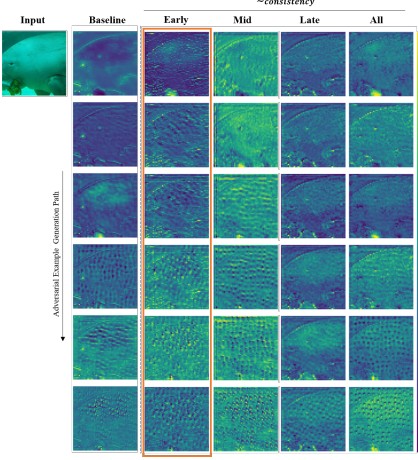

Figure S2: Comparison of feature activation maps by block.

regions away from those core cues, resulting in weaker
transfer performance.

**Quantifying semantic information.** The very concept of *semantic information* is a deep and challenging topic within information theory and computer vision, lacking a single, universally accepted, and operational definition. For example, foundational theories define meaning through the truthfulness of a representation to reality Floridi (2002), or the grammatical structure of its components Zhu et al. (2007). While profound, these conceptual frameworks remain abstract. Attempts to formalize a quantifiable theory of semantics, such as the information-theoretic treatise by D'Alfonso (2011), highlight this challenge. While providing a valuable formal basis, such work has yet to yield operational metrics that can be optimized directly within deep learning models from the first principles.

Given this well-documented gap between theory and practice, the standard and most rigorous approach in computer vision is to probe semantic coherence using concrete, multi-faceted empirical methods. To build a robust foundation for our *semantic anchoring on the early generator intermediate block* claim, we employed two orthogonal analyses to test our hypothesis from different perspectives:

- **Energy-Level Stability (Frequency Domain)**: To verify the *coarse* structure embedded in the *low* frequency components, we computed the spectral energy ratio compared to the total by band following Eq. S1 (cutoff=0.2). Our spectral analysis confirms that our method preserves low-frequency energy more effectively than the baseline. This provides evidence that the coarse visual structure of the image is maintained throughout the generation process.
- **Object-Level Coherence (Semantic Object)**: A closer look into semantic foreground IoU stability can be viewed in a complementary manner alongside the spectral energy analysis. We conducted k-means clustering on each intermediate block features, generated binary (object/background) object masks for each block features, and measured their IoU against the ground truth pixel-wise labels of ImageNet-S Gao et al. (2022) dataset. We then calculated the standard deviation of these IoU scores across the generator's blocks–a metric we term as *vararibility* in IoU. A lower variability score signifies that the object's core structure is preserved more coherently across generator blocks during generation. Our method achieves a significantly lower variance than the baseline, confirming superior object-level stability.

$$F(u,v) = \sum_{m=0}^{H-1} \sum_{n=0}^{W-1} f[m,n] \, \exp\left( -\mathrm{i}2\pi\big(\tfrac{um}{H} + \tfrac{vn}{W}\big)\right), \;\; \text{for feature map } f \qquad \text{(FFT)} \qquad \text{(S1)}$$

$$S(u,v) = \big|F(u,v)\big|^2, \qquad \text{(Power spectrum)}$$

$$R_{\mathrm{low}}(\tau) = \frac{\sum\limits_{k,\ell} \mathbf{1}\big\{\sqrt{k^2+\ell^2} \leq \tau\big\} S(k,\ell)}{\sum\limits_{k,\ell} S(k,\ell)},$$

$$R_{\mathrm{high}}(\tau) = 1 - R_{\mathrm{low}}(\tau).$$

Although a first-principles theory of semantics remains an open challenge for the broader scientific community, our work provides a strong and verifiable empirical basis for our claim. The combined evidence from both energy-level (frequency) and object-level (foreground IoU) analyses demonstrates that early-generator block anchoring effectively maintains semantic content, commonly shared by diverse architectures across domains. This coherence, in turn, provides a clear explanation for the observed boost in adversarial transferability across architectures, domains and tasks.

**Marginal improvement due to shared objective.** We acknowledge that PDCL Yang et al. (2024b) is the strongest amongst the existing generative attack baselines. Our approach is complementary yet convergent with ideals of PDCL and GAMA Aich et al. (2022) in attacking the semantics. While PDCL and GAMA leverage CLIP Radford et al. (2021)-based semantics alignment in surrogate models, our method uniquely targets semantic distortion within the internal feature hierarchy of the generator via Mean Teacher smoothing and self-feature consistency. Though previous methods and our method work in different operational spaces (internal feature hierarchy of the generator vs surrogate model), the aligned goals as well as the generator and surrogate features interacting

through backpropagation leads to diminishing returns when combined. This ceiling effect explains why numerical improvements over PDCL are modest in cross-model settings.

However, we stress that while the cross-model results exhibit only minor degradation, our method yields more pronounced gains over the PDCL baseline in cross-domain and cross-task settings, where data and model shifts are more severe. Notably, in cross-domain results, while ASR, FR, and Accuracy consistently improve, we observe a drop in ACR with ours compared to PDCL alone. This suggests that PDCL's strong surrogate alignment may overfit to high-level features, inadvertently leading to corrected predictions under domain shift. Our ACR metric uniquely captures this effect, which goes unnoticed by conventional metrics. Ultimately, we believe this demonstrates that our method contributes in more realistic and challenging generalization settings (cross-domain, cross-task) than in the saturated cross-model setting. We acknowledge that this observation remains as a potential limitation of our method when layered atop strong vision-language aligned baselines such as PDCL. Thus, even when improvement margins are modest, our work introduces a conceptually novel and generator-centric mechanism that is compatible with PDCL, helps expose previously overlooked behaviors (via ACR), and delivers consistent and transferable performance boosts, especially under the strictest black-box constraints.

In summary, our ablations reveal two situations in which the proposed early-block self-feature consistency provides only modest gains:

- **Transfer to dense–prediction task**: When evaluated on tasks that require pixel-level precision such as SS, our hinge-based self-consistent generator produces less disruptive AEs. In the cross-task setting, DeepLabV3+ IoU decreases by 1.35%p and Faster R-CNN mAP50 falls by 0.20%p. Figs. S8 and S9 show that the perturbations suppress large regions but rarely erode thin boundaries or very small objects. Because the hinge margin halts gradients once coarse alignment is achieved, optimization focuses on low-frequency structure, which benefits cross-model and cross-domain transfer more than cross-tasks that require fine-grained spatial details.

- **Combination with CLIP-guided attacks**: At $\epsilon_{\text{test}} = 10$, augmenting GAMA with ours lowers cross-domain acc. by 2.47%p but trims model accuracy by only 1.18%p and mAP50 by 0.16%p; PDCL changes are even smaller, all $\leq 1\%$p. Because GAMA and PDCL already use a CLIP image-text similarity loss that enforces high-level global semantic alignment, our early-block anchor largely overlaps their objective, leaving limited headroom. Future work could involve adaptive margins that complement rather than replicate CLIP guidance.

**Potentially compatible foundation model alternatives.** While CLIP is a powerful vision-language foundation model, its representations are heavily shaped by global semantic alignment across modalities. This design prioritizes coarse, aligned image-text associations and often abstracts away local or mid-level spatial structure, which is precisely the kind of structure our generator-centric perturbations are designed to manipulate. As a result, our structure-aware perturbations, which emphasize consistency in early generator layers (e.g., edges, object boundaries), may be less impactful when paired with CLIP-based classifiers, leading to the observed marginal improvements.

Nonetheless, our method and CLIP-guided approaches share a high-level goal: disrupting semantic integrity in learned feature spaces. CLIP achieves this via global cross-modal alignment, whereas our method directly injects semantic distortions during perturbation synthesis. This overlap may lead to saturation effects when stacking the two, particularly when the downstream classifier lacks spatial sensitivity. To this end, we list below several foundation models that emphasize semantic structure, rather than purely global or language-aligned semantics may synergize better with our approach. Promising alternatives include:

- **DINOv2** Oquab et al. (2023) and **iBOT** Zhou et al. (2022): self-supervised, image-only models that retain strong spatial awareness and token-level feature localization. Their patch-level attention maps are often more structure-sensitive in early layers.

- **SAM** (Segment Anything) Kirillov et al. (2023) and **OWL-ViT** Minderer et al. (2022): models trained with region-aware objectives or bounding box supervision, offering spatial grounding across scales.

- **SEEM** Zou et al. (2023), **OpenSeg** Liang et al. (2023), and other dense prediction models: trained for segmentation or region-level tasks, these retain fine spatial resolution across semantic hierarchies.

In addition, we would like to highlight **TokenCut** Shin et al. (2022), which encourages part-aware decomposition in early transformer layers via spectral clustering. Similarly, **SegViT** Zhang et al. (2022a) and **DenseCLIP** Rao et al. (2022) enhance token-level representations with structural priors for region-specific prediction. These architectures inherently encode mid-level part semantics and object contours, aligning more naturally with our method's emphasis on preserving and perturbing meaningful structure. Overall, these models offer varying degrees of structure sensitivity, depending on task supervision and architectural inductive biases. We view this as both a current limitation and a valuable opportunity: as foundation models increasingly integrate dense, structured objectives, our generator-centric perturbations could target and amplify the resulting spatial semantics in a more compatible way.

**Practical value of the Accidental Correction Rate (ACR).** Here, we highlight the usefulness of the proposed ACR metric as previously outlined in Table 1 of the main paper. ACR was introduced specifically to expose cases in which a perturbation repairs an error already present in the clean prediction: a behavior that Accuracy, Attack-Success Rate (ASR) and Fooling Rate (FR) cannot disentangle. In this light, we argue that *this metric is of significance for both adversary and defender*, as it focuses on the reliability of the attack and haphazard defense due to an inadvertent transition to *correct* predictions, respectively. In Table S3, we report all four metrics for increasing test budgets $\epsilon_{\text{test}}$, averaged over the same cross-domain / cross-model settings and pinpoint key observations:

- **Complementary role to ASR.** At $\epsilon = 4$, ACR (incorrect→correct) is 14% while ASR (correct→incorrect) is 8%, yielding a net gain of 6% correct predictions. Because ASR accounts for only harmful flips, it overlooks this positive balance; ACR makes it explicit.

- **ACR is non-monotonic, unlike Acc/ASR/FR.** It peaks at $\epsilon = 4$ and then falls as stronger noise overwhelms corrective effects. This trend offers a very different view from the defender's side: evaluation should consider not only how many errors an attack creates but also how many it inadvertently corrects.

- **Actionable insight.** Defenders might tolerate or even harness low-budget perturbations that raise ACR, while attackers in safety-critical settings may need to penalize accidental corrections to avoid unintentionally improving model performance.

Table S3: Performance evaluation under different $\varepsilon$ test values for other metrics than the accuracy.

| $\varepsilon_{test}$ | Accuracy ↓ | ASR ↑ | FR ↑ | ACR ↓ |
|---|---|---|---|---|
| 2 | 89.93 / 74.67 | 1.88 / 3.32 | 3.70 / 6.94 | 9.33 / 6.08 |
| 4 | 84.95 / 70.34 | 7.90 / 10.22 | 11.01 / 16.42 | 14.37 / 9.26 |
| 8 | 60.13 / 53.82 | 30.94 / 32.67 | 38.09 / 39.35 | 11.95 / 9.36 |
| 10 | 47.10 / 44.13 | 49.02 / 44.02 | 51.66 / 50.66 | 9.66 / 8.32 |
| 16 | 23.00 / 30.29 | 75.18 / 62.89 | 76.46 / 67.03 | 5.56 / 5.83 |

ASR alone offers an incomplete and sometimes misleading view of adversarial attack effectiveness. It is **blind to beneficial flips**, failing to account for cases where perturbations actually improve predictions by turning an incorrect label into the correct one. This can lead to **inflated ASR scores under label noise**, where attacks appear successful by correcting existing errors. Furthermore, **ASR lacks granularity over the type of prediction change**, treating all flips as equivalent. This becomes especially problematic in cross-model comparisons, where identical ASR values can conceal substantial differences in behavior, particularly in how often attacks correct the clean model's mistakes. **Among existing metrics, ASR captures only correct→incorrect transitions, FR aggregates all changes without distinction, and only ACR isolates the critical incorrect→correct transitions.**

**Summary of our approach.** We further summarize our findings and approach as follows. We empirically discover that existing generative works preserve the semantic integrity of the benign input image at the early intermediate blocks better than the later blocks, under the assumption that the downsampling blocks merely serve as feature extractors Zhang et al. (2022b). To better structure adversarial noise generation in the intermediate blocks, we formulate our method to maintain at least coarse semantic structures in the *early* intermediate blocks, thereby yielding much more perturbed features towards the end of the intermediate stage, which then results in enhanced transferability of the crafted AEs compared to the baseline. Through extensive cross-setting evaluations, we validate our approach of tuning the progression of adversarial noise generation in the generator's feature level

as a compatible method to the existing generative attack framework Naseer et al. (2019); Nakka & Salzmann (2021); Zhang et al. (2022b); Aich et al. (2022); Yang et al. (2024a;b); Nakka & Alahi (2025) without much overheads.

# E    EXPERIMENTAL DETAILS

## E.1    EVALUATION SETTINGS

We define three black-box evaluation scenarios that differ in the attacker's knowledge of target data and models. In the cross-model setting, the adversary has access to the same data distribution used to train the unseen target models but must craft attacks using a substitute model rather than querying the targets directly. In the cross-domain setting, the attacker works solely with out-of-domain data and has no access to target-domain datasets (e.g., CUB-200-2011 Wah et al. (2011), Stanford Cars Krause et al. (2013), FGVC Aircraft Maji et al. (2013)) or the ability to query target-domain models such as ResNet-50 He et al. (2016), SE-Net, or SE-ResNet101 Hu et al. (2018). Finally, in the cross-task setting, likewise, the adversary is completely agnostic to the target's data, models, and even the task itself, representing the strict black-box challenge.

**Victim model specifications.**    we selected a total of 22 different model architectures that span from CNNs to Vision Mamba variants, whose pre-trained model weights are available openly through TorchVision Marcel & Rodriguez (2010), Timm Wightman, and the proprietary GitHub repositories. We list the sources in Table S4.

## E.2    EVALUATED MODELS

**Victim models.**    For *cross-model* evaluation, we employ ImageNet-1K (224 × 224 resolution, 1,000 classes) pre-trained classification models of various architectures with their publicly available model weights. We source the pre-trained models from TorchVision Marcel & Rodriguez (2010) and Timm Wightman libraries. Compared to previous approaches Nakka & Salzmann (2021); Yang et al. (2024a;b) demonstrating cross-model architecture transferability, we expand the evaluation to a wider scope of target model architectures for enhanced architecture-agnostic transferability. We tested our attack against a total of 21 different model architectures (11 CNN-based He et al. (2016); Huang et al. (2017); Szegedy et al. (2016); Radosavovic et al. (2020); Tan et al. (2019); Iandola et al. (2016); Tan & Le, 6 ViT-based Tu et al. (2022); Liu et al. (2021); Touvron et al. (2021); Bao et al. (2021); Cai et al. (2022), 2 Mixers Tolstikhin et al. (2021); Trockman & Kolter (2022), and 2 Vision Mamba-based Zhu et al. (2024); Hatamizadeh & Kautz (2025)) for cross-model evaluations.

For *cross-domain* evaluation, we validate our attack against three different models (i.e. ResNet50 He et al. (2016), SE-Net and SE-ResNet101 Hu et al. (2018)) pre-trained on fine-grained datasets, CUB-200-2011 Wah et al. (2011) (200 classes), Stanford Cars Krause et al. (2013) (196 classes), FGVC Aircraft Maji et al. (2013) (100 classes), of 448 × 448 resolution. For *cross-task*, we select a CNN-based and a ViT-based model for each task of semantic segmentation (SS) and object detection (OD), whose pre-trained weights are openly accessible as with the ImageNet-1K pre-trained weights. Specifically, we test against DeepLabV3+ Chen et al. (2018) and SegFormer Xie et al. (2021) for SS and Faster R-CNN Girshick (2015) on `detectron2` and DETR Carion et al. (2020) for OD. We validate on Cityscapes Cordts et al. (2016) and COCO'17 Lin et al. (2014) for SS and OD tasks, respectively.

**Against robust models.**    We tested our attack against robust models, i.e. adversarially trained Inception-V3 Kurakin et al. (2016), ViT Dosovitskiy et al. (2021) and ConvNeXt Singh et al. (2023) models, and robust input processing methods such as JPEG (75%) Guo et al. (2017a), bit reduction (BDR; 4-bit) Xu et al. (2018) and randomization (R&P) Xie et al. (2018) in Table 4.

## E.3    IMPLEMENTATION DETAILS

Throughout the experiments, we train the perturbation generator with $\epsilon = 10$ using data from ImageNet-1K Russakovsky et al. (2015) containing 1.2 M natural images of 224×224 resolution, following Poursaeed et al. (2018); Naseer et al. (2019); Nakka & Salzmann (2021); Zhang et al.

Table S4: Sources of victim models used to evaluate the attack performance, grouped by task.

| Task | Victim Model | Source |
|---|---|---|
| Image Classification | (a) ResNet50 He et al. (2016) | TorchVision Marcel & Rodriguez (2010) |
| | (b) ResNet152 He et al. (2016) | TorchVision Marcel & Rodriguez (2010) |
| | (c) Dense121 Huang et al. (2017) | TorchVision Marcel & Rodriguez (2010) |
| | (e) Dense169 Huang et al. (2017) | TorchVision Marcel & Rodriguez (2010) |
| | (f) InceptionV3 Szegedy et al. (2016) | TorchVision Marcel & Rodriguez (2010) |
| | (g) RegNetY Radosavovic et al. (2020) | TorchVision Marcel & Rodriguez (2010) |
| | (h) MNAsNet Tan et al. (2019) | TorchVision Marcel & Rodriguez (2010) |
| | (i) SqueezeNet Iandola et al. (2016) | TorchVision Marcel & Rodriguez (2010) |
| | (j) EfficientV2 Tan & Le | TorchVision Marcel & Rodriguez (2010) |
| | (k) ConvNeXt-B Liu et al. (2022) | TorchVision Marcel & Rodriguez (2010) |
| | (l) ResNeXt Xie et al. (2017) | TorchVision Marcel & Rodriguez (2010) |
| | (m) ViT-B/16 Dosovitskiy et al. (2021) | TorchVision Marcel & Rodriguez (2010) |
| | (n) ViT-L/16 Dosovitskiy et al. (2021) | TorchVision Marcel & Rodriguez (2010) |
| | (o) Swin-B/16 Liu et al. (2021) | TorchVision Marcel & Rodriguez (2010) |
| | (p) DeiT-B Touvron et al. (2021) | Timm Wightman |
| | (q) BEiT-B Bao et al. (2021) | Timm Wightman |
| | (r) EfficientViT Cai et al. (2022) | Timm Wightman |
| | (s) MLP-Mixer-B Tolstikhin et al. (2021) | Timm Wightman |
| | (t) ConvMixer-B Trockman & Kolter (2022) | Timm Wightman |
| | (u) Vision Mamba-B Zhu et al. (2024) | https://github.com/hustvl/Vim |
| | (v) MambaVision-B Hatamizadeh & Kautz (2025) | https://github.com/NVlabs/MambaVision |
| Semantic Segmentation (SS) | DeepLabV3+ Chen et al. (2018) | https://github.com/VainF/DeepLabV3Plus-Pytorch |
| | SegFormer Xie et al. (2021) | https://github.com/NVlabs/SegFormer |
| Object Detection (OD) | Faster R-CNN Girshick (2015) | https://github.com/facebookresearch/detectron2 |
| | DETR Carion et al. (2020) | https://github.com/facebookresearch/detr |

(2022b); Aich et al. (2022); Yang et al. (2024a;b), for one epoch using the Adam Kingma & Ba (2015) optimizer $\beta = (0.5, 0.99)$. We set the learning rate $lr = 2e^{-4}$. We also use the mid-level layer feature at $k = 16$ (Maxpooling.3) of VGG-16 surrogate for our baseline Zhang et al. (2022b). We set $\lambda_{cons.} = 0.7$ throughout our experiments for stable generator training at the feature level, and the EMA update parameter $\eta = 0.999$ following Tarvainen & Valpola (2017). We selected $L_{early} = \{1, 2\}$ for matching the first and second intermediate residual blocks within the generator. We compare our attacks against the state-of-the-art baselines that rely on the same ResNet-based generator to craft adversarial examples, i.e. CDA Naseer et al. (2019), LTP Nakka & Salzmann (2021), BIA Zhang et al. (2022b), GAMA Aich et al. (2022), FACL-Attack Yang et al. (2024a), and PDCL-Attack Yang et al. (2024b).

**Dataset statistics.** We describe the statistics of the datasets used for training and evaluation in Table S5. Note that we do not use the training sets from CUB-200-2011 Wah et al. (2011), Stanford Cars Krause et al. (2013), or FGVC Aircraft Maji et al. (2013) for the strict black-box cross-domain.

Table S5: Training and evaluation dataset statistics.

| Dataset | ImageNet-1K Russakovsky et al. (2015) | CUB-200-2011 Wah et al. (2011) | Stanford Cars Krause et al. (2013) | FGVC Aircraft Maji et al. (2013) |
|---|---|---|---|---|
| Train | 1.2 M | 5,994 (Not Used) | 8,144 (Not Used) | 6,667 (Not Used) |
| Val. | 50,000 | 5,794 | 8,041 | 3,333 |
| # Classes | 1,000 | 200 | 196 | 100 |
| Resolution | 224×224 | 448×448 | 448×448 | 448×448 |

**Computational costs.** Since our approach only involves computational overheads during the training of the perturbation generator, there is *no inference time overhead*. During training, we describe in the table on the right that the time for a single forward pass with a batch size of 1 incurs an additional +12.18 (ms) compared to the baseline Zhang et al. (2022b) time of 44.72 (ms), and an additional +28.31 (MB) in memory compared to that for the baseline of 1,404.13 (MB), which are averaged over 1,000 iterations (See Tables S6, S7 ).

These slight increases owe to added computations for the forward (no backward) pass overhead on the teacher generator, $\mathcal{G}_{\theta'}(\cdot)$, in addition to the student, $\mathcal{G}_{\theta}(\cdot)$, and are significantly minor relative to the baseline costs. We note that the training was performed on a single NVIDIA RTX A6000 GPU.

E.4   ADDITIONAL QUANTITATIVE RESULTS

**Results with different surrogate models.** While we performed experiments against the VGG-16 surrogate model for fair comparison with previous works, we also provide, in Tables S9 and

Table S6: Total training compute for the baseline and w/ Ours.

| Method | Train time (hh:mm) | Peak memory (MB) | GPU type | Train batch size |
|--------|--------|--------|--------|--------|
| Baseline | 5:00 | 1,384.62 | NVIDIA RTX A6000 (1×) | 48 |
| w/ Ours | 5:40 | 1,442.23 | | |

Table S7: Average per-train-step compute over 10k steps with a batch size of 1.

| Method | Student fwd (ms) | Teacher fwd (ms) | Backward (ms) | Total (ms) | Backward cost (GFLOPs) | Backward CUDA time (ms) |
|--------|--------|--------|--------|--------|--------|--------|
| Baseline | 7.1 | – | 25.6 | 32.7 | 0.0012 | 19.714 |
| w/ Ours | 6.8 | 3.9 | 28.4 | 39.1 | 0.0044 | 20.192 |

S10, our improvements when trained against other surrogate models such as VGG-19, ResNet-152, and DenseNet-169 as practiced in Zhang et al. (2022b); Yang et al. (2024a;b). Ours added to the baseline trained against all three surrogate models across models and domains, except for cross-model against Dense169, consistently enhances the attack transferability. As with our results against VGG-16 in Tables 1 and 2, our method effectively boosts the attack capacity regardless of the type of surrogate model used for training the generator. We believe the slight increase in cross-model average when using DenseNet-169 as the surrogate model is driven mainly by pronounced gains on a few architectures (e.g. MNASNet and DeiT) while most other models also benefited, albeit to a lesser extent. In the cross-domain evaluation, adversarial examples generated with DenseNet-169 consistently deliver a substantial average performance boost, underscoring its effectiveness across differing data distributions.

In qualitative comparisons across different surrogate models in Fig. S3, our Grad-CAM Selvaraju et al. (2017) visualizations reveal activation patterns that depart significantly from the baseline. Rather than highlighting only the main object region, our method further amplifies those top responses and draws out additional high-sensitivity areas that the baseline misses. When we step through the stages of adversarial noise generation, we see that our approach consistently places perturbations along object edges and contours. By focusing noise on these shared, model-agnostic features instead of scattering it elsewhere, our method not only seeks to align the generated adversarial noise with the most semantically meaningful regions but also achieves stronger transferability.

Table S8: Comparison of cross-task attack strength with ours added to each baseline. Ours further enhances the transferability consistently across semantic segmentation and object detection tasks. Boldface means better results.

| Cross-task | Task | | | | | |
|--------|--------|--------|--------|--------|--------|--------|
| | Semantic Segmentation (mIoU ↓) | | Avg. | Object Detection (mAP50 ↓) | | Avg. |
| | DeepLabV3+ | SegFormer | | Faster R-CNN | DETR | |
| Benign | 76.21 | 71.89 | 74.05 | 61.01 | 62.36 | 61.69 |
| CDA Naseer et al. (2019) | 25.63 | **20.16** | 22.90 | 32.78 | 26.29 | 29.54 |
| w/ Ours | **25.16** | 20.26 | **22.71** | **31.98** | **25.68** | **28.83** |
| LTP Nakka & Salzmann (2021) | 23.71 | 26.97 | 25.34 | 29.39 | 22.41 | 25.90 |
| w/ Ours | **22.27** | **26.68** | **24.48** | **26.85** | **22.18** | **24.52** |
| BIA Zhang et al. (2022b) | 23.89 | 25.60 | 24.75 | 28.43 | 21.01 | 24.72 |
| w/ Ours | **22.05** | **24.75** | **23.40** | **28.34** | **20.70** | **24.52** |
| GAMA Aich et al. (2022) | 24.10 | 27.53 | 25.82 | 28.01 | **20.71** | 24.36 |
| w/ Ours | **23.67** | **25.59** | **24.63** | **27.60** | 20.79 | **24.20** |
| FACL Yang et al. (2024a) | 23.75 | 26.40 | 25.08 | 27.94 | 20.91 | 24.43 |
| w/ Ours | **23.38** | **25.01** | **24.20** | **27.64** | **20.29** | **23.97** |
| PDCL Yang et al. (2024b) | 24.42 | 26.05 | 25.24 | 28.48 | 21.38 | 24.93 |
| w/ Ours | **22.51** | **25.88** | **24.20** | **27.66** | **20.73** | **24.20** |

**Cross-task evaluations.** Cross-task results in Table S8 show only minor differences among existing generative attack methods, with all approaches achieving similarly low attack success rates in this setting. These uniformly modest outcomes highlight the difficulty of transferring adversarial examples crafted on an image classification–oriented surrogate model to tasks with different objectives, since the perturbations fail to align with the target task's feature representations. Nevertheless, integrating our early-block semantic consistency into each baseline yields small but consistent improvements, demonstrating that preserving coarse semantic cues still provides a performance boost even under the strict black-box evaluation.

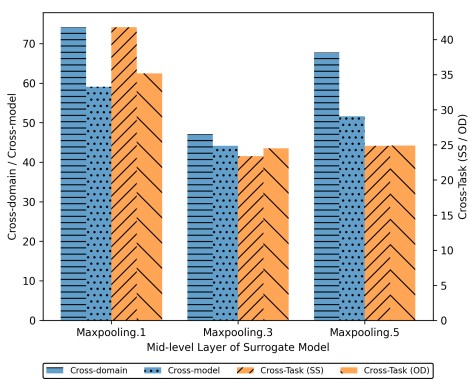

Figure S4: Ablation study on the mid-level layer of the VGG-16 surrogate model.

Table S9: **Quantitative cross-model transferability results**. We report the average improvements ($\Delta$ %p) of our method relative to each baseline, with better results marked in a darker color. For VGG-19, Res152, Dense169 surrogate, (a-d) correspond to {Res50, Res152, Dense121, Dense169}, {VGG16, VGG19, Dense121, Dense169}, and {VGG16,VGG19,Res50,Res152}, respectively, as black-box models.

| Cross-model | | CNN | | | | | | | | | | | Transformer | | | | | | Mixer | | Mamba | | |
|---|---|---|---|---|---|---|---|---|---|---|---|---|---|---|---|---|---|---|---|---|---|---|---|
| **Method** | **Metric** | (a) | (b) | (c) | (d) | (e) | (f) | (g) | (h) | (i) | (j) | (k) | (l) | (m) | (n) | (o) | (p) | (q) | (r) | (s) | (t) | (v) | Avg. |
| Benign | Acc. (%) ↓ | 74.60 | 77.33 | 74.22 | 75.74 | 76.19 | 77.95 | 66.50 | 55.91 | 79.12 | 81.49 | 75.42 | 80.67 | 79.28 | 81.19 | 80.48 | 79.10 | 57.91 | 69.90 | 66.53 | 66.53 | 73.21 | 73.77 |
| **Surrogate model: VGG19** | | | | | | | | | | | | | | | | | | | | | | | |
| BIA w/ Ours | Acc. ($\Delta$%p) ↓ | -1.49 | +0.88 | -1.70 | -3.49 | -4.04 | -1.07 | +1.06 | -2.31 | +0.46 | -4.40 | +0.39 | -0.23 | -0.17 | -1.14 | +0.04 | -0.40 | +0.22 | -0.46 | +0.11 | -0.12 | -0.02 | **-0.85** |
| | ASR ($\Delta$%p) ↑ | +1.93 | -1.12 | +2.17 | +4.43 | +5.18 | +1.34 | -1.43 | +3.98 | -0.65 | +5.18 | -0.44 | +0.35 | +0.16 | +1.19 | -0.06 | +0.48 | -0.35 | +0.65 | -0.15 | +0.17 | +0.11 | **+1.10** |
| | FR ($\Delta$%p) ↑ | +1.72 | -0.89 | +1.97 | +3.95 | +4.68 | +1.24 | -1.19 | +2.89 | -0.55 | +4.77 | -0.37 | +0.51 | +0.07 | +1.26 | -0.14 | +0.27 | -0.56 | +0.69 | -0.27 | +0.13 | +0.10 | **+0.97** |
| | ACR ($\Delta$%p) ↓ | -0.23 | +0.11 | -0.35 | -0.56 | -0.41 | -0.13 | +0.34 | -0.18 | -0.20 | -0.82 | +0.24 | +0.28 | -0.19 | -0.92 | -0.02 | -0.10 | +0.02 | -0.04 | +0.03 | -0.02 | +0.23 | **-0.14** |
| **Method** | **Metric** | (a) | (b) | (c) | (d) | (e) | (f) | (g) | (h) | (i) | (j) | (k) | (l) | (m) | (n) | (o) | (p) | (q) | (r) | (s) | (t) | (v) | Avg. |
| Benign | Acc. (%) ↓ | 70.15 | 70.95 | 74.22 | 75.74 | 76.19 | 77.95 | 66.50 | 55.91 | 79.12 | 81.49 | 75.42 | 80.67 | 79.28 | 81.19 | 80.48 | 79.10 | 57.91 | 69.90 | 66.53 | 66.53 | 73.21 | 73.26 |
| **Surrogate model: Res152** | | | | | | | | | | | | | | | | | | | | | | | |
| BIA w/ Ours | Acc. ($\Delta$%p) ↓ | +0.38 | -0.87 | -2.96 | -0.38 | -5.13 | -7.36 | -5.35 | -0.71 | -3.87 | -6.17 | -3.75 | -0.34 | -0.57 | -0.37 | +0.00 | -0.38 | -0.59 | -3.77 | -2.27 | -0.23 | -0.70 | **-2.16** |
| | ASR ($\Delta$%p) ↑ | -0.52 | +1.22 | +3.76 | +0.39 | +6.36 | +9.09 | +7.37 | +1.32 | +4.75 | +7.16 | +4.70 | +0.41 | +0.83 | +0.51 | +0.08 | -0.97 | +0.47 | +6.42 | +2.02 | +1.66 | +9.36 | **+2.74** |
| | FR ($\Delta$%p) ↑ | -0.49 | +0.95 | +3.23 | +0.27 | +5.70 | +8.14 | +6.29 | +0.97 | +4.37 | -1.23 | +7.24 | -1.22 | -0.45 | -0.94 | +0.30 | -0.86 | +0.57 | +5.38 | +1.60 | +1.42 | +8.78 | **+3.22** |
| | ACR ($\Delta$%p) ↓ | -0.46 | -0.74 | -0.73 | -1.65 | -2.64 | -1.19 | -0.31 | -0.78 | -1.47 | -0.40 | -1.75 | -0.10 | +0.07 | +0.67 | +0.53 | -0.19 | +0.28 | -0.86 | -0.45 | -0.25 | -2.93 | **-0.74** |
| **Method** | **Metric** | (a) | (b) | (c) | (d) | (e) | (f) | (g) | (h) | (i) | (j) | (k) | (l) | (m) | (n) | (o) | (p) | (q) | (r) | (s) | (t) | (v) | Avg. |
| Benign | Acc. (%) ↓ | 70.15 | 70.95 | 74.60 | 77.33 | 76.19 | 77.95 | 66.50 | 55.91 | 79.12 | 81.49 | 75.42 | 80.67 | 79.28 | 81.19 | 80.48 | 79.10 | 57.91 | 69.90 | 66.53 | 66.53 | 73.21 | 73.35 |
| **Surrogate model: Dense169** | | | | | | | | | | | | | | | | | | | | | | | |
| BIA w/ Ours | Acc. ($\Delta$%p) ↓ | -1.63 | -3.00 | -2.25 | -7.65 | -12.99 | -4.66 | -0.13 | -3.01 | -7.68 | +1.19 | -6.71 | +1.00 | +0.36 | +0.91 | -0.10 | +0.74 | -0.16 | -4.74 | -1.50 | -1.19 | -7.63 | **-2.90** |
| | ASR ($\Delta$%p) ↑ | +2.13 | +3.27 | +2.77 | +9.41 | +16.23 | +5.64 | +0.04 | +4.77 | +9.32 | -1.55 | +8.22 | -1.26 | -0.47 | -0.96 | +0.24 | -0.97 | +0.47 | +6.42 | +2.02 | +1.66 | +9.36 | **+3.69** |
| | FR ($\Delta$%p) ↑ | +1.79 | +3.27 | +2.41 | +8.48 | +13.97 | +4.98 | +0.14 | +3.42 | +8.66 | -1.23 | +7.24 | -1.22 | -0.45 | -0.94 | +0.30 | -0.86 | +0.57 | +5.38 | +1.60 | +1.42 | +8.78 | **+3.22** |
| | ACR ($\Delta$%p) ↓ | -0.46 | -0.74 | -0.73 | -1.65 | -2.64 | -1.19 | -0.31 | -0.78 | -1.47 | -0.40 | -1.75 | -0.10 | +0.07 | +0.67 | +0.53 | -0.19 | +0.28 | -0.86 | -0.45 | -0.25 | -2.93 | **-0.74** |

Table S10: **Additional quantitative cross-domain transferability results**. We report the average improvement margins of our method added to each baseline, averaged over three models for each domain using different surrogate models (VGG-19, Res152, Dense169). We report the improvements ($\Delta$ %p) with ours relative to the baseline Zhang et al. (2022b). Better averaged results are marked in **boldface**.

| Cross-domain | CUB-200-2011 | | | | Stanford Cars | | | | FGVC Aircraft | | | | Avg. | SemSeg (SS) | | Avg. | ObjDet (OD) | | Avg. |
|---|---|---|---|---|---|---|---|---|---|---|---|---|---|---|---|---|---|---|---|
| Method | Acc ↓ | ASR ↑ | FR ↑ | ACR ↓ | Acc ↓ | ASR ↑ | FR ↑ | ACR ↓ | Acc ↓ | ASR ↑ | FR ↑ | ACR ↓ | Acc. (%) | DeepLabV3+ | SegFormer | mIoU ↓ | Faster R-CNN | DETR | mAP50 ↓ |
| Benign | 86.91 | – | – | – | 93.56 | – | – | – | 92.07 | – | – | – | 90.85 | 76.21 | 71.89 | 74.05 | 61.01 | 62.36 | 61.69 |
| **Surrogate model: VGG19** | | | | | | | | | | | | | | | | | | | |
| BIA Zhang et al. (2022b) (%) | 52.47 | 41.08 | 45.63 | 9.77 | 71.09 | 25.16 | 27.32 | 17.15 | 52.28 | 44.14 | 46.96 | 11.03 | 58.61 | 28.11 | 25.86 | 26.99 | 28.85 | 21.77 | 25.31 |
| w/ Ours ($\Delta$%p) | **-10.05** | **+11.20** | **+10.39** | **-2.26** | **-11.48** | **+12.04** | **+11.73** | **-2.91** | **-10.87** | **+11.56** | **+11.04** | **-2.79** | **-10.80** | **-2.59** | **-0.69** | **-1.64** | **+0.46** | **+0.12** | **+0.29** |
| **Surrogate model: Res152** | | | | | | | | | | | | | | | | | | | |
| BIA Zhang et al. (2022b) (%) | 49.52 | 44.51 | 48.53 | 9.96 | 50.71 | 46.60 | 48.44 | 12.81 | 40.43 | 56.83 | 59.14 | 9.01 | 46.89 | 32.34 | 31.63 | 31.98 | 33.02 | 26.02 | 29.52 |
| w/ Ours ($\Delta$%p) | **-6.27** | **+7.07** | **+6.62** | **-0.82** | **+0.09** | **-0.15** | **-0.20** | **-0.43** | **-7.17** | **+7.72** | **+7.07** | **-0.87** | **-4.45** | **+2.44** | **+2.55** | **+2.50** | **-0.39** | **+0.99** | **+0.30** |
| **Surrogate model: Dense169** | | | | | | | | | | | | | | | | | | | |
| BIA Zhang et al. (2022b) (%) | 30.01 | 66.46 | 68.97 | 6.73 | 34.08 | 64.13 | 65.34 | 9.29 | 23.23 | 75..24 | 5.74 | 76.44 | 29.11 | 27.70 | 30.31 | 29.01 | 31.53 | 26.89 | 29.21 |
| w/ Ours ($\Delta$%p) | **-3.68** | **+4.04** | **+3.92** | **-1.20** | **-8.54** | **+8.99** | **+8.66** | **-2.32** | **-11.60** | **+12.32** | **+11.62** | **-3.36** | **-7.94** | **+0.08** | **-2.74** | **-1.33** | **-0.37** | **-0.76** | **-0.56** |

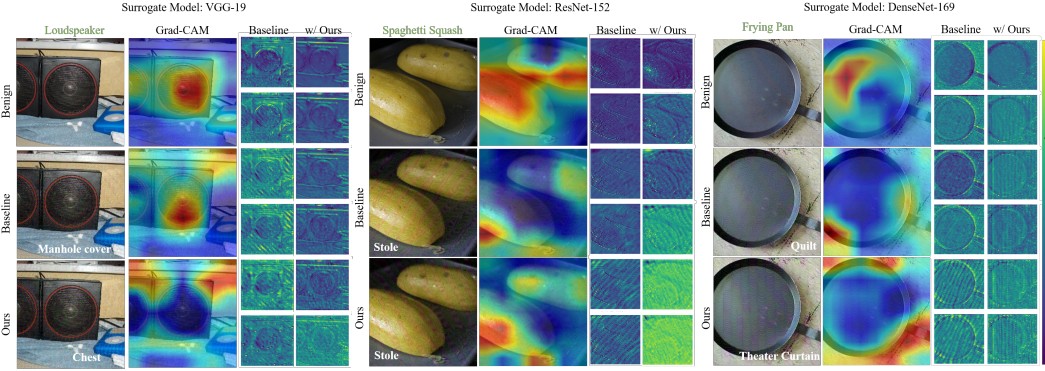

Figure S3: Comparison of the crafted AEs, Grad-CAMs, intermediate feature activation maps from the baseline, and with ours (columns 1–4, respectively). We present the qualitative results from training against other surrogate models: VGG-19 (Left), ResNet-152 (Center), and DenseNet-169 (Right) as commonly compared in existing generative attacks Zhang et al. (2022b); Aich et al. (2022); Yang et al. (2024a;b). The correct label and attacked prediction results are marked in **green** and white, respectively.

**Mid-level layer variations.** To further verify that
our proposed method is compatible with the baseline Zhang et al. (2022b) shown to perform best at the selected mid-level layer of the VGG-16 surrogate model (i.e. Maxpooling.3), we conducted an ablation study of the mid-level layer in Fig. S4. For reference, Maxpooling.1–5 have resolutions

of $112^2$, $56^2$, $28^2$, $14^2$, and $7^2$, respectively. Across domain, model, and two tasks (SS and OD), we observe that our method added to the baseline still maintains the best strength of the attack at the selected mid-level layer, Maxpooling.3, compared to the other early or late layers.

**Ablation study on the hyperparameter $\tau$.** To assess the sensitivity of $\tau$ in Eq. 2, we conducted a sweep of hyperparameter values in Fig. S5. Across the range of values from 0 to 1, we find that our optimal value of $\tau = 0.6$ best balances the strength of the attack across all four cross-settings.

**Slight improvements in perceptual quality.** In Table S11, we compare PSNR, SSIM, and LPIPS scores for each baseline alone versus the same baseline augmented with our early-block semantic consistency mechanism. Across all baselines, adding our method results in a slight PSNR increase while SSIM and LPIPS remain effectively unchanged. These minimal or positive changes confirm that our approach does not introduce any perceptual degradation. Instead, it preserves, and in some cases slightly enhances, the visual fidelity of adversarial examples even as it strengthens their transferability.

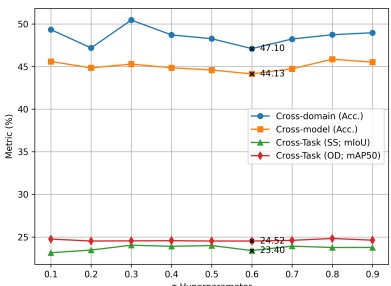

Figure S5: Sensitivity of $\tau$.

Table S11: Comparison of cross-setting performance and image perceptual quality of AEs.

| Method | Cross-setting (Avg.) | | | | Perceptual Quality | | |
|---|---|---|---|---|---|---|---|
| | Domain (Acc.) | Model (Acc.) | Task (SS; mIoU) | Task (OD; mAP50) | PSNR ↑ | SSIM ↑ | LPIPS ↓ |
| CDA | 69.94 | 50.27 | 22.90 | 29.54 | 29.11 | 0.78 | 0.43 |
| w/ Ours | **54.82** | **43.38** | **22.71** | **28.83** | 29.17 (+0.06) | 0.78 (–) | 0.43 (–) |
| LTP | 49.91 | 48.33 | 25.34 | 25.90 | 29.11 | 0.76 | 0.47 |
| w/ Ours | **40.76** | **41.99** | **24.48** | **24.52** | 29.26 (+0.15) | 0.77 (+0.01) | 0.49 (+0.02) |
| BIA | 51.07 | 45.17 | 24.75 | 24.72 | 28.08 | 0.75 | 0.49 |
| w/ Ours | **47.10** | **44.13** | **23.40** | **24.52** | 28.76 (+0.68) | 0.75 (–) | 0.49 (–) |
| GAMA | 48.56 | 44.58 | 25.82 | 24.36 | 28.62 | 0.74 | 0.49 |
| w/ Ours | **46.09** | **43.46** | **24.81** | **24.20** | 28.69 (+0.07) | 0.74 (–) | 0.49 (–) |
| FACL | 44.05 | 42.00 | 25.08 | 24.43 | 28.61 | 0.74 | 0.49 |
| w/ Ours | **41.78** | **40.92** | **24.20** | **23.97** | 28.67 (+0.05) | 0.74 (–) | 0.49 (–) |
| PDCL | 43.91 | 42.84 | 25.24 | 24.93 | 28.68 | 0.74 | 0.48 |
| w/ Ours | **43.06** | **42.77** | **24.20** | **24.20** | 28.70 (+0.02) | 0.74 (–) | 0.49 (–) |

## E.5 ADDITIONAL QUALITATIVE RESULTS

In these additional qualitative results in Fig. S6, we observe two clear patterns in the adversarial masks. First, straight lines trace the edges of objects, reinforcing the primary structural cues. Second, circular ring shapes appear in the background, helping to disperse noise across non-object regions. Grad-CAM visualizations on the right show that our method also drives adversarial activations to much higher levels than those seen in the benign image and boosts areas that exhibited only modest responses under baseline attacks. By combining precise noise along the boundaries with amplified feature activations, our approach anchors noise to the most meaningful contours while strengthening weaker signals, producing stronger and more transferable adversarial examples.

We also provide additional qualitative results on the cross-task (SS and OD) settings. In our additional segmentation examples in Fig. S7, we observe that our method does not just blur or hide parts of the road, but it actually makes the model stop recognizing entire road areas and even small objects like pedestrians or cars. The baseline attack might only erase a few isolated pixels or blend edges, but ours turns whole stretches of road into "ignore," wiping out those predictions in one go. In other words, our method uniformly removes both large surfaces and tiny details, so the segmented map ends up missing key pieces of the scene that the baseline leaves untouched. Similarly, in our additional object detection examples in Fig. S8, our attack causes the model to stop predicting any localized boxes around objects (RoI), completely removing every predicted region of interest, whereas the baseline often leaves boxes in place or only shifts them slightly.

**Feature difference map analysis.** Going further from the investigation of the difference map in the existing work in `block3` only, which is the input to the residual blocks, we expanded the analysis into all the blocks, paying particular attention to the intermediate blocks (`resblocks`). To this end, we compare each baseline to ours and visualized block-wise feature difference in Fig. S9. We observed that our method further boosts the object-centric regions (foreground) towards the early intermediate blocks (`reblocks`), and gradually induces perturbation to be generated towards background, or regions away from the objects directly.

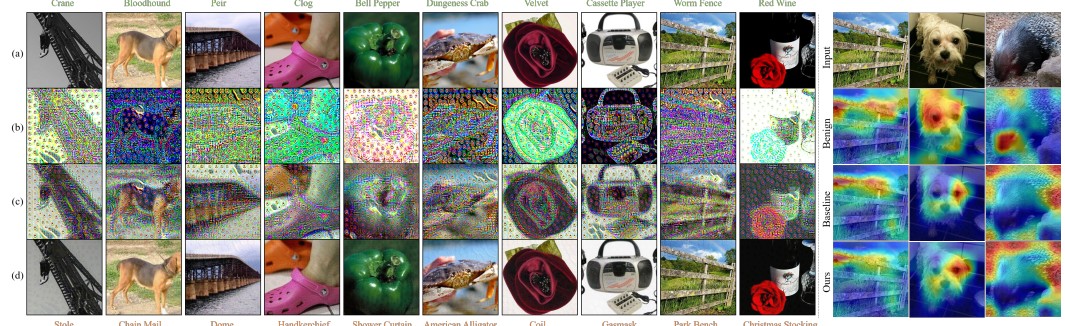

Figure S6: **Additional qualitative results.** Our semantic structure-aware attack successfully guides the generator to focus perturbations particularly on the semantically salient regions, effectively fooling the victim classifier. *Left:* (a) benign input image, (b) generated perturbation (normalized for visual purposes only), (c) unbounded adversarial image, and (d) bounded adversarial image. The label on top (green) and bottom (orange) denote the correct label and prediction after the attack, respectively. *Right:* We highlight that our method induces Grad-CAM Selvaraju et al. (2017) to focus on *drastically different regions* in our adversarial examples compared to both the benign image and the adversarial examples crafted by the baseline Zhang et al. (2022b). Moreover, our approach *noticeably spreads and reduces the high activation regions* observed in the benign and baseline cases, enhancing the transferability of our adversarial perturbations.

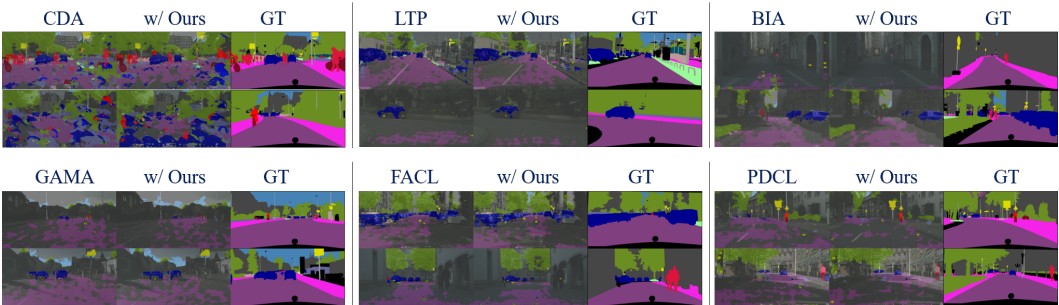

Figure S7: **Additional cross-task (SS) qualitative results.**

**Applicability to state-of-the-art generator-based targeted attacks.** We conduct state-of-the-art targeted black-box attack experiments following M3D Zhao et al. (2023b) and CGNC Fang et al. (2024b) on ImageNet for a single training epoch, using the same VGG-family models as surrogates. Perturbations are bounded by $\ell_\infty$ with $\epsilon = 16/255$. Higher TSR indicates a stronger targeted attack. For M3D, TSR is averaged over target classes $\{24, 99, 245, 344, 471, 555, 661, 701, 802, 919\}$. For CGNC, TSR is averaged over target classes $\{150, 426, 843, 715, 952, 507, 590, 62\}$ in the normal mode.

Table S12: Target success rate (TSR, %) for CLIP-guided CGNC attack (surrogate: VGG19). Higher TSR indicates a stronger targeted attack. TSR is averaged over target classes $\{150, 426, 843, 715, 952, 507, 590, 62\}$ in normal mode. All experiments use ImageNet, VGG-family surrogates, and $\ell_\infty$ perturbations with $\epsilon = 16/255$.

| Method | Victim model | | | | | | | Avg. |
|---|---|---|---|---|---|---|---|---|
| | VGG16 | GoogLeNet | Inc-v3 | Res152 | Dense121 | Inc-v4 | IncRes-v2 | |
| CGNC Fang et al. (2024a) | 14.71 | 2.03 | 2.77 | 2.68 | 8.31 | 2.41 | 0.96 | 4.84 |
| w/ Ours | 47.50 | 7.90 | 10.96 | 12.24 | 31.29 | 13.36 | 3.98 | **18.18** |

**Key observations.**

- **Large average gain.** Our method improves the average TSR from **4.84%** $\rightarrow$ **18.18%**, achieving a **3.7× relative increase**, despite the single-epoch training constraint.
- **Consistent gains across victim architectures.** Every victim model benefits, with especially strong improvements on **DenseNet121 (+22.98%)** and **Inception-v4 (+10.95%)**, indicating that generator-internal semantic consistency remains effective even under CLIP guidance.

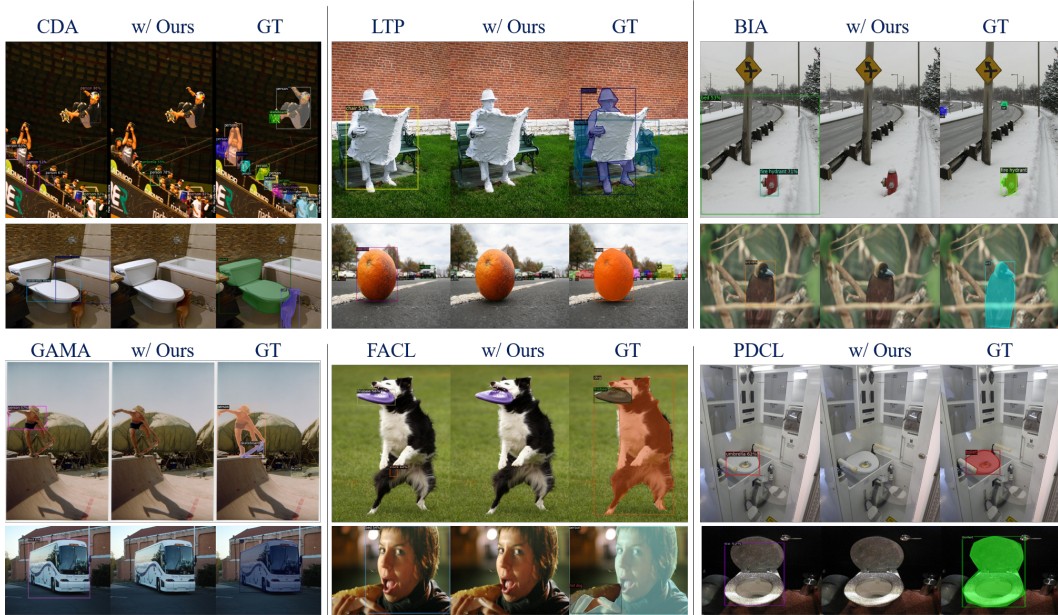

Figure S8: **Additional cross-task (OD) qualitative results.**

Table S13: Performance comparison of M3D and M3D w/Ours across architectures. Final column shows average.

| Model Target | 24 | 99 | 245 | 344 | 471 | 555 | 661 | 701 | 802 | 919 | Avg. |
|---|---|---|---|---|---|---|---|---|---|---|---|
| **DenseNet121** | | | | | | | | | | | |
| M3D Zhao et al. (2023a) | 71.24 | 77.12 | 73.35 | 85.37 | 71.75 | 73.79 | 60.63 | 78.70 | 69.66 | 17.34 | 67.90 |
| w/ Ours | 76.53 | 77.11 | 82.64 | 87.74 | 82.81 | 78.72 | 74.12 | 78.29 | 79.43 | 47.20 | **76.46** |
| **ResNet50** | | | | | | | | | | | |
| M3D Zhao et al. (2023a) | 68.54 | 75.48 | 77.67 | 79.43 | 77.64 | 80.05 | 54.48 | 89.04 | 55.85 | 9.51 | 66.77 |
| w/ Ours | 71.02 | 73.60 | 80.69 | 88.04 | 87.64 | 83.88 | 68.97 | 84.88 | 69.60 | 45.36 | **75.37** |
| **ResNet152** | | | | | | | | | | | |
| M3D Zhao et al. (2023a) | 50.73 | 62.76 | 63.15 | 69.53 | 60.96 | 57.82 | 37.28 | 73.87 | 37.82 | 11.29 | 52.52 |
| w/ Ours | 60.35 | 58.80 | 68.61 | 79.73 | 73.64 | 65.47 | 56.12 | 70.29 | 52.77 | 39.63 | **62.54** |
| **WRN-50-2** | | | | | | | | | | | |
| M3D Zhao et al. (2023a) | 64.55 | 68.95 | 73.51 | 69.53 | 73.04 | 66.53 | 46.85 | 84.39 | 45.17 | 13.15 | 60.57 |
| w/ Ours | 72.41 | 69.22 | 77.94 | 79.61 | 82.97 | 69.83 | 66.16 | 76.32 | 58.20 | 41.01 | **69.37** |

**Key observations.**

- **Consistent gains on a strong targeted attack.** The overall average TSR across all four victims increases from **61.94% → 70.94%**, with absolute gains of **+8.56%p (DenseNet121)**, **+8.6%p (ResNet50)**, **+10.02%p (ResNet152)**, and **+8.8%p (WRN-50-2)**.

- **Across-target robustness.** Improvements are observed across a randomized set of 10 target classes, not just "easy" ones. For example, on **ResNet50**, class 919 improves from 9.51% → 45.36%, showing that generator-centric regularization effectively steers features toward diverse target semantics even under tight training.

**Attack robustness against purification defense.** Further looking into how robust our attack performs against purification methods such as NRP Naseer et al. (2020), we report the improvements with Ours in Table S17. On most of the baselines, our method addition maintains lower Accuracy, ASR and FR scores than the baseline alone, while on the recent advanced methods (e.g. FACL Yang et al. (2024a) and PDCL Yang et al. (2024b)), our method very slightly maintains similar scores with those of the baseline alone on the three metrics. On the other hand, on the more challenging ACR metric, we observe that our method slightly improves from the baseline, entailing that our generative feature-level tuning further reduces the number of inadvertently corrected samples.

Running DiffPure on ImageNet is computationally intensive and the cost grows with the number of samples, so within the rebuttal time frame, we conducted preliminary experiments on a random subset of 1k validation images. We plan to extend to the full 50k val set given sufficient time.

Our framework is designed to improve adversarial transferability on undefended models, following standard protocols in CDA, LTP, BIA, GAMA, FACL, and PDCL, rather than to construct an adaptive attack specifically tailored to circumvent purification defenses.

Under DiffPure with $\epsilon = 10/255$ and $t = 150$, incorporating our method yields robust accuracies that remain comparable to the baselines on both victims. Fluctuations are small in magnitude and are plausibly explained by the stochastic diffusion process and the limited 1k sample size, rather than by a systematic loss of robustness.

Table S14: Preliminary robustness evaluations against DiffPure ($\epsilon = 10/255$, $t = 150$) on 1k samples (Cls. Acc., %).

| Victim | CDA Naseer et al. (2019) | w/ Ours | LTP Nakka & Salzmann (2021) | w/ Ours | BIA Zhang et al. (2022b) | w/ Ours |
|---|---|---|---|---|---|---|
| **Res152** | 67.7 | 66.3 | 66.1 | 68.0 | 66.1 | 66.8 |
| **Dense121** | 62.0 | 60.5 | 62.9 | 64.0 | 62.1 | 61.4 |

| Victim | GAMA Aich et al. (2022) | w/ Ours | FACL Yang et al. (2024a) | w/ Ours | PDCL Yang et al. (2024b) | w/ Ours |
|---|---|---|---|---|---|---|
| **Res152** | 65.8 | 66.6 | 65.4 | 66.2 | 65.3 | 65.2 |
| **Dense121** | 62.0 | 61.6 | 61.8 | 62.1 | 61.5 | 61.4 |

**Key observations:**

- On ResNet152, accuracies with Ours stay close to the baselines, with modest increases (e.g., LTP, BIA, GAMA, FACL) and modest decreases (e.g., CDA), all within a narrow band.
- On DenseNet121, deviations are similarly small, and several pairs (e.g., LTP, FACL) show slight improvements.
- Overall, these preliminary results indicate that our method remains largely compatible with strong diffusion based purification. The semantic consistency enforced during generation improves transfer on standard models without causing a significant loss of robustness under DiffPure compared to the original baselines.

**Additional results on baselines and w/ Ours against against input processing methods.** In addition to the results in Table 4 against robustly trained models, we further report the results on the other baselines that are left out due to page limitations in Table S15.

**Additional results on input processing defense.** Beyond the main experiments, we further evaluate our method under standard input-processing defenses. Rotation (deg) applies small random rotations sampled from a bounded angle range to preserve semantic content while perturbing pixel-level alignment Guo et al. (2017b); Xie et al. (2018).

*Smoothing* uses standard spatial smoothing filters (Gaussian, median, and mean) to attenuate small high-frequency adversarial perturbations while largely preserving coarse structure, at the cost of some local blurring Guo et al. (2017b); Xu et al. (2018).

*Total variation minimization* (TVM) performs TV-based denoising by approximately minimizing a reconstruction loss with a total-variation regularizer, yielding a piecewise-smooth reconstruction that preserves major edges while suppressing small oscillatory perturbations Guo et al. (2017b); Rudin et al. (1992).

*Pixel deflection* (PD) randomly selects a subset of pixels and replaces each with the value of a randomly chosen neighbor within a local window to stochastically disrupt finely tuned adversarial patterns without destroying global semantics Prakash et al. (2018).

Taken together, JPEG-style compression (JPEG), bit-depth reduction (BDR), random resizing–padding (R&P), and smoothing, as well as the transformations reported in the additional table, correspond to the standard family of input-processing defenses widely used in prior work on transferable attacks, including our baselines and recent studies such as TransferAttackEval Zhao et al. (2025) and CGNC Fang et al. (2024b). TVM and pixel deflection can be regarded as stronger, yet

Table S15: Defense evaluation comparisons against other baselines.

| Method | Metric | Adv.IncV3 | Adv.ViT | Adv.ConvNeXt | JPEG | BDR | R&P | Avg. |
|---|---|---|---|---|---|---|---|---|
| Benign | Acc. (%) ↓ | 76.33 | 48.82 | 58.44 | 74.68 | 74.68 | 76.58 | 68.26 |
| **BIA** | Acc. (%) ↓ | 68.54 | 45.64 | 53.88 | 63.49 | 47.82 | 44.78 | 54.03 |
| | ASR (%) ↑ | 14.95 | 11.72 | 10.26 | 20.24 | 40.76 | 44.59 | 23.75 |
| | FR (%) ↑ | 24.02 | 25.48 | 19.40 | 28.09 | 48.06 | 51.60 | 32.78 |
| | ACR (%) ↓ | 15.30 | 4.96 | 3.46 | 11.45 | 11.30 | 10.56 | 9.51 |
| **BIA w/ Ours** | Acc. (%) ↓ | 67.92 | 45.33 | 53.62 | 60.83 | 44.07 | 39.01 | **51.80** |
| | ASR (%) ↑ | 15.75 | 11.95 | 10.65 | 23.74 | 45.37 | 51.63 | **26.52** |
| | FR (%) ↑ | 24.83 | 25.31 | 19.60 | 31.61 | 52.22 | 57.86 | **35.28** |
| | ACR (%) ↓ | 15.23 | 4.57 | 3.38 | 11.48 | 10.29 | 9.08 | **9.01** |
| **GAMA** | Acc. (%) ↓ | 66.71 | 45.74 | 53.76 | 59.27 | 41.08 | 37.60 | 50.69 |
| | ASR (%) ↑ | 17.27 | 11.65 | 10.50 | 25.66 | 49.01 | 53.34 | 27.91 |
| | FR (%) ↑ | 26.37 | 25.70 | 19.68 | 33.51 | 55.42 | 59.42 | 36.68 |
| | ACR (%) ↓ | 15.06 | 5.09 | 3.50 | 11.07 | 9.38 | 8.62 | 8.79 |
| **GAMA w/ Ours** | Acc. (%) ↓ | 66.37 | 45.50 | 53.81 | 58.99 | 39.07 | 36.00 | **49.96** |
| | ASR (%) ↑ | 17.76 | 11.78 | 10.45 | 26.07 | 51.49 | 55.36 | **28.82** |
| | FR (%) ↑ | 26.78 | 25.34 | 19.61 | 33.86 | 57.63 | 61.18 | **37.40** |
| | ACR (%) ↓ | 15.19 | 4.74 | 3.56 | 11.19 | 8.88 | 8.36 | **8.65** |
| **FACL** | Acc. (%) ↓ | 65.68 | 45.17 | 53.12 | 47.25 | 38.36 | 33.31 | 47.15 |
| | ASR (%) ↑ | 18.68 | 12.44 | 11.70 | 41.08 | 52.43 | 58.78 | 32.52 |
| | FR (%) ↑ | 27.75 | 26.24 | 21.19 | 47.99 | 58.39 | 64.16 | 40.95 |
| | ACR (%) ↓ | 15.22 | 4.73 | 3.66 | 9.90 | 8.91 | 8.03 | 8.41 |
| **FACL w/ Ours** | Acc. (%) ↓ | 65.49 | 44.89 | 53.14 | 47.93 | 33.24 | 28.64 | **45.56** |
| | ASR (%) ↑ | 18.88 | 12.74 | 11.58 | 40.25 | 58.85 | 64.60 | **34.48** |
| | FR (%) ↑ | 27.90 | 26.12 | 20.88 | 47.28 | 64.03 | 69.19 | **42.57** |
| | ACR (%) ↓ | 15.07 | 4.46 | 3.54 | 10.10 | 7.94 | 7.00 | **8.01** |
| **PDCL** | Acc. (%) ↓ | 67.57 | 45.24 | 53.61 | 58.10 | 39.84 | 37.01 | 50.23 |
| | ASR (%) ↑ | 16.20 | 12.10 | 10.74 | 27.15 | 50.56 | 54.09 | 28.47 |
| | FR (%) ↑ | 25.29 | 25.43 | 19.84 | 34.92 | 56.77 | 59.98 | 37.04 |
| | ACR (%) ↓ | 15.24 | 4.55 | 3.47 | 10.91 | 9.12 | 8.54 | **8.64** |
| **PDCL w/ Ours** | Acc. (%) ↓ | 67.53 | 45.13 | 53.48 | 57.63 | 39.53 | 35.67 | **49.83** |
| | ASR (%) ↑ | 16.24 | 12.38 | 10.88 | 27.82 | 51.06 | 55.78 | **29.03** |
| | FR (%) ↑ | 25.27 | 25.69 | 19.92 | 35.54 | 57.29 | 61.50 | **37.54** |
| | ACR (%) ↓ | 15.16 | 4.59 | 3.36 | 11.09 | 9.42 | 8.31 | 8.66 |

conceptually similar, input transformations in this family. Evaluating under this common protocol makes our results directly comparable to existing attacks, and across these defenses we observe that adding our generator-side regularizer consistently lowers accuracy and increases FR/ASR relative to each underlying baseline, indicating that our method enhances attack effectiveness against the standard input pre-processing defenses.

**Attack robustness on zero-shot image classification.** We also evaluated our method on the zero-shot image classification task with the well-known CLIP Radford et al. (2021) vision-language model in Table S18. Here, we observe that, except for BIA Zhang et al. (2022b) and FACL Yang et al. (2024a), we observe boosted attacked accuracy when we add our method to the baselines Naseer et al. (2019); Nakka & Salzmann (2021); Aich et al. (2022); Yang et al. (2024b). We conjecture that the slight attack strength degradation owes to the baseline method that has already been well-fitted to generate adversarial examples effective for the zero-shot setting based on the relatively lower accuracy scores than the rest. We posit that the respective well-trained generator is already adept enough that our method may interfere with the learned generator weights negatively, and there may exist a maximum capacity at which AEs from generative model-based attacks can attack victim models.

Table S16: Additional input processing defenses.

| | | Avg. | Random Rotation (deg) | | | | Smoothing (Kernel) | | | Total Var. Min. | Random Pixel |
|---|---|---|---|---|---|---|---|---|---|---|---|
| | | | 30 | 50 | 70 | 90 | Gaussian | Median | Mean | TVM | PD |
| **BIA** | Acc. (%) ↓ | 37.08 | 33.59 | 29.20 | 26.27 | 24.92 | 58.75 | 54.27 | 59.06 | 41.17 | 41.62 |
| | ASR (%) ↑ | 53.95 | 58.38 | 63.86 | 67.52 | 69.26 | 26.66 | 32.44 | 26.15 | 49.05 | 48.48 |
| | FR (%) ↑ | 59.56 | 63.89 | 68.66 | 71.81 | 73.34 | 34.81 | 40.36 | 34.46 | 55.38 | 54.98 |
| | ACR (%) ↓ | 8.36 | 7.91 | 6.97 | 6.40 | 6.31 | 11.94 | 11.77 | 11.76 | 9.87 | 9.96 |
| **BIA w/ Ours** | Acc. (%) ↓ | **34.05** | 28.61 | 28.43 | 22.27 | 21.18 | 55.55 | 49.61 | 57.20 | 37.42 | 38.15 |
| | ASR (%) ↑ | **57.75** | 64.52 | 64.79 | 72.44 | 73.86 | 30.84 | 38.44 | 28.63 | 53.61 | 52.77 |
| | FR (%) ↑ | **63.01** | 69.32 | 69.50 | 76.14 | 77.43 | 38.88 | 45.82 | 36.70 | 59.60 | 58.86 |
| | ACR (%) ↓ | **7.80** | 6.62 | 6.76 | 5.36 | 5.34 | 12.01 | 11.37 | 11.88 | 8.76 | 9.09 |
| **GAMA** | Acc. (%) ↓ | 31.66 | 26.10 | 26.25 | 20.18 | 19.10 | 52.47 | 45.53 | 54.05 | 35.08 | 35.78 |
| | ASR (%) ↑ | 60.67 | 67.59 | 67.48 | 75.03 | 76.46 | 34.61 | 43.45 | 32.55 | 56.41 | 55.52 |
| | FR (%) ↑ | 65.60 | 71.93 | 71.88 | 78.35 | 79.57 | 42.41 | 50.40 | 40.47 | 62.00 | 61.17 |
| | ACR (%) ↓ | 7.12 | 5.90 | 6.19 | 4.88 | 4.88 | 11.15 | 10.32 | 11.18 | 7.88 | 7.97 |
| **GAMA w/ Ours** | Acc. (%) ↓ | **30.30** | 25.31 | 21.75 | 19.41 | 18.41 | 52.09 | 43.89 | 53.91 | 32.60 | 33.61 |
| | ASR (%) ↑ | **62.37** | 68.63 | 73.06 | 76.01 | 77.29 | 35.09 | 45.46 | 32.76 | 59.55 | 58.28 |
| | FR (%) ↑ | **67.08** | 72.92 | 76.65 | 79.26 | 80.33 | 42.71 | 52.09 | 40.62 | 64.77 | 63.66 |
| | ACR (%) ↓ | **6.86** | 5.93 | 5.16 | 4.77 | 4.65 | 11.09 | 9.80 | 11.25 | 7.48 | 7.68 |
| **FACL** | Acc. (%) ↓ | 27.12 | 22.00 | 18.64 | 16.68 | 15.84 | 45.91 | 36.29 | 48.38 | 32.40 | 33.11 |
| | ASR (%) ↑ | 66.41 | 72.75 | 77.08 | 79.49 | 80.59 | 42.91 | 55.01 | 39.73 | 59.88 | 59.00 |
| | FR (%) ↑ | 70.66 | 76.42 | 80.21 | 82.20 | 83.13 | 49.88 | 60.58 | 46.97 | 64.60 | 64.31 |
| | ACR (%) ↓ | 6.44 | 5.22 | 4.95 | 4.43 | 4.42 | 10.14 | 8.46 | 10.34 | 7.74 | 7.89 |
| **FACL w/ Ours** | Acc. (%) ↓ | **25.30** | 19.68 | 16.91 | 15.20 | 14.61 | 43.08 | 33.43 | 46.18 | 30.56 | 31.32 |
| | ASR (%) ↑ | **68.69** | 75.78 | 79.19 | 81.35 | 82.13 | 46.35 | 58.54 | 42.50 | 62.15 | 61.28 |
| | FR (%) ↑ | **72.68** | 79.02 | 81.99 | 83.83 | 84.49 | 52.95 | 63.78 | 49.47 | 67.02 | 66.33 |
| | ACR (%) ↓ | **6.08** | 5.15 | 4.47 | 4.15 | 4.16 | 9.29 | 7.77 | 9.97 | 7.25 | 7.66 |
| **PDCL** | Acc. (%) ↓ | 31.34 | 26.04 | 22.50 | 21.84 | 19.02 | 52.68 | 45.87 | 56.05 | 32.97 | 34.40 |
| | ASR (%) ↑ | 60.87 | 67.76 | 72.16 | 73.00 | 76.50 | 31.79 | 43.04 | 29.95 | 59.06 | 57.78 |
| | FR (%) ↑ | 65.75 | 72.10 | 75.91 | 76.60 | 79.65 | 39.75 | 50.02 | 38.04 | 64.29 | 63.17 |
| | ACR (%) ↓ | 7.09 | 6.20 | 5.39 | 5.35 | 4.67 | 11.42 | 10.41 | 11.30 | 7.46 | 7.87 |
| **PDCL w/ Ours** | Acc. (%) ↓ | **30.80** | 25.20 | 21.84 | 19.58 | 18.63 | 53.77 | 45.34 | 55.29 | 32.64 | 33.71 |
| | ASR (%) ↑ | **61.76** | 68.80 | 73.00 | 75.85 | 77.04 | 32.87 | 43.68 | 30.98 | 59.54 | 58.26 |
| | FR (%) ↑ | **66.54** | 73.03 | 76.60 | 79.06 | 80.16 | 40.66 | 50.63 | 38.92 | 64.34 | 63.74 |
| | ACR (%) ↓ | **7.02** | 6.00 | 5.35 | 4.97 | 4.80 | 11.03 | 10.22 | 11.39 | 7.63 | 8.06 |

**Random trials for Baseline Zhang et al. (2022b) with Ours.** In Table S20, we further show that our method exhibits stable training results (mean±std.dev.) as shown from multiple random seed trials evaluated on all four cross-settings.

**Attack robustness against real-world systems.** In order to be on the path to the paradigm of commercial models that are currently being actively deployed, we tested our attack strategy against multi-modal large language models on zero-shot image classification and image captioning in Tables S18 and S19, respectively. For the zero-shot image classification task with the well-known CLIP Radford et al. (2021) vision language model, we observe that, except for BIA Zhang et al. (2022b) and FACL Yang et al. (2024a), we observe improved attacked accuracy when we add our method to the baselines Naseer et al. (2019); Nakka & Salzmann (2021); Aich et al. (2022); Yang et al. (2024b). We conjecture that the slight attack strength degradation owes to the baseline method that has already been well-fitted to generate adversarial examples effective for the zero-shot setting based on the relatively lower accuracy scores than the rest. We posit that the respective well-trained generator is already adept enough that our method may interfere with the learned generator weights negatively, and there may exist a maximum capacity at which AEs from generative model-based attacks can attack victim models. Against LLaVA 1.5-7B Liu et al. (2023a), our attack on the image captioning task, compared with the baseline, shows competitive attack potential. Although our attack is crafted using an image-only CNN surrogate model and its impact on similar architectures is most notable, we observe that attacking the image branch of the concurrent multi-modal models can also be a viable option for adversarial attacks. We defer this to future exploration along with the text-side attacks. Our method thus well demonstrates the potential to further impair the recognition capabilities of large, deployed multimodal models, including vision-language systems such as LLaVA and GPT-4o.

**Spectral energy comparison by band.** We emphasize that the spectral ratios in Table 6 and the band-wise t-SNE Maaten & Hinton (2008) visualizations of generator intermediate features (after `resblock3`) in Fig. S10 are obtained in two different but complementary ways. For Table 6, we work purely in the frequency domain: for each generator output, we compute the 2D FFT, partition

Table S17: Adversarial transferability results with our method against the purification method tested on the Inc-V3 victim model, and random seed testing. Better results in **boldface**.

| Method | Metric | Purification | | Avg. |
|---|---|---|---|---|
| | | NRP | NRP-ResNet | |
| Benign | Acc. % | | 76.19 | |
| CDA | Acc. (%)↓ | 71.14 | 67.04 | 69.09 |
| | ASR (%)↑ | 10.29 | 15.94 | 13.11 |
| | FR (%)↑ | 17.45 | 24.00 | 20.73 |
| | ACR (%)↓ | 11.73 | 12.53 | **12.13** |
| w/ Ours | Acc. (%)↓ | 70.90 | 66.16 | **68.53** |
| | ASR (%)↑ | 10.73 | 17.07 | **13.90** |
| | FR (%)↑ | 18.15 | 25.25 | **21.70** |
| | ACR (%)↓ | 12.11 | 12.51 | 12.31 |
| LTP | Acc. (%)↓ | 72.19 | 67.65 | 69.92 |
| | ASR (%)↑ | 8.82 | 15.22 | 12.02 |
| | FR (%)↑ | 15.59 | 23.15 | 19.37 |
| | ACR (%)↓ | 11.42 | 12.83 | **12.13** |
| w/ Ours | Acc. (%)↓ | 71.78 | 65.51 | **68.65** |
| | ASR (%)↑ | 9.45 | 17.97 | **13.71** |
| | FR (%)↑ | 16.34 | 25.98 | **21.16** |
| | ACR (%)↓ | 11.69 | 12.63 | 12.16 |
| BIA | Acc. (%)↓ | 73.84 | 71.93 | 72.89 |
| | ASR (%)↑ | 6.37 | 9.20 | 7.79 |
| | FR (%)↑ | 12.46 | 16.35 | 14.41 |
| | ACR (%)↓ | 19.51 | 11.53 | **11.02** |
| w/ Ours | Acc. (%)↓ | 73.85 | 71.40 | **72.63** |
| | ASR (%)↑ | 6.36 | 10.00 | **8.18** |
| | FR (%)↑ | 12.63 | 17.31 | **14.97** |
| | ACR (%)↓ | 10.52 | 11.89 | 11.21 |
| GAMA | Acc. (%)↓ | 74.42 | 72.30 | 73.36 |
| | ASR (%)↑ | 5.35 | 8.53 | 6.94 |
| | FR (%)↑ | 10.96 | 15.35 | 13.16 |
| | ACR (%)↓ | 9.67 | 10.96 | **10.32** |
| w/ Ours | Acc. (%)↓ | 74.31 | 71.78 | **73.05** |
| | ASR (%)↑ | 5.65 | 9.32 | **7.49** |
| | FR (%)↑ | 11.60 | 16.32 | **13.96** |
| | ACR (%)↓ | 10.17 | 11.29 | 10.73 |
| FACL | Acc. (%)↓ | 74.23 | 71.95 | 73.09 |
| | ASR (%)↑ | 5.68 | 9.07 | 7.38 |
| | FR (%)↑ | 11.57 | 15.94 | 13.76 |
| | ACR (%)↓ | 9.94 | 11.22 | 10.58 |
| w/ Ours | Acc. (%)↓ | 74.21 | 72.00 | 73.11 |
| | ASR (%)↑ | 5.69 | 8.98 | 7.34 |
| | FR (%)↑ | 11.42 | 15.81 | 13.62 |
| | ACR (%)↓ | 9.90 | 11.11 | **10.51** |
| PDCL | Acc. (%)↓ | 74.04 | 71.55 | **72.80** |
| | ASR (%)↑ | 6.16 | 9.63 | **7.90** |
| | FR (%)↑ | 12.26 | 16.82 | **14.54** |
| | ACR (%)↓ | 10.66 | 11.34 | 11.00 |
| w/ Ours | Acc. (%)↓ | 74.17 | 71.76 | 72.97 |
| | ASR (%)↑ | 5.92 | 9.32 | 7.62 |
| | FR (%)↑ | 11.91 | 16.43 | 14.17 |
| | ACR (%)↓ | 10.45 | 11.20 | **10.83** |

Table S18: Real-world system evaluations (400 random images).

| Task | Zero-Shot Cls. (Acc. %) | | |
|---|---|---|---|
| Data subset | ImageNet-R | ImageNet-1K | |
| Model | GPT-4o-mini | GPT-4o-mini | GPT-4.1 |
| Benign | 27.25 | 34.25 | 17.50 |
| Baseline | 6.75 | 5.00 | 3.00 |
| w/ Ours | 4.75 | 1.25 | 1.00 |

Table S19: Multi-modal large language model evaluations with a prompt 'Provide a short caption for this image'.

| Task | Image Captioning (MS COCO) on LLaVA 1.5-7B | | | | |
|---|---|---|---|---|---|
| Metric | BLEU-4 | METEOR | ROUGE-L | CIDEr | SPICE |
| Benign | 34.9 | 29.1 | 58.0 | 122.7 | 22.9 |
| BIA | 35.0 | 28.6 | 57.4 | 124.9 | 22.7 |
| w/ Ours | 34.5 | 28.4 | 57.0 | 123.1 | 22.2 |

Table S20: Random trials for Ours added to the baseline Zhang et al. (2022b).

| Trial | Cross- | Accuracy↓ | ASR↑ | FR↑ | ACR↓ |
|---|---|---|---|---|---|
| 1 | Domain Model | 47.99 | 48.14 | 50.81 | 10.24 |
| | | 46.12 | 42.67 | 48.69 | 8.34 |
| | Task (OD) | 24.51 (mAP50) | | | |
| | Task (SS) | 23.02 (mIoU) | | | |
| 2 | Domain Model | 47.95 | 48.19 | 50.86 | 10.31 |
| | | 46.26 | 42.53 | 48.49 | 8.32 |
| | Task (OD) | 24.52 (mAP50) | | | |
| | Task (SS) | 24.33 (mIoU) | | | |
| 3 | Domain Model | 47.10 | 49.02 | 51.66 | 9.66 |
| | | 45.86 | 43.04 | 49.00 | 8.28 |
| | Task (OD) | 24.52 (mAP50) | | | |
| | Task (SS) | 23.20 (mIoU) | | | |
| Avg. | Domain Model | **47.58** ± 0.41 | **48.45** ± 0.40 | **51.11** ± 0.39 | **10.07** ± 0.29 |
| | | **46.08** ± 0.17 | **42.75** ± 0.22 | **48.73** ± 0.21 | **8.31** ± 0.03 |
| | Task (OD) | **24.52** ± 0.01 (mAP50) | | | |
| | Task (SS) | **23.52** ± 0.58 (mIoU) | | | |

the spectrum into a low band ($\rho < 0.2$) and its complement ($\rho \geq 0.2$) based on the normalized radial distance $\rho$ from the DC component, and then integrate the power over each band. The reported numbers are therefore scalar ratios of Fourier-domain energy between low and the remaining higher frequencies. In contrast, for Fig. S10 we first apply radial masks to the FFT of the intermediate feature maps after `resblock3`, isolate either the low band ($\rho < 0.2$) or an extreme high band ($\rho \geq 0.8$), and then perform an inverse FFT. This yields band-limited reconstructions back in the image space of the features, which we average over the validation set. The visualization then shows the spatial low- and extreme-high-frequency content, rather than showing raw power spectrum of each band.

The qualitative effects in Fig. S10 are consistent with the quantitative trends in Table 6. Across baselines, SCGA increases the low-band energy ratio and decreases the high-band ratio in Table 6. In the visualization, this appears as low-band reconstructions with brighter and more compact regions for SCGA, compared to the more diffuse patterns of the baselines. These bright areas indicate that a larger fraction of feature variance is carried by smoothly varying, semantically aligned components in the low band, which explains the higher low-band energy ratios. In contrast, the high-band reconstructions with SCGA are less peaky and more spatially dispersed, with fewer intensely bright spots. This visual pattern matches the reduced and less concentrated high-frequency energy in Table 6 and suggests that sharp surrogate-specific artifacts are attenuated and spread out. The PDCL case is a

mild exception. Table 6 still reports a net shift of energy toward the low band, but in Fig. S10 the remaining extreme-high-frequency content becomes slightly more localized, which is consistent with the relatively modest gains of PDCL+SCGA in the transferability results.

 **Side-by-side visualization with the baseline.** To directly compare our method against the baseline qualitatively, we visualize them in Fig. S11. Here, we observe that the predictions after each attack are highly similar, yet the object-aligned patterns in the perturbations are vastly different. Our perturbation demonstrates more vivid perturbations concentrate on the foreground regions without blurs in the noise pattern, suggesting that our intended object-aligned perturbations improved compared to those of the baseline.

### E.6    LIMITATIONS AND BROADER SOCIETAL IMPACTS

Our method exposes vulnerabilities in generative attack pipelines, yet its transferability gains remain bounded by the underlying generator architecture. By revealing these constraints in publicly available generative models, we contribute to exposing safety vulnerabilities of neural networks. The demonstrated transferability of generator internal semantic-aware perturbations underscores the need for adversarial robustness and motivates integrating safety measures, such as early-block regularization or semantic-consistency checks, into future network designs. Moreover, our approach targeting the adversarial perturbation process directly differs in principle from those that explicitly target benign-adversarial divergence in the surrogate model level. Therefore, our method stands as a compatible method to enhance those methods further, not to be assessed on the same grounds.

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

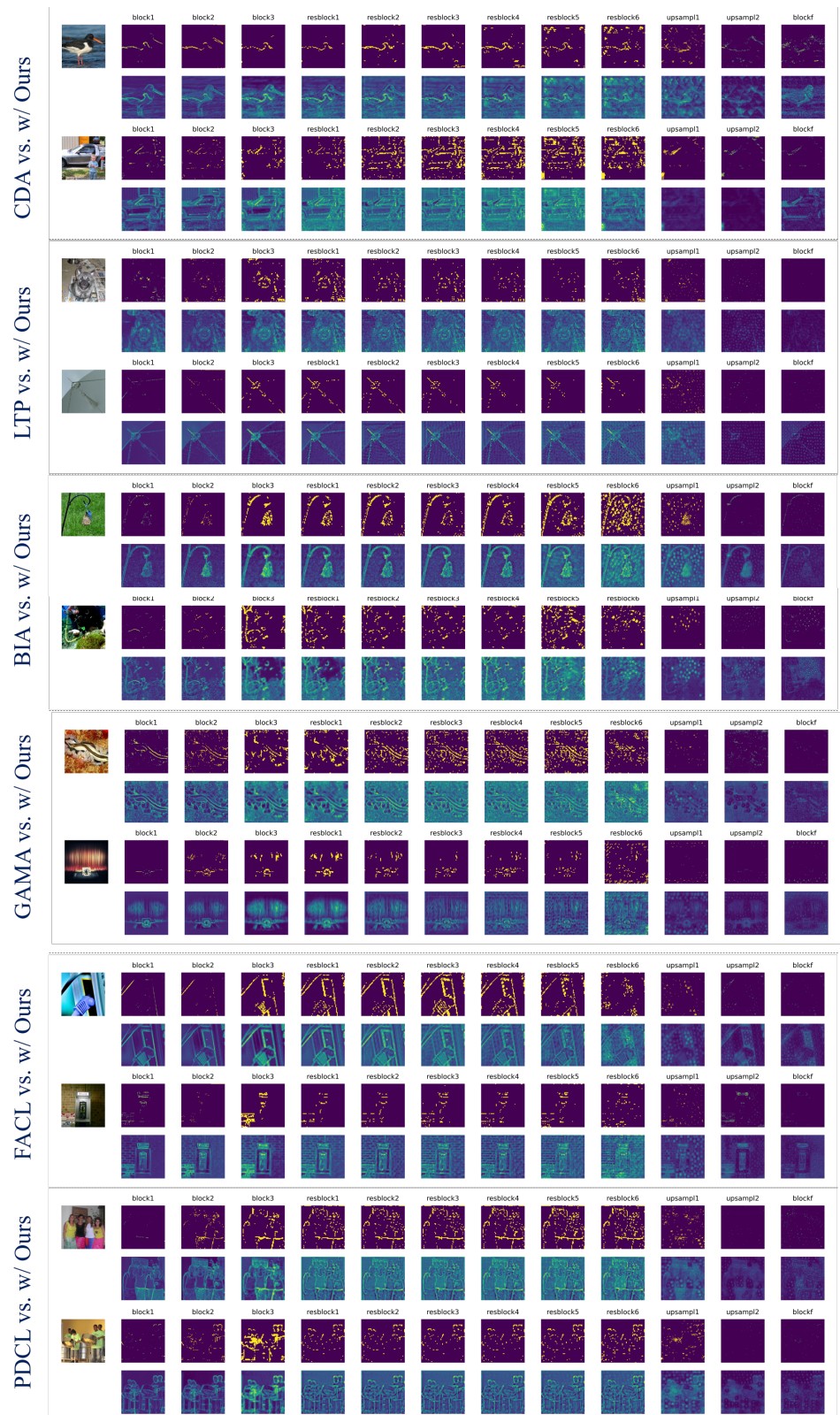

Figure S9: **Feature difference map comparisons.** Our method noticeably adds noise to object-salient regions in the generator intermediate features, as visible by the distinctive difference from each baseline. For each input on the leftmost column, we visualize the output feature map of each block in the generator in each column (*left→right*): thresholded ($\tau = 0.6$) binary mask (*row 1*) after min-max normalization and feature activation difference maps (*row 2*). In the in the `resblocks` in particular, our method further guides perturbations around the object semantic structure, enhancing transferable noise generation.

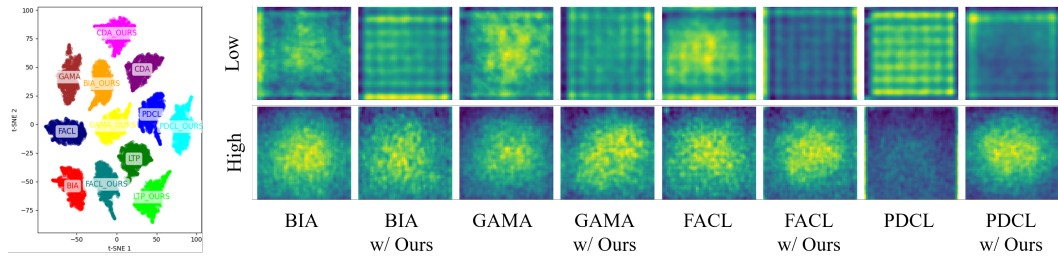

Figure S10: **Left**: t-SNE visualization of the intermediate generator features after `resblock3` for all methods, showing how the additions of Ours (SCGA) shift the internal feature geometry. The largest displacement appears for our primary baseline, BIA. **Right**: Low- and extreme high-band components of the same intermediate features, reconstructed in the image space by applying radial masks to their 2-D FFTs ($\rho < 0.2$ and $\rho \geq 0.8$) and averaging over the validation set. For each baseline and its SCGA-augmented variant, SCGA yields low-band reconstructions with brighter and more compact regions and high-band reconstructions that are less peaky and more spatially dispersed. This joint behavior agrees with the spectral ratios in Table 6 and indicates a redistribution of energy toward semantically aligned low frequencies together with attenuation of sharp surrogate-specific high-frequency artifacts.

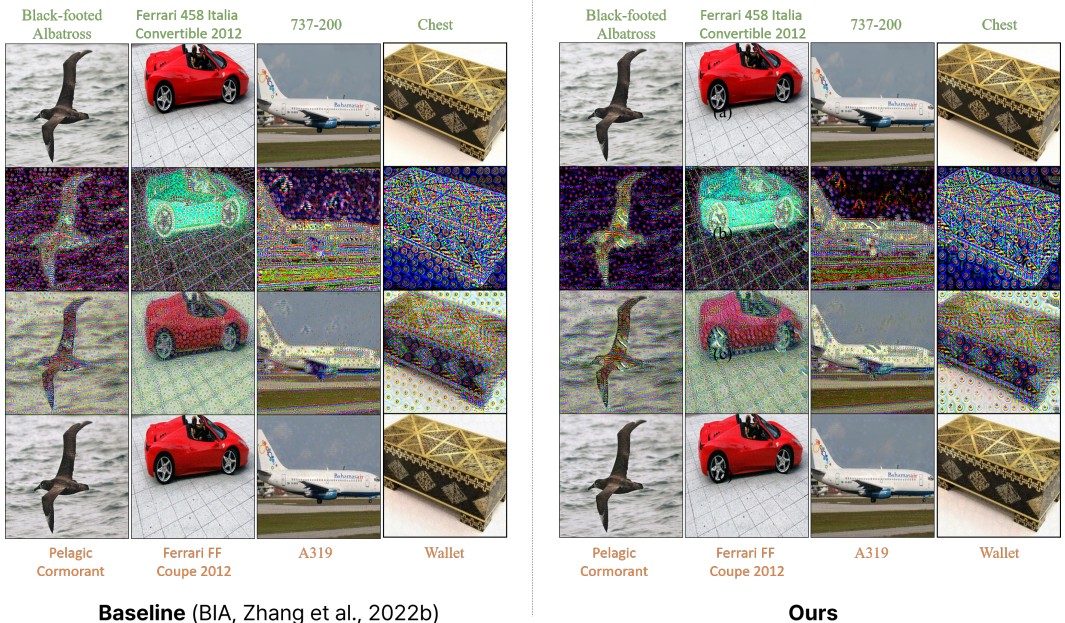

Figure S11: Side-by-side visualization of results on CUB-200-2011, Stanford Cars, FGVC Aircraft, and ImageNet. The attacked predictions are similar, yet the perturbation patterns are visibly different.