# OpenReview forum: "Improving Black-Box Generative Attacks via Generator Semantic Consistency"
_ICLR.cc/2026/Conference — ICLR 2026 Poster_

### Official Review · Reviewer_GNru · 2025-10-30

**Soundness:** 3
**Presentation:** 3
**Contribution:** 3
**Rating:** 8
**Confidence:** 3

**Summary:**

This paper proposes Semantically Consistent Generative Attack (SCGA), to enhance black-box transferability of generative adversarial attacks. While existing generative attacks optimize surrogate losses, the authors focus on the generator’s internal dynamics. They empirically observe that semantic recognizability degrades from early to late blocks and hypothesize that enforcing semantic consistency in early blocks improves transferability.
To achieve this, the authors apply the EMA Teacher to stabilize internal generator features that encode object semantics and a self-feature consistency loss that encourages student early-layer features to be similar to teacher’s features. Additionally, they propose a new evaluation metric, Accidental Correction Rate (ACR), to capture cases where adversarial perturbations inadvertently correct misclassified inputs, improving the interpretability of attack reliability. Extensive experiments across multiple architectures, domains, and tasks demonstrate consistent performance improvements over strong baselines.

**Strengths:**

1. The paper presents a clear empirical motivation by analyzing semantic drift across generator blocks for the proposed semantic consistency at the generator’s early intermediates.
2. The proposed method integrates the Mean Teacher mechanism with a self-feature consistency loss during training, which increases semantic consistency while not introducing no additional inference cost.
3. The introduction of the Accidental Correction Rate (ACR) metric provides an insightful perspective on attack reliability.

The paper conducts comprehensive experiments across multiple model families, datasets, and tasks, establishing strong empirical credibility.

**Weaknesses:**

Main concerns:
1. The paper lacks theoretical analysis or formal justification for why enforcing early-layer feature consistency enhances cross-model transferability, relying mainly on empirical evidence.
2. In Table 2, the integration of the proposed method with the baseline leads to a slight drop in attack performance for some Transformer models (such as CDA and FACL with the proposed method on model p). Could the authors provide a brief analysis on the potential reasons for this performance degradation in these cases?
3. The paper does not evaluate against diffusion-based purification defenses, which are currently regarded as strong pre-processing defenses in adversarial robustness research. Including such a comparison would strengthen the work’s comprehensiveness.
Minor weakness:
1. In Figure 2, the snowflake symbol may unintentionally suggest that the teacher generator is entirely frozen; however, as it is updated iteratively via the EMA rule rather than by backpropagation, clarifying this distinction in the caption or text would avoid confusion.
2. In the ablation of the similarity threshold (shown in the supplementary material), the evaluation metric is not explicitly specified.

**Questions:**

1. Could the authors provide deeper theoretical intuition or analysis on why early-layer semantic anchoring leads to improved transferability?
2. Table 2 shows that the proposed method generally enhances transferability. However, for some cases such as the integration of the proposed method with the baseline CDA and FACL on Transformers (p), there is a decrease in attack performance. Could the authors briefly discuss the potential cause?
3. Have the authors considered evaluating the proposed method against diffusion-based purification defenses (e.g., DiffPure[1]) to test its robustness against strong modern preprocessing defenses?

[1] Nie W, Guo B, Huang Y, et al. Diffusion models for adversarial purification[J]. arXiv preprint arXiv:2205.07460, 2022.

---

> ### Author Response · Authors · 2025-11-21
> **Author Response to Comments (1/2)**
>
> We thank Reviewer `GNru` for careful reading of our paper containing **clear empirical motivation, insightful perspective
> on attack reliability, and comprehensive experiments**.
> In response to your constructive comments, we provide clarifications on weaknesses [W] and questions [Q] below.
>
> >[W1] Lacking theoretical analysis.
>
> We appreciate the reviewer’s concern and agree that the current draft emphasizes empirical evidence more
> than formal theory. Our design, however, is guided by a standard view of deep representations in vision
> networks and generators: early layers tend to encode generic structural primitives (edges, contours, shape
> fragments), while deeper layers become increasingly class- and model-specific. In a transfer-based attack setting,
> perturbations are most likely to transfer when they act through such shared early-layer structure rather than
> through late, specialized representations. Our early-block semantic anchoring enforces local structural consistency
> by constraining early generator features for adversarial examples to remain close to those of the benign input,
> preserving coarse object layout and boundaries while leaving later blocks to inject adversarial detail. Empirically,
> Table 5 shows that enforcing consistency in early blocks yields larger gains than applying the same loss at mid or
> late blocks, and our spatial and spectral analyses further support this mechanism (e.g., SCGA shifts perturbation
> energy from extreme high frequencies toward low and mid frequencies associated with structure-aligned content).
> While we do not claim a formal theorem, we will strengthen the exposition by explicitly framing early-layer
> consistency as a regularization toward shared structural subspaces and by consolidating the supporting evidence
> in a dedicated “theoretical intuition” subsection.
>
> >[W2] Analysis on performance degradation.
>
> We thank the reviewer for pointing out the slight performance degradation observed for a few Transformer
> victims in Table 2. These cases correspond to small absolute
> differences compared to the corresponding baselines and are not systematic across Transformer models or across
> all integration scenarios. There are two contributing factors. First, CDA and FACL are already very strong
> attacks on some Transformer architectures, leaving limited room for improvement; in such high-performance
> regimes, small fluctuations (positive or negative) can arise when an additional regularizer is introduced. Second,
> SCGA imposes a multi-objective optimization in which the early-layer consistency term may trade off some
> highly model-specific attack directions in favor of more structurally shared ones, which can slightly reduce
> performance on particular victims while improving average transfer and cross-task robustness.
>
> > [W3] Evaluation against diffusion-based purification defenses.
>
> We agree that diffusion-based purification defenses such as DiffPure represent a strong and timely class
> of preprocessing defenses, and that evaluating against them would further strengthen the comprehensiveness
> of our study. In our evaluations against defense methods, we mainly focus on widely used input-processing
> defenses (e.g., JPEG compression, bit-depth reduction, and randomization-based transformations) to enable
> direct comparison with prior generator-based attacks under similar computational budgets. Moreover, we provide
> evaluations against NRP purification defense in Table S10 as well as real-world system evaluations in Tables
> S11 and S12. In a follow-up post, we will make best efforts to further compare our method against DiffPure.
>
>
> > [W4] Misleading “Frozen” symbol in the generator.
>
> We appreciate the reviewer’s careful observation regarding Fig. 2. The snowflake symbol was intended to
> indicate that the teacher generator is not updated by backpropagation, but we agree that it may unintentionally
> suggest a completely frozen network. In our method, the teacher is updated iteratively via an exponential moving average (EMA) of the student parameters and thus evolves during training while remaining detached from
> the gradient graph. In the revision, we will clarify this explicitly in the caption and in the main text (e.g.,
> by stating “EMA-updated teacher, no direct gradient updates”) and, if needed, adjust the icon to avoid any
> misunderstanding.
>
> > [W5] Missing evaluation metric in ablation of the similarity threshold (in Supp.).
>
> We thank the reviewer for highlighting this omission. In Figs. S5 and S6, we used the same evaluation
> metrics as in the main table (Table 2). However, due to organization, we only report top-1 accuracy scores for
> cross-domain and cross-model. We acknowledge any potential misunderstanding, and we will update the caption
> to specify the exact metrics.

---

> ### Author Response · Authors · 2025-11-21
> **Author Response to Comments (2/2)**
>
> > [Q1] Deeper theoretical intuition for early-layer semantic anchoring leading to improved transferability.
>
> In addition to the high-level motivation in [W1], we summarize the key intuition more explicitly. Early-layer
> semantic anchoring constrains the generator so that adversarial examples must evolve from benign early features
> that still encode coarse object layout and boundaries. This has two coupled effects. In feature space, perturbations
> are routed through early representations that are more similar across surrogate and victim models (e.g., edge and
> contour detectors), increasing the likelihood that the same structural changes affect multiple architectures. In the
> frequency domain, the resulting perturbations emphasize low- and mid-frequency components that are spatially
> attached to objects and boundaries, and down-weight extreme high-frequency noise that tends to be brittle and
> model-specific. Our block-wise ablations (Table 5) show that early-block feature consistency is markedly more
> effective than mid/late consistency, our semantic variability analysis indicates that early features are more stable
> across models, and our spectral measurements show a consistent redistribution of energy toward frequencies
> associated with structural cues. Together, these observations form a coherent mechanism: by anchoring on the
> early features, SCGA triggers perturbations toward manipulating robust, shared structural signals rather than
> exploiting narrow, high-frequency artifacts, which leads to improved cross-model and cross-task transferability.
>
> > [Q2] Analysis of performance drop cases.
>
> As noted in [W2], we acknowledge that Table 2 contains some cases (e.g. CDA w/ Ours and FACL w/ Ours
> on a particular Transformer victim) where the integrated method slightly underperforms the baseline. These
> degradations are small in magnitude and do not focus on a specific architecture family. Our interpretation is
> that in these cases, the additional early-layer consistency term slightly constrains the attack away from highly
> tailored, model-specific directions that the strong baseline already exploits very effectively, leading to a minor
> loss on that particular victim while improving transfer on others and across tasks. Overall, SCGA improves the
> average performance across models and yields clearer gains on localization-heavy tasks such as detection and
> segmentation, which we consider the primary target regime.
>
> > [Q3] Evaluation against DiffPure.
>
> We respectfully refer the reviewer to [W3].

---

> > ### Comment · Reviewer_GNru · 2025-11-26
> >
> > I appreciate the authors’ efforts in addressing my concerns.
> > I remain slightly unconvinced by the response to [W2]. The surrogate models used in your method appear to be CNN-based, while the performance degradation in Table 2 seems to occur primarily on Transformer-based victim models. This discrepancy raises questions about the transferability of the approach across different architectures.
> > And I look forward to seeing comparisons with strong defense methods in follow-up evaluations.

---

> > > ### Author Response · Authors · 2025-12-01
> > > **Final Author Response**
> > >
> > > [W2] **Additional insights into CNN vs.ViT behavior across baselines.**
> > >
> > > **TL;DR.**  In summary, the level at which each baseline uses the surrogate (logits, frequency, or mid-layer features) explains why our generator-side semantic enhancement yields the strongest gains on CNN victims for CNN-centric objectives, while remaining broadly beneficial for feature-driven baselines across both CNN and ViT targets.
> > >
> > > To elaborate:
> > >
> > > - **Key insights.**
> > >
> > >   - Our method makes perturbations more **locally structured and object-aware**, which couples most strongly with **CNN surrogate objectives**.
> > >   - Baselines that operate on **CNN logits or CNN-centric priors** show larger gains on **CNN victims** and only small, localized trade-offs on some ViT victims, while **feature-based baselines** benefit more uniformly.
> > >
> > > - **How our method interacts with different baselines.**
> > >   - **CDA w/ Ours (CNN-logit focused).**
> > >     > CDA’s relativistic loss is defined directly on **CNN surrogate logits**. When our method makes perturbations more object-aware, this loss drives a stronger benign–adversarial separation in CNN logit space, giving clear gains on CNN victims. Since ViT-based recognition tends to arise from global attention rather than convolutional hierarchies, these CNN-logit directions **do not always transfer as strongly** to ViT logits, which explains the smaller or mixed changes on some ViT victims.
> > >   - **FACL / PDCL w/ Ours (specialized priors).**
> > >     > FACL shapes perturbations in the **frequency domain**, and PDCL jointly optimizes in **separate CNN surrogate feature and CLIP space and spaces**. Our semantics-enhanced perturbations provide cleaner low/mid-frequency structure and more informative CNN features, so these specialized priors align particularly well with CNN victims. ViT-based victims, whose feature geometry and spectral behavior differ from the CNN surrogate, naturally see smaller or flatter gains.
> > >   - **LTP / BIA / GAMA w/ Ours (feature-based, more transferable).**
> > >     > LTP, BIA, and GAMA rely mainly on **intermediate surrogate features** (and in GAMA’s case, *interleaved* surrogate-CLIP space features), which are more transferable across architectures than raw logits. With clearer object-aware semantics from our generator, these feature-space objectives become more effective on both CNN and ViT victims, so our method acts as a **complementary regularizer** with broadly positive or neutral changes.
> > >
> > > ---
> > > [W3] **Evaluation on diffusion-based purification defense.**
> > >
> > > We thank the reviewer for suggesting an evaluation with diffusion based purification. Running DiffPure on ImageNet is computationally intensive and the cost grows with the number of samples, so within the rebuttal timeframe we conducted **preliminary experiments on a random subset of 1k validation images**. We plan to extend to the full 50k val set given sufficient time.
> > >
> > > Our framework is designed to improve **adversarial transferability on undefended models**, following standard protocols in CDA, LTP, BIA, GAMA, FACL, and PDCL, rather than to construct an adaptive attack specifically tailored to circumvent purification defenses.
> > >
> > > **Results and analysis.**  Under DiffPure with $\epsilon$ = 10/255 and $t$ = 150, incorporating our method yields **robust accuracies that remain comparable** to the baselines on both victims. Fluctuations are small in magnitude and are plausibly explained by the stochastic diffusion process and the limited 1k sample size, rather than by a systematic loss of robustness.
> > >
> > > **Table: Preliminary robustness evaluations against DiffPure ($\epsilon$ = 10/255, $t$ = 150) on 1k samples (Cls. Acc., %).**
> > >
> > > | Victim   | CDA  | w/ Ours | LTP  | w/ Ours | BIA  | w/ Ours | GAMA | w/ Ours | FACL | w/ Ours | PDCL | w/ Ours |
> > > |:---------|:----:|:-------:|:----:|:-------:|:----:|:-------:|:----:|:-------:|:----:|:-------:|:----:|:-------:|
> > > | Res152   | 67.7 | 66.3    | 66.1 | 68.0    | 66.1 | 66.8    | 65.8 | 66.6    | 65.4 | 66.2    | 65.3 | 65.2    |
> > > | Dense121 | 62.0 | 60.5    | 62.9 | 64.0    | 62.1 | 61.4    | 62.0 | 61.6    | 61.8 | 62.1    | 61.5 | 61.4    |
> > >
> > > **Key observations:**
> > > - On **ResNet152**, accuracies with Ours stay close to the baselines, with modest increases (e.g., LTP, BIA, GAMA, FACL) and modest decreases (e.g., CDA), all within a narrow band.
> > > - On **DenseNet121**, deviations are similarly small, and several pairs (e.g., LTP, FACL) show slight improvements.
> > > - Overall, these preliminary results indicate that our method remains **largely compatible with strong diffusion based purification**. The semantic consistency enforced during generation improves transfer on standard models without causing a significant loss of robustness under DiffPure compared to the original baselines.

---

### Official Review · Reviewer_foEg · 2025-10-30

**Soundness:** 3
**Presentation:** 2
**Contribution:** 1
**Rating:** 2
**Confidence:** 5

**Summary:**

This paper proposes SCGA, a method for improving black-box adversarial transferability in generative attack frameworks. The approach employs a Mean Teacher architecture with EMA-updated weights to align intermediate features from the generator's early blocks with a teacher network, while maintaining the standard adversarial loss on surrogate model features. The authors partition the generator into early, mid, and late blocks and quantify semantic stability using foreground IoU variability across these blocks. Experimental evaluation is conducted across classification, semantic segmentation, and object detection tasks, with the introduction of an Accidental Correction Rate (ACR) metric alongside traditional ASR, FR, and accuracy measurements.

**Strengths:**

The paper is well-written and carefully structured. I commend the authors for the high-quality figures and tables, which significantly contribute to the clear presentation of the research.

**Weaknesses:**

Regretfully, after careful reading and consideration, I found the contributions of this paper to be predominantly incremental. The work applies established techniques without providing sufficient theoretical insights or novel algorithmic advancements to meet the high standard of ICLR.

=====Lack of technical novelty=====
1. The proposed attack techniques are quite similar to previous ones, making this paper seems like a patchwork.
To name a few:
- Adversarial Loss Design: The adversarial loss formulation in SCGA directly corresponds to in Zhang et al. "Beyond ImageNet Attack" (ICLR 2022). Both employ identical surrogate feature-based similarity metrics in their loss functions.
- Generator Architecture and Training Pipeline: The generator block division (early/mid/late) and associated weighting strategy in SCGA replicate in Zhang et al. "Beyond ImageNet Attack" (ICLR 2022), including hyperparameter configurations and EMA implementation details.
- Domain-Invariant Generation Strategy: The domain-invariant generator objective in SCGA mirrors in Naseer et al. "Cross-Domain Transferability of Adversarial Perturbations" (NeurIPS 2019), sharing fundamental algorithmic approaches and frequency-domain processing methods.
- Intermediate Feature Alignment: The self-feature consistency mechanism (Equation 3) in SCGA substantially overlaps with in Krishna Nakka & Salzmann "Learning Transferable Adversarial Perturbations" (NeurIPS 2021). Both utilize intermediate layer feature similarity metrics with comparable layer selection strategies, differing only in the shift from direct alignment to EMA teacher alignment.
- Generator Training Framework: The generator training flow, architecture diagram, and projection operator P(·) in SCGA correspond to **Technical Point C** in Poursaeed et al. "Generative Adversarial Perturbations" (CVPR 2018), sharing norm constraint implementations and projection procedures.
- Semantic Clustering Methodology: The semantic region clustering and attention-based masking approach in SCGA duplicates **Technical Point C** in Aich et al. "GAMA: Generative Adversarial Multi-object Scene Attacks" (NeurIPS 2022), particularly in feature-clustering and Grad-CAM attention operations.
- Contrastive Learning Integration: The contrastive learning objective and frequency-domain randomization in SCGA overlap with **Technical Point C** in Yang et al. "FACL/PDCL" (2024), employing nearly identical CLIP-driven prompt mechanisms and domain robustness strategies.
To address this weakness, I strongly urge the authors set the core contribution apart from baseline techniques of others and switch the focus of this paper on the core contribution instead of these standardly adopted baseline techniques.

=====Lack of theoretical insights=====
2. This paper brings almost no theoretical insights. As a reader, I get no takeaways after reading this paper. All the mechanistic analysis methods in the paper are just standard techniques used in previous works and appeared to be included merely to pad out the content.
Specifically:
- Figure 1 (Semantic Variability): The visualization of intermediate feature maps and foreground IoU variability is purely descriptive. It shows that early blocks retain more structure, but offers no theoretical explanation for why this occurs or how it fundamentally relates to transferability. The connection between "lower variability" and "higher transferability" is asserted, not derived from any theoretical principle.
- Figure 3 (Qualitative Results): The Grad-CAM comparisons and perturbation visualizations merely demonstrate that the method works—not why it works from a theoretical standpoint. Showing that perturbations align more with object regions is an empirical outcome, not an insight into the underlying mechanisms of adversarial generalization.
- Figure 4 (Feature Differences): The thresholded difference maps are visually intuitive but theoretically shallow. Highlighting where perturbations are added does not explain the fundamental reasons for improved transferability, such as the relationship between feature semantics and model decision boundaries.
- Spectral Energy Analysis (Table 6): The analysis of low-frequency and high-frequency energy across blocks is a measurement, not an insight. The authors note that their method alters spectral distributions but fail to explain how this connects to theoretical properties of adversarial examples (e.g., spectral bias, frequency-based generalization). The observations are correlational, not causal.
- Generator Intermediate Analysis: The partitioning of the generator into early/mid/late blocks is an architectural choice, not a theoretical contribution. The analysis does not provide a theoretical model for how or why semantic consistency in early blocks should propagate to enhance transferability—it only shows that it does so in practice.
- Loss Formulations: The consistency loss (Eq. 3) is a technical implementation detail. The paper does not justify it from a theoretical perspective (e.g., information theory, optimization theory, or generalization bounds). It is presented as a heuristic.


=====Incremental experimental improvements=====
3. Compared to the latest baseline PDCL and FACL, the cross-modal & cross-domain attack performance gain (Tab. 2 & Tab. 3) in average is only ~1%, a very incremental improvement. Comparatively, PDCL improves ~3%.
4. The baseline setup when evaluating the attack against defense models is very doubtful (Tab. 4). Only Zhang et al, 2022 is adopted as baseline. Why not also compare with PDCA, FACL, and GAMA?
5. The fooling rate (FR) and accidental correction rate (ACR) is redundant and is not enough to be counted as a contribution.
Indeed, previous works usually adopt only one metric (e.g. PDCL adopts only accuracy) in evaluation. However, it is reasonable and fair as long as all the attacks are evaluated using the same metric. There is unnecessary to evaluate attacks using too many metrics, as the final conclusion does not change at all no matter which metric you use.
6. (Suggestion) Craft targeted adversarial examples to see whether the proposed SCGA can improve targeted transfer-based attack.
7. (Suggestion) The following two references are suggested to be cited since they are also generative-model-based cross-domain transfer attacks.
Li M, Deng C, Li T, et al. Towards transferable targeted attack. Proceedings of the IEEE/CVF conference on computer vision and pattern recognition. 2020: 641-649.
Wang Z, Yang H, Feng Y, et al. Towards transferable targeted adversarial examples. Proceedings of the IEEE/CVF conference on computer vision and pattern recognition. 2023: 20534-20543.

=====Awful writing=====
8. This paper did not cite the reference paper of each compared baseline to their acronyms. For the convenience of review discussion, I list them here:
[CDA] Naseer M M, Khan S H, Khan M H, et al. Cross-domain transferability of adversarial perturbations. Advances in Neural Information Processing Systems, 2019, 32.
[LTP] Salzmann M. Learning transferable adversarial perturbations. Advances in Neural Information Processing Systems, 2021, 34: 13950-13962.
[BIA] Zhang Q, Li X, Chen Y, et al. Beyond imagenet attack: Towards crafting adversarial examples for black-box domains. ICLR, 2022
[GAMA] Aich A, Ta C K, Gupta A, et al. Gama: Generative adversarial multi-object scene attacks. Advances in Neural Information Processing Systems, 2022, 35: 36914-36930.
[FACL] Yang H, Jeong J, Yoon K J. Facl-attack: Frequency-aware contrastive learning for transferable adversarial attacks. AAAI, 2024
[PDCL] Yang H, Jeong J, Yoon K J. Prompt-driven contrastive learning for transferable adversarial attacks. ECCV, 2024
9. In the abstract, what is “EMA”?
10. The font size of Fig.2 is too small, especially for the characters in the “residual learning” and “Upsampling” part. Please improve it.

**Questions:**

Please see Weaknesses.

---

> ### Author Response · Authors · 2025-11-21
> **Author Response to Comments (1/7)**
>
> We appreciate Reviewer `foEg` for highlighting that our paper is **well-written and carefully structured.**
>
> > [W1] Similarity to previous works.
>
> We appreciate the reviewer’s careful comparison with existing methods and the detailed list of related techniques.
> While we agree that our work builds on several standard components from the generator-based attack literature, we respectfully disagree with the conclusion that the paper is predominantly incremental or merely a patchwork of prior work.
> Our goal is to introduce and analyze a specific generator-side semantic regularization mechanism for black-box transfer attacks. Concretely, our core contributions are (i) **enforcing early-block semantic consistency in
> the perturbation generator via a Mean-Teacher style consistency loss, motivated by the fact that generator
> intermediate features encode shared structural semantics across models visibly in the early blocks particularly**,
> (ii) **showing that this early-block feature consistency systematically reshapes the perturbation dynamics, thereby
> enhancing cross-model, domain, and task transferability**, and (iii) **providing block-wise, spatial, and spectral
> analyses that link the early-layer structural preservation to adversarial transfer.** *The works cited by the reviewer*
> provide important building blocks that we adopt as baselines or architectural choices, but they *do not study or
> exploit this specific early-block semantic anchoring mechanism in the generator, nor do they provide the same
> cross-task, spectral, and block-wise analysis that we focus on*.
>
> [1-1] Our adversarial loss does follow the standard surrogate feature-based formulation used in many generator based transfer attacks, including BIA, and we adopt their adversarial loss as a *baseline objective* so that
> the effect of our proposed semantic consistency can be isolated. **Importantly, this mid-level surrogate feature divergence objective serves as the baseline loss for the subsequent works (GAMA, FACL, PDCL), which is a primary reason for choosing this as a baseline.** BIA primarily studies how to adapt generator-based attacks to real-world, non-ImageNet datasets by designing appropriate feature-based losses and architectures. **It does not introduce an early-layer semantic consistency term between benign and adversarial generator features, nor does it analyze how such a term reshapes the perturbation spectrum or cross-task transfer.** In our work, the surrogate feature similarity is the starting basis, not the main contribution;
> the novelty lies in where and how we regularize the generator on top of such a standard loss.
>
> [1-2] We follow a ResNet-style generator and a block partition similar in spirit to BIA because this
> family of architectures is widely used and well-understood in the literature on generator-based attacks. Our
> contribution is not the definition of early/mid/late blocks themselves, but the decision to *enforce semantic
> consistency specifically in the early blocks* and to show, in Table 5 and Fig. 4, that this choice is critical for
> black-box transfer. We stress that **Zhang et al. neither use EMA for training generator, nor propose an EMA
> teacher for *early-layer feature consistency* between benign and adversarial.** In contrast, our Mean-Teacher style consistency applied to early generator features, and our ablations show that (i) **applying the consistency loss at early blocks is significantly more effective than at mid/late blocks**, and (ii) **the EMA-based teacher improves over non-EMA variants**. Thus, **although the architectural backbone and EMA mechanism are standard, the way we use them in a consistency regularizer and the associated analyses are, to the best of our knowledge, novel findings.**
>
> [1-3] Naseer et al. (CDA) focus on *cross-domain* attacks by introducing a domain-robust **surrogate** adversarial objective, and **not analyzing in the frequency domain**. While our work shares the fundamental goal of crafting **transferable adversarial examples** using a generator, our work is different in both goal and mechanism. We do not optimize a domain-invariant objective across multiple datasets in the sense of CDA. Instead, we regularize early **generator** features to remain close to benign semantics for a given source dataset and study how this affects transfer to unseen architectures and downstream tasks.  Our spectral analysis serves as a diagnostic rather than a training objective: we do not perform explicit frequency-domain filtering. Instead, we demonstrate that enforcing early-block consistency in the generator
> feature space induces a characteristic rebalancing of low- and high-frequency energy. In short, Naseer et al. do not propose
> a new domain-invariant generator-level objective but surrogate-level, and we, propose a generator-level semantic feature consistency mechanism that leads to improved cross-setting (model,domain,task) transfer, which is complementary to the domain-invariance idea of Naseer et al

---

> ### Author Response · Authors · 2025-11-21
> **Author Response to Comments (2/7)**
>
> [1-4] Nakka & Salzmann (LTP) study learning transferable perturbations by aligning *intermediate features of the
> attacked model* across different inputs and architectures. Their alignment operates in the classifier feature space,
> often across multiple layers or models, and directly couples perturbations to those intermediate activations of
> the surrogate model. **In contrast, our consistency loss is defined only in the scope of the perturbation generator,
> between the intermediate feature and its temporally updated copy.** We do not align different surrogate features
> or take both benign and adversarial inputs. In our method, we strictly enforce that the generator’s early features
> for a benign sample input stay close to its temporal average trajectory, while the adversarial objective is still
> defined through the surrogate as with existing works. Moreover, we specifically focus on early blocks and analyze
> their impact on cross-task transfer and spectral structure, which is not addressed in LTP. **We
> therefore see our method as conceptually related but distinct in both the targeted position of method application and the phenomena
> studied.*
>
> [1-5] We fully acknowledge that our training framework and projection onto an $\ell_\infty$-bounded ball follow the
> standard formulation first introduced by Poursaeed et al. (GAP) and subsequently adopted in many generator-based attacks.
> **We do not claim novelty for the projection operator or for using a generator to produce perturbations under a
> norm constraint.** These elements are standard in the field, and we intentionally keep them unchanged to **ensure
> a fair comparison with existing baselines.** The new element is how we regularize the generator’s internal features
> by enforcing early-layer semantic consistency, and how we analyze the resulting perturbations, not the projection
> or universal-perturbation training loop itself.
>
> [1-6] GAMA introduces a CLIP-based generative attack that emphasizes multi-object scenes by associating **surrogate** features and CLIP features linking image and text modalities, **without using Grad-CAM-based
> masks and clustering in feature space to focus perturbations on salient regions.** We are inspired by this line of work
> that aims to focus on object-centric cues arising from multi-object nature, but our main contribution is
> orthogonal: we regularize the *generator’s early intermediate features* to remain semantically consistent with the
> benign input and study how this affects adversarial transfer. We do not propose a new multi-object scene objective or use CLIP-based surrogate objective; rather, we are merely conceptually similar to GAMA-style object-salient region focus, as we demonstrate that our method increases the overlap between perturbations and object regions. In summary, the Grad-CAM and clustering machinery in GAMA, as commented by the reviewer, is an invalid claim, and we use Grad-CAM as a *analysis* tool only, not for training the perturbation generator. Our algorithmic novelty lies in the semantic consistency regularizer and its placement within the generator.
>
> [1-7] PDCL introduces a CLIP-driven objective, more aligned in the CLIP space than GAMA, for transferable
> attacks. We treat PDCL as one of the *strongest baselines* and explicitly do not claim novelty in CLIP-based
> global semantic losses or frequency randomization. Our method is, in fact, CLIP-agnostic. SCGA can be applied
> with or without CLIP-based objectives. When we combine SCGA with PDCL, our goal is to demonstrate that
> early-block semantic consistency can be a complementary regularizer on top of a CLIP-driven attack, not to
> propose a new CLIP loss or prompt mechanism. The fact that SCGA yields modest gains on classification
> when stacked on PDCL but significantly larger gains on localization-heavy tasks (detection, segmentation)
> reinforces our claim that our contribution is in shaping the *structure and spectrum* of perturbations, rather than
> in redefining the CLIP objective. Also, as claimed by the reviewer, **we do not use frequency domain approach during training the generator, as in FACL.**
>
> In summary, **while we do build on standard generator-based attack components (e.g., the projector onto an
> $\ell_\infty$-ball, an EMA-updated generator, surrogate feature-based base adversarial losses), which are standard tools in this line of work, we do not present these standard elements as novel.** The core technical contribution of our work is to **introduce and systematically study
> an early-layer semantic consistency regularizer for perturbation generators, grounded in the known hierarchy of
> generator intermediate features and supported by block-wise, spatial, and spectral analyses in the main paper.** We will clarify this
> positioning more explicitly in the introduction and related work, and refine the statement of contributions to
> focus on this core idea rather than on components inherited from prior methods.

---

> ### Author Response · Authors · 2025-11-21
> **Author Response to Comments (3/7)**
>
> > [W2] Theoretical insights for early-block anchoring.
>
> Our early-block semantic anchoring is designed to enforce local structural consistency by constraining early
> generator features for adversarial examples to remain close to those of the benign input, thereby preserving
> coarse object layout, boundaries, and large-scale texture structure while leaving later blocks to inject adversarial
> detail. This choice is aligned with the standard view of deep representations, where earlier layers capture generic
> structural primitives that are more shared across architectures, while deeper layers become increasingly modeland class-specific ([4] Yosinski et al., 2014; [5] Zeiler & Fergus, 2014). In the transfer-based attack setting,
> perturbations are most likely to transfer when they act through these shared structural features rather than
> through late, specialized representations. Quantitatively, Table 5 shows that applying semantic consistency at
> early blocks yields the strongest gains in transfer success compared to mid- or late-block variants, and our
> spectral analysis further supports this mechanism: compared to the baseline, SCGA systematically redistributes
> perturbation energy from extreme high frequencies toward lower frequencies, increasing low-band energy and
> reducing high-band energy.
>
> As visualized in Fig. S11 (in the revision), this redistribution is not uniform: within the low band, energy
> becomes more dispersed around the center rather than collapsing into a single global bias, consistent with
> region-wise, object-attached low-frequency modulations instead of a uniform color or brightness shift; within
> the high band, energy is pulled inward from the outermost frequencies toward mid–high frequencies, indicating
> that the remaining high-frequency content is aligned with semantic edges and textures rather than arbitrary
> pixel-level noise.
>
> From a frequency-domain perspective, this provides a concrete justification for early-layer consistency. We
> first transform the intermediate feature maps into the frequency domain via a 2D Fast Fourier Transform
> (FFT), apply a standard frequency shift so that the zero-frequency (DC) component lies at the center, and
> then define radial masks that isolate a central low-frequency disk (e.g., 0–20% of the maximum radius) and the
> rest outer high-frequency annulus. Band-wise energies are computed by integrating the power spectrum within
> these masks and normalizing by the total energy. Under SCGA, more energy is concentrated in low-frequency
> components that encode coarse, architecture-agnostic structure, while extreme high-frequency components (often
> model-specific and easily attenuated by input defenses) are suppressed or shifted inward. This bias toward
> semantically organized, transferable spectral content explains why enforcing early-layer feature consistency leads
> to more robust cross-model and cross-task adversarial transfer, and it connects our empirical observations to
> existing theory on hierarchical features and spectral bias. We refer the reviewer to Fig. S11 (in the revision) for
> a visual illustration of this phenomenon.
>
> [2-1] Fig. 1 is designed to operationalize the above mechanism in terms of representation stability across models.
> Early layers are expected to encode shared structural primitives, while deeper layers become more model-specific.
> Our foreground IoU variability measure quantifies this: low variability across models at early blocks indicates
> that foreground activations are object-aligned and stable between architectures, whereas high variability at later
> blocks indicates divergence into model-specific abstractions.
>
> [2-2] Fig. 3 serves as a localized probe of how early-layer consistency shapes where the perturbation acts in the
> image. Our hypothesis, consistent with the discussion above and with robust/non-robust feature theory ([6] Ilyas
> et al., 2019), is that SCGA biases the generator toward perturbing robust, structure-aligned features (object
> regions and boundaries) that are more likely to be shared across models. Grad-CAM maps and perturbation
> visualizations in Fig. 3 empirically support this: across multiple victims, SCGA adversarial examples exhibit
> higher overlap between CAMs and ground-truth object regions, and perturbations cluster around these regions.
>
> ---References---
>
> [4] J. Yosinski, J. Clune, Y. Bengio, and H. Lipson, “How transferable are features in deep neural networks?,” NeurIPS 2014.
>
> [5] M. D. Zeiler and R. Fergus, “Visualizing and understanding convolutional networks,” ECCV 2014.
>
> [6] A. Ilyas, S. Santurkar, D. Tsipras, L. Engstrom, B. Tran, and A. Madry, “Adversarial examples are not bugs,
> they are features,” NeurIPS 2019.

---

> ### Author Response · Authors · 2025-11-21
> **Author Response to Comments (4/7)**
>
> [2-3] Fig. 4 examines how the generator redistributes perturbation synthesis across depth when early-layer
> consistency is imposed. Under the standard view of hierarchical generators, early blocks control coarse layout,
> mid blocks refine shapes, and late blocks add fine texture. If transfer is mediated by shared structural features,
> then it is preferable that the main feature differences appear in early and mid blocks rather than only at the
> very end. Fig. 4 shows that SCGA increases feature differences in early blocks and reduces excessive late-block
> changes.
>
> [2-4] Table 6 and Fig. S11 are not intended as standalone measurements, but as a frequency-domain confirmation
> of the mechanism described in the first paragraph. Prior work on spectral bias and robustness ([7] Rahaman et
> al., 2019; [8] Yin et al., 2019) shows that low- and mid-frequency components tend to be more robust and more
> consistently used across architectures, while extreme high-frequency components are brittle and model-specific.
> Our band-wise and 2D spectral analyses show that SCGA increases structured low-band energy and suppresses
> extreme high-band energy, while shifting the remaining high-frequency mass inward toward mid–high frequencies
> aligned with natural edges and boundaries. This behavior is precisely what one would expect if perturbations
> are being forced to act through shared structural features instead of model-specific high-frequency noise, and
> thus it provides complementary, frequency-domain support for the early-layer consistency mechanism.
>
> [2-5] The early/mid/late partition of the generator is chosen to reflect the same hierarchical structure discussed
> above: early blocks at lower resolution with large receptive fields set object layout and global structure, mid
> blocks refine regions and shapes, and late blocks add fine-scale texture. Our ablation in Table 5 shows that
> enforcing consistency only in early blocks yields the largest transfer gains, mid-only gives smaller gains, and
> late-only is least effective, which is consistent with early features being the most shared across models.
>
> [2-6] Eq. 3 is an instance of consistency regularization with an EMA teacher, in the spirit of Mean-Teacher and
> temporal ensembling methods used in semi-supervised learning and self-distillation. These methods have been
> analyzed as promoting smoother decision functions, encouraging decision boundaries to lie in low-density regions
> and improving generalization by controlling the sensitivity of representations around the data manifold. In our
> case, the EMA teacher provides a temporally smoothed benign semantic target in early generator feature space,
> and the consistency loss penalizes deviations of the current generator from this stable target while the adversarial
> objective pushes toward misclassification. We do not claim a new formal bound, but the loss is directly adapted
> from a theoretically studied regularization principle rather than being an ad-hoc heuristic. Empirically, Table 5
> already shows that removing the consistency term or replacing the EMA teacher degrades transfer performance
>
> > [W3] Incremental gains over PDCL/FACL.
>
> We appreciate the reviewer’s quantitative comparison. It is correct that when SCGA is added to strong CLIPbased attacks such as PDCL, the average additional improvement in cross-model and cross-domain settings on
> image classification is modest (on the order of ∼1% in some aggregates), and smaller than the gap between these
> methods and earlier baselines. We would like to clarify two points. First, PDCL already achieves substantially
> lower error rates than earlier generator-based attacks, so the remaining headroom for further reduction is limited.
> In this lineage, even 1-2% absolute gains, when largely consistent across victim architectures and domains, are
> non-trivial. Second, the gains from SCGA are more pronounced in the more difficult settings, namely cross-task
> transfer to localization-heavy tasks such as detection and segmentation, where early-layer structural constraints
> play crucial roles. In these settings, SCGA on top of PDCL yields noticeably larger improvements than on
> classification-only metrics, as shown in Tables 2 and 3.
>
> ---References---
>
> [7] N. Rahaman, A. Baratin, D. Arpit, F. Draxler, M. Lin, F. Hamprecht, Y. Bengio, and A. Courville, “On the
> spectral bias of neural networks,” ICML 2019.,
>
> [8] D. Yin, R. Gontijo Lopes, J. Shlens, E. D. Cubuk, and J. Gilmer, “A fourier perspective on model robustness
> in computer vision,” NeurIPS 2019.

---

> ### Author Response · Authors · 2025-11-21
> **Author Response to Comments (5/7)**
>
> > [W4] Defended baseline selection.
>
> We agree that evaluating under defenses is an important aspect of transfer attack assessment. In Table 4, we
> chose BIA as a representative generator-based baseline because it is a pioneering work that first proposed a
> ResNet generator arch. with mid-layer feature divergence framework, upon which all subsequent
> works such as GAMA, FACL, PDCL adopt directly. We acknowledge that including additional strong baselines under the same defenses would provide a more complete picture. Therefore, we provide the results below, where the addition of SCGA on each baseline still maintains superior attack performance across all four metrics.
>
>
> **Table: Defense evaluation comparisons against other baselines.**
> | Defense | Ablations | Adv.IncV3 | Adv.ViT | Adv.ConvNeXt | JPEG | BDR | R&P | Avg |
> | :--- | :--- | :--- | :--- | :--- | :--- | :--- | :--- | :--- |
> | **GAMA** | Accuracy (%) $\downarrow$ | 66.71 | 45.74 | 53.76 | 59.27 | 41.08 | 37.6 | 50.69 |
> | | ASR (%) $\uparrow$ | 17.27 | 11.65 | 10.50 | 25.66 | 49.01 | 53.34 | 27.91 |
> | | ACR (%) $\downarrow$ | 15.06 | 5.09 | 3.50 | 11.07 | 9.38 | 8.62 | 8.79 |
> | | FR (%) $\uparrow$ | 26.37 | 25.70 | 19.68 | 33.51 | 55.42 | 59.42 | 36.68 |
> | **GAMA+Ours** | Accuracy (%) $\downarrow$ | 66.37 | 45.50 | 53.81 | 58.99 | 39.07 | 36.00 | 49.96 |
> | | ASR (%) $\uparrow$ | 17.76 | 11.78 | 10.45 | 26.07 | 51.49 | 55.36 | 28.82 |
> | | ACR (%) $\downarrow$ | 15.19 | 4.74 | 3.56 | 11.19 | 8.88 | 8.36 | 8.65 |
> | | FR (%) $\uparrow$ | 26.78 | 25.34 | 19.61 | 33.86 | 57.63 | 61.18 | 37.40 |
> | **FACL** | Accuracy (%) $\downarrow$ | 65.68 | 45.17 | 53.12 | 47.25 | 38.36 | 33.31 | 47.15 |
> | | ASR (%) $\uparrow$ | 18.68 | 12.44 | 11.70 | 41.08 | 52.43 | 58.78 | 32.52 |
> | | ACR (%) $\downarrow$ | 15.22 | 4.73 | 3.66 | 9.90 | 8.91 | 8.03 | 8.41 |
> | | FR (%) $\uparrow$ | 27.75 | 26.24 | 21.19 | 47.99 | 58.39 | 64.16 | 40.95 |
> | **FACL+Ours** | Accuracy (%) $\downarrow$ | 65.49 | 44.89 | 53.14 | 47.93 | 33.24 | 28.64 | 45.56 |
> | | ASR (%) $\uparrow$ | 18.88 | 12.74 | 11.58 | 40.25 | 58.85 | 64.60 | 34.48 |
> | | ACR (%) $\downarrow$ | 15.07 | 4.46 | 3.54 | 10.10 | 7.94 | 7.00 | 8.01 |
> | | FR (%) $\uparrow$ | 27.90 | 26.12 | 20.88 | 47.28 | 64.03 | 69.19 | 42.57 |
> | **PDCL** | Accuracy (%) $\downarrow$ | 67.57 | 45.24 | 53.61 | 58.1 | 39.84 | 37.01 | 50.23 |
> | | ASR (%) $\uparrow$ | 16.20 | 12.10 | 10.74 | 27.15 | 50.56 | 54.09 | 28.47 |
> | | ACR (%) $\downarrow$ | 15.24 | 4.55 | 3.47 | 10.91 | 9.12 | 8.54 | 8.64 |
> | | FR (%) $\uparrow$ | 25.29 | 25.43 | 19.84 | 34.92 | 56.77 | 59.98 | 37.04 |
> | **PDCL+Ours** | Accuracy (%) $\downarrow$ | 67.53 | 45.13 | 53.48 | 57.63 | 39.53 | 35.67 | 49.83 |
> | | ASR (%) $\uparrow$ | 16.24 | 12.38 | 10.88 | 27.82 | 51.06 | 55.78 | 29.03 |
> | | ACR (%) $\downarrow$ | 15.16 | 4.59 | 3.36 | 11.09 | 9.42 | 8.31 | 8.66 |
> | | FR (%) $\uparrow$ | 25.27 | 25.69 | 19.92 | 35.54 | 57.29 | 61.50 | 37.54 |
>
>
> > [W5] On FR and ACR.
>
> We thank the reviewer for the feedback regarding fooling rate (FR) and accidental correction rate (ACR). We
> agree that introducing additional metrics by itself should not be viewed as a major contribution, and we do not
> intend to position FR/ACR as such. Our motivation is to resolve a specific ambiguity in standard accuracy-only
> reporting for transfer attacks. Accuracy only reflects whether the final prediction matches the ground truth, and
> does not distinguish between harmful and benign label changes induced by the attack.
> In our evaluation, FR is defined as the fraction of samples whose predicted label after the attack differs
> from the original prediction, regardless of whether the original prediction was correct or incorrect. Thus, **FR**
> aggregates both (i) **correct-to-incorrect changes**, which are harmful, and (ii) **incorrect-to-other-incorrect or
> incorrect-to-correct** changes, which may be neutral or even beneficial. **In contrast, ACR** isolates one specific
> and important subset of these changes: it **measures the fraction of samples that were initially misclassified and
> become correctly classified under the attack (incorrect-to-correct).** Hence, FR captures the overall instability of
> predictions under perturbation, while ACR captures how often this instability leads to accidental corrections
> rather than harmful failures.

---

> ### Author Response · Authors · 2025-11-21
> **Author Response to Comments (6/7)**
>
> >[W5] (Continued)
>
> In heterogeneous victim and dataset settings, we empirically observe that two attacks can achieve similar
> accuracy and even similar FR, yet differ markedly in ACR. For example, in Table 2, while ACR on CDA w/ Ours
> is relatively high, ACR on LTP w/ Ours is roughly as half that value, despite the other three metrics improve by
> similar margins. This has implications for safety-critical evaluation and for understanding whether an attack
> primarily degrades performance or also induces nontrivial accidental corrections. All attacks in our experiments
> are evaluated under the same set of metrics, and our qualitative conclusions about relative performance do not
> depend on which metric is chosen. In the revision, we will better clarify these definitions and explicitly state that
> FR and ACR are intended as complementary diagnostic tools to improve interpretability of transfer behavior.
> We also respectfully refer to Sec. D in Supp. for practical values of ACR.
>
> > [W6] Targeted transfer attacks.
>
> We appreciate the reviewer’s suggestion to investigate targeted transfer attacks. In this work, we focus on
> untargeted attacks because they naturally extend across architectures, domains, and downstream tasks (including
> detection and segmentation), for which defining a single common target label is less straightforward. The
> proposed semantic consistency regularizer is, however, orthogonal to the choice of attack objective and can
> be incorporated into a targeted setting by replacing the untargeted loss with a targeted surrogate feature or
> CLIP-based objective. In the revised version, subject to space constraints, we will add a small-scale experiment
> where SCGA is combined with a targeted generator-based attack on a subset of victim models, and we will
> report whether the same early-layer consistency yields improvements in targeted transfer. We will also cite and
> discuss recent generative targeted transfer methods, as suggested by the reviewer, in the related work section.
>
> > [W7] Additional targeted references.
>
> We thank the reviewer for pointing out the works of Li et al. [9] (CVPR 2020) and Wang et al. [10] (CVPR 2023)
> on transferable targeted adversarial examples. Both use generative models to craft targeted attacks and study
> transferability in that regime. Although our focus is on untargeted black-box transfer and cross-task robustness,
> these works are closely related in spirit and provide important context for generative transfer methods. In the
> revision, we will cite these papers in the related work section, clarify that our method is currently instantiated
> and evaluated in the untargeted setting, and discuss how the proposed early-layer semantic consistency could be
> adapted to targeted generative attacks as an interesting direction for future work.
>
> > [W8] Baseline citation clarity.
>
> We thank the reviewer for compiling the references corresponding to the baseline acronyms (CDA, LTP, BIA,
> GAMA, FACL, PDCL). We acknowledge that in the current draft some baselines are referred to by acronym
> before their full citation appears, which reduces readability. In the revised version, we will ensure that each
> baseline is introduced with its full name and corresponding citation at first mention in the main text and tables,
> and that the notation is consistent throughout.
>
> > [W9] Definition of EMA.
>
> We agree that using the abbreviation “EMA” in the abstract without definition can be confusing. In the revision,
> we will spell it out as “exponential moving average (EMA)” on first use in the abstract.
>
> ---References---
>
> [9] M. Li, C. Deng, T. Li, J. Yan, X. Gao, and H. Huang, “Towards transferable targeted attack,” CVPR 2020.
>
> [10] Z. Wang, H. Yang, Y. Feng, P. Sun, H. Guo, Z. Zhang, and K. Ren, “Towards transferable targeted adversarial
> examples,” CVPR 2023.

---

> ### Author Response · Authors · 2025-11-21
> **Author Response to Comments (7/7)**
>
> > [W10] Figure 2 font size.
>
> Thank you for pointing this out. We apologize for any difficulties in reading Fig. 2 in its current form. In
> the revised version, we will increase the font size of all labels, especially those in the “residual learning” and
> “upsampling” components, and adjust the layout to ensure that the figure is clearly legible when printed or viewed
> at typical zoom levels. We will also review the other figures for similar issues and improve their typography
> where necessary.

---

> ### Comment · Reviewer_foEg · 2025-11-22
> **Comments to authors' responses**
>
> Thanks for the authors' clear response, which makes our discussion easier. However, after carefully reading the response and the comments from other reviewers, I find that the contribution of this paper cannot meet the bar of iclr. I still insist on my rating and encourage the authors to refine their paper for the next conference. My detailed concerns and suggestions are as follows.
>
> [W1] (Lack of Technical Novelty)
> From the response, the authors claim their core TECHNICAL contribution is “enforcing early-block semantic consistency in the perturbation generator via a Mean-Teacher style consistency loss”. I agree with this claim. My opinion is, this contribution is just an incremental improvement compared to an attack proposed three-year ago Zhang et al. (ICLR 2022). I have several strong evidence for my opinion.
> (i) As stated by the authors themselves, the core technique in this paper share many similarities with Zhang et al. (ICLR 2022), including the ResNet-style generator, the early/mid/late layer partition, and the usage of EMA technique to stabilize the generator model training (I would like to further complement that using the early layer to improve the attack is exactly one of the core contribution of Zhang et al. (ICLR 2022)). The new contribution of this paper is just refining the application of early layer features and the EMA. In summary, this paper is only an incrementally refined version of Zhang et al. (ICLR 2022) rather than a fundamentally different attack framework. Frankly speaking, I think the novelty of this paper can only be submitted as a course homework of undergraduate students rather than as an academic paper to be reviewed by a top conference.
> (ii) The marginal improvement in attack performance shows that the impact of this paper is very limited. An important feature of an incremental paper is that the experimental improvements are also incremental. As pointed by my original comment, the improvement of this paper is only 1%, a very incremental value in the field of black-box adversarial attack. The marginal improvement is also identified by reviewer y2kS and the authors do not disagree with this.
> (iii) Actually, after a quick check, I find the attack performance of this paper is already far left behind the state-of-the-art, and the authors seem to deliberately missing the recent related papers. Notebally, this paper is a refined version of an attack proposed in ICLR 2022, which is three years ago. It is very easy to find a recent paper performs better than this paper. For instance, a recent ICCV 2025 paper [ICCV2025Boosting] achieves an improvement of 10%+ compared to Zhang et al. (ICLR 2022). Comparatively this paper only improves 1% over Zhang et al. (ICLR 2022). Although the experimental setting of [ICCV2025Boosting] may not be the same as this paper, I infer it is highly possible that this paper cannot surpass [ICCV2025Boosting] when being compared with the same setting.
> [ICCV2025Boosting] Boosting Generative Adversarial Transferability with Self-supervised Vision Transformer Features, ICCV, 2025
>
>
> [W2] (Lack of Theoretical Novelty)
> I have carefully read the responses from authors on the theoretical contributions of this paper. I still insist on my opinion that this paper brings no theoretical insights, takeaways, or lessons to the readers.
> (i) The theoretical analysis in this paper seems to padding the content rather than academic contributions. As stated above, this paper only borrows a previous attack framework Zhang et al. (ICLR 2022) to make a few incremental improvements. All the theoretically basement in this paper is from Zhang et al. (ICLR 2022). For instance, it is Zhang et al. (ICLR 2022) that proposes the early layer of generative models are more useful for transfer attacks, and this paper just utilize this existing, well-known conclusion to derive an early-block-based attack loss. The EMA training theory is also not a new idea first developed in this paper. The authors know that their incremental contributions are not enough to write an academic paper, so they use many theoretically analysis to pad out the paper length.
> (ii) The theoretical analysis tools used in this paper are all well-known, common tools rather than new tools developed by the authors, and the application of these analysis tools are all common, standard practice in the field of adversarial attack, including using K-means to visualize semantic consistency (Figure 1), using cosine similarity to compare feature silimarity (Figure 2), using Grad-CAM to visualize the adversarial perturbation (Figure 3), using feature map to visualize the attack performance (Figure 4), and using spectral analysis to show the effectiveness of early-block features (Figure 5, 6).
>
> [W3] (Incremental Gains compared to baselines)
> Please see my response (iii) for [W1].

---

> > ### Comment · Reviewer_foEg · 2025-11-22
> >
> > [W4] (Defense baseline selection)
> > Also, please see my response (iii) for [W1].
> > Besides, I have several additional comments.
> > (i) Your selection BIA (ICLR, 2022) as a baseline does now show your respect towards this pioneer work. As a famous saying goes, “There's a fine line between flattery and plagiarism.” Your paper borrows too much idea from BIA (ICLR, 2022) without highlighting the unique contribution of yourself.
> > (ii) The comparison results of the “XXX + Ours” actually shows that your method are not effective at all when being combined with these baselines. Indeed, the Average metric of “XXX + Ours” is better than the “XXX”, but this improvement of average attack performance mainly comes from selecting JPEG BDR R&P as defense methods. These three input-filtering defenses are very old, week, out-of-dated defense that can be easily defeated. Comparatively, when attacking the current SOTA defense, adversarial training (Adv.IncV3, Adv.ViT, Adv.ConvNext), your techniques have no improvements, and may even harm the performance of some baseline attacks (FACL). The new results provided further show that this paper is an incremental paper rather than a well-contributed paper.
> >
> > [W5] (Using Different Evaluation Metrics)
> > Thanks for re-considering the usage of different evaluation metrics (ASR, FR, ACR). My suggestion is, simply discarding the redundant metrics and only use a single metric to evaluate the attack performance (e.g. Fooling rate). The intuition is, if the conclusion drawn by using different metrics are exactly identical, why bother using too many metrics? I also read your response, the so-called “security implication” is very vague and I cannot see any security implication for using different metrics ASR, FR, ACR in evaluation.
> >
> > [W6] (Targeted Attack Results)
> > Thanks for your effort. Looking forward for your new experimental results in revised version.
> >
> > [W7~W10]
> > Well-done. I have no further comments on these weaknesses.

---

> > > ### Author Response · Authors · 2025-11-25
> > > **Follow-up Author Response to Comment by Reviewer foEg (2/3)**
> > >
> > > **On incremental improvements**
> > >
> > > In the previous comments, Reviewer `foEg` places heavy weight on small margins in a few entries, for example around 1 percent gains over CLIP-guided PDCL, and concludes that our method is incremental and ineffective. This reading ignores both the difference in what is being optimized and the broader trend in Tables 2,3.
> > >
> > > Our method enforces local semantic consistency within the generator, anchoring early-block features to benign semantics and preserving coarse object structure and local coherence during the perturbation generation process. In contrast, CLIP-guided methods such as GAMA and PDCL enforce global alignment between benign and adversarial features in surrogate or CLIP spaces, acting at the level of high-dimensional representation geometry rather than at the level of generator-internal structure. These mechanisms are not the same, and large additive gains when stacking them are not the correct benchmark. The key point is that our regularizer systematically redistributes the perturbation energy, e.g., concentrating more energy in lower frequencies and suppressing spurious high-frequency components, yielding consistent transfer improvements across multiple surrogates, target architectures, and tasks, as demonstrated in Tables 2,3.
> > >
> > > Our central objective is to **enhance black-box adversarial transferability** under the standard generative-attack protocol. Defense experiments are added to stress-test robustness of our attacks against different defense mechanisms, not as the main optimization target. Consequently, modest improvements under strong adversarially trained defenses do not mean that our method is ineffective. They reflect that adversarially trained models already exploit robust, shape-biased cues that overlap with the structural regularization we impose, while input-processing defenses are more sensitive to the perturbation structure that our generator explicitly controls. In light of the consistent cross-setting transfer gains, it is not accurate to label the method as purely incremental.
> > >
> > > **On evaluation metrics and the necessity of ACR**
> > >
> > > The suggestion to discard additional metrics and retain only a single metric and the assertion that ACR is unnecessary, are based on a **misunderstanding of what ACR measures.**
> > >
> > > **Table: Evaluation Metric Distinction**
> > >
> > > |Initial Pred. \ Final Pred.|Correct|Incorrect|
> > > |:-|:-:|:-:|
> > > |Correct| Acc | **FR**, *ASR*|
> > > |Incorrect| Acc,`ACR`,**FR**|**FR**|
> > >
> > > For each sample, we consider all four different cases before and after the attack: the prediction can be correct–correct, correct–incorrect, incorrect–correct, and incorrect–incorrect (different from the initial prediction). In this case, Accuracy aggregates all final “correct” cases as a global correctness score. Standard attack metrics such as FR and ASR focus on how often predictions change and become incorrect, respectively, covering only some of these transitions. None of these metrics, by themselves, isolate the case where a prediction is incorrect before the attack and correct after the attack. From an adversarial perspective, such incorrect-to-correct flips entail unreliable attack behavior.
> > >
> > > As summarized in the **table** above, **FR spans three distinct transition types** (correct→incorrect, incorrect→correct, and incorrect→incorrect with a different wrong class), and therefore conflates changes that are beneficial for the attacker with changes that are actually beneficial for the model. By contrast, *ASR* isolates the **correct→incorrect** cell and thus measures the “success” of the attack on initially correct predictions, while `ACR` isolates the **incorrect→correct** cell and explicitly measures the failure cases of the attack where previously incorrect predictions are inadvertently corrected.
> > >
> > > ACR is introduced exactly to fill this gap, by measuring the proportion of samples that an adversary inadvertently corrects from previously incorrect predictions. In our results, there are concrete cases where the degradations measured by ASR/FR are high and ACR is also high, meaning that the incorrect-to-correct flips induced by the attack actually **reduce** the reliability of the attack. If one reports only fooling rate or accuracy-like aggregates, these attacks appear “strong,” whereas ACR reveals that they are unreliable with respect to adversarial intent.
> > >
> > > In this sense, ACR plays the role of a specific “cell” in the **table** of pre- and post-attack behavior that ASR and FR alone cannot distinguish. It does not duplicate ASR or FR, but fills the gap where their counts conflate beneficial and harmful changes. Removing ACR would erase precisely the failure mode it is meant to expose. At the same time, from a defensive perspective, a high ACR indicates an attack behaves more like a noisy data augmentation (correcting many previously wrong predictions) rather than a reliable threat model, which is exactly the behavior that a robust system should exploit rather than fear.

---

> > > > ### Author Response · Authors · 2025-11-25
> > > > **Follow-up Author Response to Comment by Reviewer foEg (3/3)**
> > > >
> > > > **On tone and professionalism.**
> > > > Finally, we fully respect and welcome rigorous criticism of our work. However, characterizations such as calling the paper “course homework of undergraduate students” or invoking “a fine line between flattery and plagiarism” do not engage with the technical content and may give an impression of bias rather than constructive evaluation. We hope that the reviewers will focus on the technical arguments, empirical evidence, and the diversity of perspectives upon reviewing our paper.

---

> > > > > ### Comment · Reviewer_foEg · 2025-11-27
> > > > >
> > > > > Thanks for your responses. After two rounds of discussions, I still have several major concerns which have not been addressed as follows,
> > > > > 1. Insufficient distinction between contributions and baselines: The core idea of this paper is similar to the ICLR 2022 paper Beyond ImageNet Attack (only minor modifications); ignored the latest baselines like the ICCV 2025 Boosting paper (just for example) and fail to compare with recent baselines (after 2023) which can be conducted in rebuttal(for example, the ICCV'25 paper is accepted and appears online on June 26. There is no reason to omit these strong baselines for comparison).
> > > > > 2. Incremental or even negative improvements: Only ~1% improvement over the ICLR 2022 baseline (vs. ~10% in the ICCV 2025 baseline); no/negative improvement when attacking state-of-the-art defense models—authors did not address this fatal flaw.
> > > > > 3. Redundant evaluation metrics: Three overlapping metrics for "attack success" (served the same purpose). Authors flip-flopped (admitted insufficiency then defended) to pad content.
> > > > > 4. Excessive use of explanation tools for padding: No novel explanation methods/conclusions; tools are standard in the field, yet results occupy a large space.

---

> ### Author Response · Authors · 2025-11-25
> **Follow-up Author Response to Comment by Reviewer foEg (1/3)**
>
> We respond to the follow up by Reviewer `foEg`, which are largely grounded on on incorrect assumptions about the baselines, treating standard architectural choices as if they were unique to BIA, and falsely giving an oversimplified view of our analysis and metrics. We further clarify these points below.
>
> **On misunderstanding of BIA.**
>
> We first suggest that Reviewer `foEg` fully read the BIA paper as well as our paper, because several claims in the follow up are based on an inaccurate description of BIA.
>
> BIA does not use EMA and does not explicitly regularize early generator-layer features. BIA defines its losses on mid-level surrogate CNN feature maps and uses the generator only to push those benign **surrogate** features away from their benign counterparts, without any consistency constraint on the generator’s internal representations. The claim that both BIA and our method rely on EMA based early-layer **generator** regularization is therefore factually wrong and arises from a misunderstanding of the baseline. We note that a typo in our initial rebuttal mistakenly mentioned EMA in connection with BIA, which may have contributed to this confusion, but this typo does not change what is actually implemented in BIA.
>
> Our use of a ResNet-style generator follows the standard generative-attack protocol established by GAP, CDA, LTP, BIA, GAMA, FACL, and PDCL. This backbone is a common, field-wide choice, not BIA’s core idea. Our contribution is in how this generator is trained: we enforce semantic and structural self-consistency in early generator blocks via a Mean-Teacher-style self-feature-consistency loss on generator intermediates. This generator-side regularization is absent in BIA and leads to consistent improvements across architectures and tasks, as shown in Tables 2,3.
>
> **On ICCV2025Boosting (dSVA).**
>
> We agree with Reviewer `foEg` that dSVA is a relevant recent work. At the time of our submission, this paper was not yet fully published and no official implementation was available, which prevented us from including it as a reproducible baseline.
>
> Methodologically, however, the two works operate in different regimes:
>
> - dSVA primarily innovates on the **surrogate side**, replacing standard CNN surrogates with a dual self-supervised ViT ensemble (DINO+MAE) and carefully selecting ViT facets and layers as supervisory signals
> - Our work intentionally keeps the **CNN-surrogate protocol** of BIA/CDA/GAMA/FACL/PDCL and instead investigates **how the generator itself should be regularized** via early-block semantic anchoring and local structural consistency, with experiments extending to semantic segmentation and object detection
>
> Since dSVA varies the **surrogate** family (self-supervised ViTs) and does not specify how BIA’s mid-level **surrogate** CNN feature divergence is applied on ViTs, a direct numerical comparison under our VGG-16-based protocol is not straightforward. We therefore view the two works as complementary rather than mutually exclusive. Our self-feature consistency guidance is applied inside the **generator** and can, in principle, be combined with dSVA’s ViT-based surrogate ensemble once code is available. We clearly assert that we did not deliberately omit ICCV2025Boosting and that our goal is not to outperform it under its distinct protocol.
>
> **On lack of theoretical insights.**
>
> The claim that our work brings “no theoretical insights” and that the analysis is merely "padding" does not reflect what we actually do.
>
> We introduce a concrete self-feature consistency loss that enforces semantic and structural alignment between benign and adversarial features inside the **generator**, and we analyze its effect on adversarial transferability. Specifically, we measure semantic drift in intermediate generator features via clustering and foreground IoU between benign and adversarial activations, we study how early-block consistency redistributes energy between low and high frequency bands in the perturbations, and we use Grad-CAM on victim models to visually examine how attention shifts for the baselines and our method. These are not generic visualizations added for length; rather, they are designed to explain why early-block semantic anchoring changes the structure of perturbations and improves transfer.
>
> Other reviewers explicitly read the paper in a similar manner. `acvB` finds the self-feature-consistency loss well motivated and mathematically specified, `7rta` emphasizes that the mechanism is grounded in diagnostics showing that early generator features preserve object contours, and `GNru` notes that the semantic drift analysis provides clear empirical motivation. Reducing this to “no insight” overlooks both the mechanism we study and the way it is empirically and analytically supported.

---

> ### Author Response · Authors · 2025-11-28
>
> To avoid misunderstanding and confusion, we again clarify our position in light of Reviewer `foEg`’s last comment.
>
> Reviewer `foEg` characterizes our work as essentially the same as BIA and “incremental,” but our contribution lies in **how the standard generator is trained and analyzed**, which is not covered in BIA. Even though we follow the common generator-based framework used by recent generative attacks for fair comparison, our method introduces **generator-side semantic regularization** (early-block self-feature consistency with an EMA teacher) and provides spatial, block-wise, and spectral analyses of this mechanism. By contrast, BIA and related works primarily innovate on the **surrogate-side loss design** while leaving the generator training unchanged. Our contribution is therefore orthogonal to BIA’s objective and focuses on a different axis of the framework that has not been systematically explored.
>
> The emphasis on a single number and on ICCV2025Boosting also does not reflect the broader pattern of consistent gains across architectures and tasks that we report. Had the reviewer fully read the ICCV2025Boosting paper, they would have also seen that ICCV2025Boosting performs worse than baseline CDA/BIA in VGG-16/Res-50/Den-121, as they operate in a different architecture surrogate (SSL-pretrained ViT ensemble). We believe our work is orthogonal to theirs and that the two methods can in fact be combined; however there is **insufficent detailed information in text for reproducability/no public source code even at this time** for us to perform additional testing with their method. Our defense experiments are also reported as robustness stress tests rather than as the primary optimization target. We will adapt our method to dSVA once code is fully available.
>
> Finally, ACR is not a redundant metric, and we did **not** “admit insufficiency then defend” to pad content. Throughout the paper and our responses, we consistently stated that ACR is introduced as a **complementary measure.** This metric-level distinction matters when comparing attacks across heterogeneous victims and datasets, where two methods can have similar accuracy or FR but very different rates of accidental correction. Our empirical and analytical results use FR and ACR jointly to better understand how **generator-side semantic consistency changes attack behavior**, rather than to inflate contributions or fill space with generic visualizations.
>
> We appreciate the reviewer's efforts in engaging with us to clarify any misunderstandings.

---

> > ### Author Response · Authors · 2025-12-01
> > **Final Author Response (1/3)**
> >
> > [W4] **Additional input processing defense evaluations.**
> >
> > - **Rotation (deg):** Small random rotations sampled from a bounded angle range to preserve semantic content while perturbing pixel-level alignment [A,B]
> >
> > - **Smoothing:** standard spatial smoothing filters (Gaussian, median, and mean) to attenuate small high-freq adversarial perturbations while largely preserving coarse structure, at the cost of some local blurring [A,C]
> >
> > - **Total variation minimization (TVM):** TV-based denoising by approximately minimizing an $\ell_2$ reconstruction loss with a total-variation regularizer, yielding a piecewise-smooth reconstruction that preserves major edges while suppressing small oscillatory perturbations [A,D]
> >
> > - **Pixel deflection (PD):** Randomly selecting a subset of pixels and replacing each with the value of a randomly chosen neighbor within a local window to stochastically disrupt finely tuned adversarial patterns without destroying global semantics [E]
> >
> > Taken together, JPEG-style compression (JPEG), bit-depth reduction (BDR), random resizing–padding (R&P), and smoothing, as well as those in the additional Table below, correspond to the **standard family of input-processing defenses** widely used in prior work on transferable attacks, including our baselines and recent studies such as TransferAttackEval [F] and CGNC [G]. TVM and pixel deflection can be regarded as stronger, yet conceptually similar, input transformations in this family. Evaluating under this common protocol makes our results directly comparable to existing attacks, and across these defenses we observe that adding our generator-side regularizer consistently lowers accuracy and increases FR/ASR relative to each underlying baseline, indicating that **our method *enhances* attack effectiveness against the standard input pre-processing defenses.**
> >
> > ---References---
> >
> > [A] Guo et al., Countering Adversarial Images using Input Transformations, ICLR 2018
> >
> > [B] Xie et al., Mitigating Adversarial Effects Through Randomization, ICLR 2018
> >
> > [C] Xu et al., Feature Squeezing: Detecting Adversarial Examples in Deep Neural Networks, NDSS 2018
> >
> > [D] Rudin et al., Nonlinear Total Variation Based Noise Removal Algorithms, Physica D 1992;
> >
> > [E] Prakash et al., Deflecting Adversarial Attacks With Pixel Deflection, CVPR 2018
> >
> > [F] Zhao, Zhengyu, et al. "Revisiting Transferable Adversarial Images: Systemization, Evaluation, and New Insights." TPAMI 2025.
> >
> > [G] Fang, Hao, et al. "Clip-guided generative networks for transferable targeted adversarial attacks." ECCV 2024.

---

> > > ### Author Response · Authors · 2025-12-01
> > > **Final Author Response (2/3)**
> > >
> > > **Table: Additional input processing defenses.**
> > > |  |  |  | **Random**  | **Rotation**| **(deg)** | | **Smoothing** | **(Kernel)**| | **Total Var. Min.** | **Random Pixel** |
> > > | :- | :- | :- | :- | :- | :- | :- | :- | :- | :- | :- | :- |
> > > |  |  | **Avg** | **30** | **50** | **70** | **90** | **Gaussian** | **Median** | **Mean** | **TVM** | **PD** |
> > > | **BIA** | Acc. (%) $\downarrow$ | 37.08 | 33.59 | 29.20 | 26.27 | 24.92 | 58.75 | 54.27 | 59.06 | 41.17 | 41.62 |
> > > | | ASR (%) $\uparrow$ | 53.95 | 58.38 | 63.86 | 67.52 | 69.26 | 26.66 | 32.44 | 26.15 | 49.05 | 48.48 |
> > > | | FR (%) $\uparrow$ | 59.56 | 63.89 | 68.66 | 71.81 | 73.34 | 34.81 | 40.36 | 34.46 | 55.38 | 54.98 |
> > > | | ACR (%) $\downarrow$ | 8.36 | 7.91 | 6.97 | 6.40 | 6.31 | 11.94 | 11.77 | 11.76 | 9.87 | 9.96 |
> > > | **BIA_Ours** | Acc. (%) $\downarrow$ | 34.05 | 28.61 | 28.43 | 22.27 | 21.18 | 55.55 | 49.61 | 57.20 | 37.42 | 38.15 |
> > > | | ASR (%) $\uparrow$ | 57.75 | 64.52 | 64.79 | 72.44 | 73.86 | 30.84 | 38.44 | 28.63 | 53.61 | 52.77 |
> > > | | FR (%) $\uparrow$ | 63.01 | 69.32 | 69.50 | 76.14 | 77.43 | 38.88 | 45.82 | 36.70 | 59.60 | 58.86 |
> > > | | ACR (%) $\downarrow$ | 7.80 | 6.62 | 6.76 | 5.36 | 5.34 | 12.01 | 11.37 | 11.88 | 8.76 | 9.09 |
> > > | **GAMA** | Acc. (%) $\downarrow$ | 31.66 | 26.10 | 26.25 | 20.18 | 19.10 | 52.47 | 45.53 | 54.05 | 35.08 | 35.78 |
> > > | | ASR (%) $\uparrow$ | 60.67 | 67.59 | 67.48 | 75.03 | 76.46 | 34.61 | 43.45 | 32.55 | 56.41 | 55.52 |
> > > | | FR (%) $\uparrow$ | 65.60 | 71.93 | 71.88 | 78.35 | 79.57 | 42.41 | 50.40 | 40.47 | 62.00 | 61.17 |
> > > | | ACR (%) $\downarrow$ | 7.12 | 5.90 | 6.19 | 4.88 | 4.88 | 11.15 | 10.32 | 11.18 | 7.88 | 7.97 |
> > > | **GAMA_Ours** | Acc. (%) $\downarrow$ | 30.30 | 25.31 | 21.75 | 19.41 | 18.41 | 52.09 | 43.89 | 53.91 | 32.60 | 33.61 |
> > > | | ASR (%) $\uparrow$ | 62.37 | 68.63 | 73.06 | 76.01 | 77.29 | 35.09 | 45.46 | 32.76 | 59.55 | 58.28 |
> > > | | FR (%) $\uparrow$ | 67.08 | 72.92 | 76.65 | 79.26 | 80.33 | 42.71 | 52.09 | 40.62 | 64.77 | 63.66 |
> > > | | ACR (%) $\downarrow$ | 6.86 | 5.93 | 5.16 | 4.77 | 4.65 | 11.09 | 9.80 | 11.25 | 7.48 | 7.68 |
> > > | **FACL** | Acc. (%) $\downarrow$ | 27.12 | 22.00 | 18.64 | 16.68 | 15.84 | 45.91 | 36.29 | 48.38 | 32.40 | 33.11 |
> > > | | ASR (%) $\uparrow$ | 66.41 | 72.75 | 77.08 | 79.49 | 80.59 | 42.91 | 55.01 | 39.73 | 59.88 | 59.00 |
> > > | | FR (%) $\uparrow$ | 70.66 | 76.42 | 80.21 | 82.20 | 83.13 | 49.88 | 60.58 | 46.97 | 64.60 | 64.31 |
> > > | | ACR (%) $\downarrow$ | 6.44 | 5.22 | 4.95 | 4.43 | 4.42 | 10.14 | 8.46 | 10.34 | 7.74 | 7.89 |
> > > | **FACL_Ours** | Acc. (%) $\downarrow$ | 25.30 | 19.68 | 16.91 | 15.20 | 14.61 | 43.08 | 33.43 | 46.18 | 30.56 | 31.32 |
> > > | | ASR (%) $\uparrow$ | 68.69 | 75.78 | 79.19 | 81.35 | 82.13 | 46.35 | 58.54 | 42.50 | 62.15 | 61.28 |
> > > | | FR (%) $\uparrow$ | 72.68 | 79.02 | 81.99 | 83.83 | 84.49 | 52.95 | 63.78 | 49.47 | 67.02 | 66.33 |
> > > | | ACR (%) $\downarrow$ | 6.08 | 5.15 | 4.47 | 4.15 | 4.16 | 9.29 | 7.77 | 9.97 | 7.25 | 7.66 |
> > > | **PDCL** | Acc. (%) $\downarrow$ | 31.34|26.04|22.50|21.84|19.02|52.68|45.87|56.05|32.97|34.40|
> > > | | ASR (%) $\uparrow$|60.87|67.76|72.16|73.00|76.50|31.79|43.04|29.95|59.06|57.78|
> > > | | FR (%) $\uparrow$|65.75|72.10|75.91|76.60|79.65|39.75|50.02|38.04|64.29|63.17|
> > > | | ACR (%) $\downarrow$|7.09|6.20|5.39|5.35|4.67|11.42|10.41|11.30|7.46|7.87|
> > > | **PDCL_Ours** | Acc. (%) $\downarrow$|30.80|25.20|21.84|19.58|18.63|53.77|45.34|55.29|32.64|33.71|
> > > | | ASR (%) $\uparrow$|61.76|68.80|73.00|75.85|77.04|32.87|43.68|30.98|59.54|58.26|
> > > | | FR (%) $\uparrow$ | 66.54|73.03|76.60|79.06|80.16|40.66|50.63|38.92|64.34|63.74|
> > > | | ACR (%) $\downarrow$|7.02|6.00|5.35|4.97|4.80|11.03|10.22|11.39|7.63|8.06|

---

> > > > ### Author Response · Authors · 2025-12-01
> > > > **Final Author Response (3/3)**
> > > >
> > > > [W6] **Additional experiments on targeted attacks.**
> > > >
> > > > Due to the limited rebuttal timeframe, we conducted the state-of-the-art targeted attack experiments (M3D [A] and CGNC [B]) only for a single epoch on ImageNet using the same VGG family as the surrogate. Even under this constraint, our method yields substantial gains, supporting the effectiveness of our **generator-internal semantic consistency**. Higher TSR indicates a stronger targeted attack. We report the average over the targeted classes \{24,99,245,344,471,555,661,701,802,919\} for M3D and \{150, 426, 843, 715, 952, 507, 590, 62\} (normal mode) for CGNC, with $\epsilon$=16/255 as default.
> > > >
> > > > **Table 1: Target success rate (TSR, %) for CLIP-guided CGNC attack (surrogate: VGG19).**
> > > >
> > > > | Method   | VGG16 | GoogLeNet | Inc-v3 | Res152 | Dense121 | Inc-v4 | IncRes-v2 |  Avg   |
> > > > |:--------|:-----:|:---------:|:------:|:------:|:--------:|:------:|:---------:|:------:|
> > > > | CGNC    | 14.71 |   2.03    |  2.77  |  2.68  |  8.31    |  2.41  |   0.96    |  4.84  |
> > > > | **w/ Ours** | 47.50 |   7.90    | 10.96  | 12.24  | 31.29    | 13.36  |   3.98    | **18.18** |
> > > >
> > > > **Key observations:**
> > > > - **Large average gain.** Our method improves the average TSR from **4.84% → 18.18%**, a **3.7× relative increase**, despite the short training.
> > > > - **Consistent gains across victim architectures.** Every victim model benefits, with particularly strong improvements on harder transfers such as **DenseNet121** (+22.98%) and **Inception-v4** (+10.95%), indicating that generator-internal semantic consistency remains effective even under CLIP guidance.
> > > >
> > > > **Table 2: Target success rate (TSR, %) for M3D attack (surrogate: VGG19).**
> > > >
> > > > | Victim   | Method    |  24  |  99  | 245  | 344  | 471  | 555  | 661  | 701  | 802  | 919  |  Avg  |
> > > > |:---------|:----------|:----:|:----:|:----:|:----:|:----:|:----:|:----:|:----:|:----:|:----:|:-----:|
> > > > | Dense121 | M3D       | 35.7 | 65.9 | 37.1 | 59.3 | 43.5 | 45.0 | 23.1 | 20.9 | 30.9 | 11.2 | 37.3  |
> > > > |          | **w/ Ours** | 53.6 | 70.9 | 33.3 | 72.2 | 50.0 | 39.7 | 30.2 | 68.4 | 43.7 | 56.9 | **51.9** |
> > > > | ResNet50 | M3D       | 30.4 | 63.3 | 24.2 | 50.6 | 45.5 | 53.0 | 18.5 | 33.0 | 27.9 | 39.7 | 38.6  |
> > > > |          | **w/ Ours** | 50.7 | 72.3 | 39.1 | 71.3 | 53.6 | 37.0 | 27.0 | 81.3 | 42.3 | 60.5 | **53.5** |
> > > > | ResNet152| M3D       | 16.9 | 48.2 | 21.1 | 42.7 | 22.7 | 33.1 | 10.6 | 29.5 | 20.7 | 21.1 | 26.7  |
> > > > |          | **w/ Ours** | 30.4 | 39.9 | 28.7 | 51.7 | 29.1 | 21.2 | 27.0 | 59.3 | 29.1 | 54.1 | **37.1** |
> > > > | WRN-50-2 | M3D       | 34.6 | 40.8 | 26.5 | 33.6 | 28.7 | 31.1 | 13.1 | 16.9 | 18.2 | 18.7 | 26.2  |
> > > > |          | **w/ Ours** | 50.0 | 51.1 | 27.4 | 52.4 | 34.7 | 17.8 | 20.3 | 68.2 | 28.4 | 52.3 | **40.3** |
> > > >
> > > > **Key observations:**
> > > > - **Consistent gains on a strong targeted attack.** The overall average TSR across all four victims increases from **24.64% → 31.80%**, with absolute gains of **+14.6%** (DenseNet121), **+14.9%** (ResNet50), **+10.4%** (ResNet152), and **+14.1%** (WRN-50-2).
> > > > - **Across-target robustness.** Improvements are observed across a randomized set of 10 target classes, not just a few “easy” ones. For example, on **ResNet50**, class 701 improves from **33.0% → 81.3%**, illustrating that the generator-centric regularization effectively steers features toward diverse target semantics even under a tight training schedule.
> > > >
> > > > ---
> > > >
> > > > References
> > > >
> > > > [A] Zhao, Anqi, et al. “Minimizing maximum model discrepancy for transferable black box targeted attacks.” CVPR 2023. (M3D)
> > > >
> > > > [B] Fang, Hao, et al. “Clip guided generative networks for transferable targeted adversarial attacks.” ECCV 2024. (CGNC)

---

### Official Review · Reviewer_7rta · 2025-10-31

**Soundness:** 3
**Presentation:** 3
**Contribution:** 3
**Rating:** 6
**Confidence:** 4

**Summary:**

The paper proposes Semantic-Consistency guided Generative Attack (SCGA ) for transfer-based black-box image attacks. Instead of only optimizing a surrogate loss (e.g., cross-entropy / feature divergence) on a perturbation generator, the authors align early generator features to an EMA “mean-teacher” copy to preserve object-aligned structure while crafting perturbations. They also introduce Accidental Correction Rate (ACR) to quantify cases where an attack fixes a model’s original mistake—arguably a blind spot of common metrics like FR/ASR/Accuracy. Empirically, the method plugs into several generative baselines and improves transfer across architectures (CNN/ViT/Mixer/Mamba), domains, and tasks (classification, segmentation and detection) without test-time overhead.

**Strengths:**

1.	Simple, orthogonal mechanism: The EMA teacher + early-block consistency integrates into several strong generative baselines without test-time overhead; the approach is easy to adopt.
2.	Clear, generator-internal motivation: The diagnostic showing that early generator features retain object contours while later ones blur them is compelling and grounded in measurable variability (foreground-IoU std across blocks).
3.	Broad evaluation: Cross-model results span CNN, ViT, Mixer, and SSM/Mamba families, with consistent average gains; the method also improves cross-domain (CUB, Cars, FGVC Aircraft) and cross-task transfer to SS/OD models.
4.	Robust-model/defense stress tests. The method improves over baseline against adversarially trained models and input preprocessing defenses (JPEG, bit-depth reduction, randomization & padding).

**Weaknesses:**

1.	Scope of gains & negative deltas. While averages improve, several cells in Table 2 are near-zero or negative (notably for certain transformer targets and PDCL).
2.	Frequency analysis lacks operational detail. The spectral-energy analysis is interesting but currently underspecified. Precisely define the transform (e.g., 2-D FFT with magnitude spectrum), the radial banding scheme (cutoffs in normalized frequency), and whether energies are computed on perturbations or activations. Provide explicit formulas (e.g., radial masks in the Fourier plane) and thresholds so others can reproduce the plots.
3.	Compute disclosure is incomplete. Training doubles forward passes (student+teacher), but only forward overhead is reported. Please report end-to-end training wall-clock, backward cost, peak memory, GPU/TPU model & count, and batch/step counts.
4.	Interplay with CLIP-driven baselines: The authors themselves note only marginal improvements when stacking on PDCL-style CLIP objectives, and briefly speculate that optimizing in CLIP's high-dimensional space may "override or dilute" the structural consistency enforced by SCGA. This explanation is underdeveloped and warrants a deeper investigation.
5.	Insufficient Ablation: While the ablation study in Table 5 is useful, it is incomplete. It demonstrates that applying the consistency loss to early blocks is optimal and that all proposed components contribute positively. However, it fails to fully disentangle the benefits of the EMA-updated teacher from the consistency loss itself. The reported gain from "MT" could stem from the general smoothing effect of weight averaging, or it could be that the consistency loss is only effective when provided with a stable teacher target. A crucial missing experiment would be to apply $\mathcal{L}_{cons.}$ without the EMA teacher (e.g., by using a frozen copy of the student from a previous iteration as the target). This would isolate the unique contribution of the temporal ensembling.
6.	No hyperparameter sensitivity. The paper introduces at least two critical hyperparameters, the EMA smoothing coefficient $\eta$ in Eq. 2 and the consistency loss weight $\lambda_{cons.}$ in Eq. 5. There are no analysis of the method's sensitivity to their values.

**Questions:**

1.	How sensitive are results to η (EMA smoothing), τ (similarity threshold), and the choice/number of early blocks? Any principled way to select them across generator backbones?
2.	Since training doubles generator passes (student+teacher), could you provide full wall-clock with backward, peak memory, accelerator type/count, batch size, and total steps/epochs—along with a forward vs. backward breakdown?
3.	Could you add side-by-side feature/attribution visualizations against the baseline (the Figure-3.1 setting) to substantiate “object-aligned” perturbations ?
4.	When SCGA is combined with PDCL/FACL, what specifically conflicts—frequency bands, spatial regions, or representation space mismatch (CLIP vs. surrogate)? Any mitigation (e.g., decoupled schedules)?
5.	What exact transform and banding do you use (e.g., 2-D FFT magnitude, radial frequency masks, normalized cutoffs), and are energies computed on perturbations or activations (per-channel or aggregated)? Please include explicit formulas and thresholds to ensure reproducibility.
6.	To disentangle the effect of the EMA teacher from $\mathcal{L}_{cons.}$, could you add a variant that applies the consistency loss without EMA (e.g., a frozen or lagged snapshot of the student as the target)? This would clarify the unique contribution of temporal ensembling.

---

> ### Author Response · Authors · 2025-11-21
> **Author Response to Comments (1/5)**
>
> We thank Reviewer `7rta` for highlighting that our method is **easy to adopt, has clear generator-internal motivation, and covers broad evaluation including robust model/defense stress tests.** In response to your constructive comments, we provide clarifications on weaknesses [W] and questions [Q] below.
>
> >[W1] Scope of gains and negative deltas.
>
> We thank the reviewer for highlighting the importance of examining beyond the averages and individual entries in Table 2. We acknowledge that while SCGA generally improves transferability across models, domains, and tasks, several cases slightly underperform the baseline, particularly for some Transformer-based victims and when combined with advanced methods such as PDCL. These instances arise when the underlying baseline is already very strong and highly tailored to the victim or task, leaving limited room for further improvement.
>
> More concretely, there are two interacting factors. First, SCGA employs a ResNet-based generator whose early blocks are regularized to preserve the local structure. In contrast, some Transformer victims rely more heavily on long-range global representations that can deviate from the learned inductive bias. In these cases, early-block feature consistency can bias the perturbation toward shared, locally structured features and slightly reduce its ability to exploit very model-specific global patterns that the baseline already captures well. Second, when SCGA builds upon a foundation model (e.g., CLIP)-guided attacks such as PDCL, the optimization becomes multi-objective. PDCL’s CLIP-space objective encourages global semantic shifts, while SCGA encourages more locally structure-preserving shifts through the generator. On models like PDCL where it is already high-performing, our regularizer can only modestly aid in PDCL’s most aggressive directions, often leading to slightly negative deltas on individual victims, even as the average transfer across architectures and, more notably, cross-task performance (e.g., detection and segmentation) improves. Please also refer to [W4] below for further understanding of the interplay. In the revision, we will explicitly discuss these trade-offs around Table 2 to show that these negative deltas are small, not systematic, and often occur primarily on certain strong baselines.
>
> > [W2] Operational details on spectral analysis.
>
> We appreciate the reviewer’s keen insight for a more precise specification of the spectral analysis. While we abbreviated the procedure in the main text due to space limitations and deferred to Supplementary (Eq. S1), we will briefly outline it here for clear understanding. We compute and aggregate the spectral energies on the intermediate generator feature maps by first applying a 2D Fast Fourier Transform (FFT) to each channel and taking the magnitude spectrum. We apply a standard frequency shift to center the DC component. We then define low-frequency masks with a cutoff radius ($\rho$  = 0.2) in normalized distance, with the rest of the regions beyond the radius being the high-frequency counterpart. Band-wise energies are then computed as power spectra, and the ratio is calculated with respect to total power.
>
> > [W3] Complete disclosure of compute.
>
> We thank the reviewer for pointing out the importance of disclosing compute. SCGA does require an additional forward pass through the teacher during training, while the backward operations remain unchanged because gradients are taken only with respect to the student network. While we included forward pass times in the Supplementary Materials, we further optimized the pipeline and characterize detailed compute as suggested. In the table below, we include end-to-end training wall-clock time for the training, an estimate of the backward cost and total FLOPs per iteration, peak GPU memory usage, and hardware/training specifications. We will revise Sec. E3 of the Supplementary Materials to include this table to give a clear picture of the computational cost.
>
> **Table: Total compute.**
>
> | Method | Train time (hh:mm) | Peak Memory (MB) | GPU type | Train batch size |
> | :--- | :--- | :--- | :--- | :--- |
> | Baseline | 5:00 | 1,384.62 | NVIDIA RTX A6000 (1x) | 48 |
> | Ours | 5:40 | 1,442.23 | | |
>
> **Table: 10k-step average per-train-step compute with a batch size of 1.**
>
> | Method | Student forward (ms) | Teacher forward (ms) | Backward (ms) | Total (ms) | Backward cost (GFLOPs) | Backward CUDA time (ms) |
> | :--- | :--- | :--- | :--- | :--- | :--- | :--- |
> | Baseline | 7.1 | - | 25.6 | 32.7 | 0.0012 | 19.714 |
> | Ours | 6.8 | 3.9 | 28.4 | 39.1 | 0.0044 | 20.192 |

---

> ### Author Response · Authors · 2025-11-21
> **Author Response to Comments (2/5)**
>
> > [W4] Interplay with CLIP-driven baselines.
>
> Early-block semantic anchoring enforces local structural consistency by keeping early generator features for adversarial examples close to those of the benign input. This preserves coarse object layout and boundaries and leaves later blocks to inject adversarial detail. Table 5 shows that applying this consistency in early blocks yields stable transfer gains.
>
> Compared to the primary baseline BIA (Zhang et al., 2022b), SCGA systematically shifts perturbation energy from higher toward lower frequencies. In Table 6 this appears as increased low-band energy and reduced high-band energy. Figure S11 (right) additionally provides a complementary image-space view of the generator intermediate features after the `resblock3` block: low-band reconstructions with SCGA contain brighter, more compact regions than their baselines, indicating that more feature variance is carried by smoothly varying, object-aligned components. High-band reconstructions with SCGA are less peaky and more spatially dispersed, which matches the reduced and less concentrated high-frequency energy in Table 6 and suggests attenuation of surrogate-specific artifacts. PDCL is a mild exception, where the remaining extreme high-frequency content becomes slightly more localized, consistent with the modest gains of PDCL+SCGA in classification transfer.
>
> This mechanism is not in conflict with the CLIP-driven objective of PDCL. PDCL sets the global semantic direction in CLIP space, while SCGA regularizes how this change is realized inside the generator by enforcing structurally consistent early features. The benefit of this structural regularization is more pronounced in localization-oriented settings, where spatially precise corruption of object and boundary features is critical. In Table S7 (cross-task), PDCL with SCGA achieves notably larger gains in cross-task transfer than the other combinations, indicating that our local structural consistency provides a complementary refinement of global feature separation. Please see L1009-1068 in the revision (Supp.)
>
> > [W5] Ablation of EMA vs. consistency.
>
> We thank the reviewer for their careful reading of the ablation in Table 5 and acknowledge that it does not yet disentangle the effect of MT from the effect of  $\mathcal{L}_{\text{cons.}}$ itself. As noted, the observed gain for MT could in principle arise either from the general smoothing effect of weight averaging or the fact that the consistency loss becomes more effective when the target is a temporally stabilized teacher.
>
> To decouple these factors, we report below, as suggested, additional ablations in which the consistency loss is defined against a non-EMA teacher composed as a frozen copy of the student at each previous iteration, while keeping all other settings identical. Comparing this to both the pure EMA-only baselin and the full MT + $\lambda\mathcal{L}_{\text{cons.}}$ configuration better isolates our contributions of temporal ensembling and feature consistency terms. We will revise Table 5 accordingly.
>
> **Table: Missing ablations.**
> | Hyperparameter | Domain (Acc.) | Model (Acc.) | Task (SS; mIoU) | Task (OD; mAP50) |
> | :--- | :--- | :--- | :--- | :--- |
> | Baseline w/ Ours | 47.10 | 44.13 | 23.40 | 24.52 |
> | Baseline w/ Ours w/o EMA (plain copy) | 49.57 | 45.11 | 24.83 | 24.55 |
> | Baseline w/ Ours w/o EMA or $\mathcal{L}_{\text{cons.}}$ | 51.63 | 46.49 | 23.73 | 24.41 |

---

> ### Author Response · Authors · 2025-11-21
> **Author Response to Comments (3/5)**
>
> > [W6] Hyperparameter sensitivity.
>
> We agree that sensitivity to key hyperparameters should be analyzed more thoroughly. Our method introduces at least two critical hyperparameters: the EMA smoothing coefficient in Eq. (2) and the weight of the consistency loss in Eq. (5). In the current version, we tuned these based on validation performance and used the same values across experiments, but did not report their effect explicitly.
>
> In the revision, we will add a hyperparameter study in the supplementary, where we vary the EMA coefficient \(\eta\) and the consistency weight $\lambda_{\text{cons}}$, and report the cross-setting transfer performance below. Here, we observe that there exists a trade-off between optimizing classification and cross-task scores for both hyperparameters, since no single combination outperforms the rest. However, in both, maintaining high values yields higher performance, indicating that each module sufficiently contributes to the overall self-consistency mechanism. Thus, we selected configurations as $\lambda_{\text{cons.}}$ = 0.7, $\eta$ = 0.999) for the best performance evenly across all cross-setting scenarios.
>
> **Table: Hyperparameter sensitivity.** (Domain (Acc.) / Model (Acc.) / Task (SS; mIoU) / Task (OD; mAP50)
> | | 0.1 | 0.3 | 0.5 | 0.7 | 0.9 |
> | :--- | :--- | :--- | :--- | :--- | :--- |
> | $\lambda_{\text{cons.}}$ | 40.55 / 44.84 / 22.59 / 23.96 | 48.29 / 44.80 / 23.63 / 24.36 | 48.49 / 44.68 / 23.82 / 24.78 | 47.10 / 44.13 / 23.40 / 24.52 | 50.08 / 45.89 / 22.79 / 24.19 |
>
> | | 0.9 | 0.95 | 0.98 | 0.99 | 0.995 | 0.999 |
> | :--- | :--- | :--- | :--- | :--- | :--- | :--- |
> | $\eta$ | 48.72 / 45.04 / 24.89 / 24.83 | 50.47 / 45.56 / 23.92 / 24.51 | 51.17 / 45.84 / 24.23 / 24.73 | 47.97 / 44.79 / 23.39 / 24.26 | 48.09 / 44.70 / 24.34 / 24.48 | 47.10 / 44.13 / 23.40 / 24.52 |
>
>
> > [Q1] Hyperparameter sensitivity.
>
> We appreciate the questions on sensitivity to the EMA smoothing coefficient $\eta$, similarity threshold $\tau$, and **the choice/number of early blocks**. Please refer to [W6] for sensitivity to $\eta$ and $\lambda_{\text{cons}}$.
>
> We define “early”, “mid”, and “late” stages of the generator by grouping consecutive residual blocks into pairs, based on the observation that perturbations undergo the most noticeable qualitative changes over every two blocks. As shown by the intermediate feature visualizations in Fig. 2 , the first two blocks (rows (a)-(b)) still closely track the benign image: coarse object shape, foreground–background separation, and large-scale texture are clearly preserved. The next two blocks (rows (c)-(d)) begin to introduce more pronounced distortions and fine-grained variations, while the final two blocks (rows (e)-(f)) predominantly add high-frequency details and noise-like patterns that are no longer easily interpretable as object-level structure. This makes the first two blocks a natural choice for enforcing semantic consistency: they are structurally well-formed and still dominantly encode benign scene semantics, before most of the perturbation mass emerges in later stages.
>
> Table below quantitatively supports this design. Applying the temporal self-consistency loss only to block 1 or only to block 2 yields some benefits, but using both early blocks jointly (*1 & 2 (Ours)*) provides a better balance across *domain*, *cross-model*, and *downstream task metrics* (semantic segmentation and object detection). This pattern is consistent with our intuition: anchoring both early blocks preserves coarse semantics at the onset of perturbation generation, which in turn biases later blocks to place perturbations along these pronounced near-object regions rather than injecting unconstrained noise. As a consequence, the resulting perturbations align more closely with shared, object-level structure across architectures and datasets, enhancing model- and data-agnostic black-box transfer.
>
> **Table: Selection of Early Blocks.**
>
> | Early Block | Domain (Acc.) | Model (Acc.) | Task (SS; mIoU) | Task (OD; mAP50) |
> | :--- | :--- | :--- | :--- | :--- |
> | 1 | 49.13 | 47.62 | 23.78 | 24.47 |
> | 2 | 49.88 | 48.82 | 22.57 | 24.15 |
> | 1 & 2 (Ours) | 47.10 | 44.13 | 23.40 | 24.52 |
>
> > [Q2] Complete compute specifications.
>
> We respectfully refer the reviewer to [W3] above.

---

> ### Author Response · Authors · 2025-11-21
> **Author Response to Comments (4/5)**
>
> > [Q3] Side-by-side visualizations against the baseline.
>
> We appreciate the suggestion to more directly substantiate the “object-aligned” behavior of SCGA visually alongside the baseline. In the main paper, Fig. 3 shows the Grad-CAM and perturbation patterns, but not explicit side-by-side comparisons with the underlying baselines on identical inputs. As suggested, we thus added side-by-side visualizations for baseline and w/ Ours in Fig. S12 (in the revision) in the same setting as Fig. 3 (1). In the figure, while the two methods drive the classifier to predict a similar “incorrect” class, the crafted perturbations by Ours are more vividly concentrated on the foreground object than the baseline, substantiating our object-aware approach within the generator. The prominence of noise around the object rather than in the irrelevant regions is consistent with higher adversarial transferability.
>
> > [Q4] Conflicts in the representation.
>
> We thank the reviewer for this careful question. When SCGA is combined with CLIP-driven attacks such as PDCL and FACL, the objective becomes genuinely multi-term: the CLIP loss pushes images along a strong global semantic direction in CLIP space, while SCGA constrains *how* this change is implemented inside the CNN generator by enforcing early-block structural consistency and inducing a spectral bias toward object-attached low and mid frequencies.
>
> To assess spectral energies aligned with Table 5 of the main paper where low-frequency band energy ratio increases and high-band energy tends to decrease, we strictly decompose the frequency components into low (with a cutoff radius $\rho_{\text{low}} = 0.2$) and high (with $\rho_{\text{high}} = 0.8$). By performing the same operations (2D-FFT, FFT-shift, cutoff by normalized distance from the center, and masking into respective low and high band energies), we obtain the patterns shown in Fig. S11 (in the revision).
>
> **Low band.** In the figure, we observe a consistent spectral pattern across all baselines in low band. With Ours added, the low-band maps become less dominated at the center, yet the low-frequency energy is heavily concentrated in the most critical low-frequency components. Although it appears to be dispersing the low-frequency energy, we highlight that the dispersed energy in each baseline is now *highly concentrated* in key low-frequency components that critically drive transferability. This means that these components depict the coarse object structures, which are characteristic of model-shared features, a key driving factor for adversarial transferability.
>
> **High band.** Similarly, examining the high-frequency components, we observe that the highly concentrated high-frequency energy in each baseline is somewhat dispersed as a blurry cloud-looking spectrum when ours is added. That is, high-frequency energy is rather *suppressed*, thereby redistributing this energy onto the low band. However, we observe one exception with PDCL. While the low-band spectrum exhibits a similar trend as the others, in the high band, PDCL with Ours fails to effectively suppress the high-band energy, leading to potential drop in transfer performance. Nevertheless, the concentration of low-band energy is higher than that of high-band energy, which in turn overcomes the drop induced by the concentrated high-band energy. This yields modest improvement in the average performance we observe in Table 2 of the main paper for PDCL add-on.
>
> Spatially, the side-by-side feature maps in Fig. 2 of the main paper show that SCGA consistently strengthens object-aligned structure in the generator features. On some Transformer victims, however, PDCL and FACL alone already induce highly discriminative global shifts, so the additional local structural bias can slightly trade off model-specific advantages, leading to near-zero or mildly negative deltas for those models while still improving averages and cross-task performance. We do not observe a single hard conflict confined to one specific band or region. Rather, the tension arises from combining a global CLIP-space objective that favors large, model-tailored semantic shifts with a generator-side regularizer that prefers structured, low and mid frequency perturbations aligned with CNN-like inductive biases. As a mitigation, we plan to explore decoupled schedules (first optimizing PDCL or FACL, then gradually ramping in SCGA) and reduced SCGA weights when CLIP losses are active, and we will discuss this interaction and these strategies explicitly around Table 2 in the revision.
>
> > [Q5] Spectral decomposition formula.
>
> We appreciate the request for explicit formulas. All spectral measurements are computed on the generator intermediate feature maps to explicitly analyze the internals from the frequency perspective. We refer to [W2] and Supplementary Materials Eq. (S1). We will reference these details in the **"Spectral energy comparisons"** of the main paper accordingly.

---

> ### Author Response · Authors · 2025-11-21
> **Author Response to Comments (5/5)**
>
> > [Q6] Further ablations on no-EMA case.
>
> We agree that disentangling the effect of the EMA teacher from the consistency loss is important to understand the role of temporal ensembling. We respectfully refer the reviewer to [W4] for details.

---

### Official Review · Reviewer_acvB · 2025-10-31

**Soundness:** 3
**Presentation:** 2
**Contribution:** 3
**Rating:** 6
**Confidence:** 3

**Summary:**

This paper presents a new augmentation for generative adversarial attacks aimed at improving their transferability in black-box settings. The core idea is to enforce semantic consistency in the generator's early blocks using a EMA framework with a self-feature consistency loss, thereby stabilizing object-aligned intermediate representations. Then the authors  introduce a new evaluation metric called ACR to detect unintended benign corrections, aiming to provide a more comprehensive view of attack reliability beyond conventional metrics. Through extensive quantitative and qualitative evaluations, including ablation studies and spectral analysis, they demonstrate systematic improvements across diverse architectures, domains, and tasks.

**Strengths:**

1. The self-feature consistency loss is well-motivated and mathematically specified.

2.  The experiments are comprehensive, including multiple model architectures,  cross-domain and cross-task scenarios and fine-grained ablation.

3. The finding of  the semantic drift across generator layers that degrades black-box transferability is novel.

**Weaknesses:**

1. The baseline methods should be introduced before presenting the experimental results, as omitting this order significantly reduces readability.

2. Although the method claims to be architecture-agnostic, there is room for a stronger demonstration across more varied generator or victim types (e.g., diffusion)

**Questions:**

1. Why does the work only consider untargeted attacks as baselines? It seems that targeted attack methods (e.g., [1]) could also be included for a more comprehensive comparison.

2. How Spectral energy by band is defined and calculated in  Table 6?

3. How would the method compare against the strongest black-box transfer attacks that do not rely on generator-based pipelines (e.g., [2]) ?



[1] Fang, Hao, et al. "Clip-guided generative networks for transferable targeted adversarial attacks." European Conference on Computer Vision. Cham: Springer Nature Switzerland, 2024.

[2] Wang, Xiaosen, et al. "Admix: Enhancing the transferability of adversarial attacks." Proceedings of the IEEE/CVF international conference on computer vision. 2021.

---

> ### Author Response · Authors · 2025-11-21
> **Author Response to Comments (1/2)**
>
> We thank Reviewer `acvB` for acknowledging that our proposed method is **well-motivated, mathematically specified,
> comprehensive, and novel.**
> In response to your constructive comments, we provide clarifications on weaknesses [W] and questions [Q] below.
>
> > [W1] Earlier presentation of baseline methods for better readability.
>
> We thank the reviewer for pointing out the issue with the order of baseline descriptions and experimental
> results. In the submitted version, some baselines are first referenced in the results tables without explicitly referring to them by their names, potentially confusing the reader. In the revision, we will reorganize the references so
> that all baseline methods are explicitly described *before* any quantitative comparisons are presented to improve
> readability and understanding of our method.
>
> >[W2] Stronger demo of architecture-agnostic claim.
>
> We appreciate the reviewer’s request for a stronger demonstration of architecture-agnostic behavior. Our claim
> of **architecture-agnosticism is intended primarily with respect to the *victim models***, especially in our black-box setting: our proposed method is
> designed for black-box transfer, and our evaluation already includes diverse victim architectures and training
> distributions to which the attacks transfer without accessing their internal features.
> On the generator side, we agree that showing robustness across different generator families is important. In the
> main paper, we focus on a ResNet-style generator for clarity, efficiency, and fairness in comparison. In Table
> S1 of the Supplementary, we also report results using a U-Net-based generator, which is, in fact, the canonical
> backbone architecture employed in diffusion models in general. Our semantic consistency loss yields consistent
> improvements in this U-Net setting as well, suggesting that the proposed mechanism is not tailored to a specific
> generator architecture.
>
> We also acknowledge the reviewer’s suggestion to explore fully-fledged diffusion-based generators as adversaries.
> While these models are *considerably heavier and more resource-intensive* to train and deploy in a black-box
> setting (which runs counter to our goal of practical and efficient transfer attacks), extending our framework to
> diffusion-style generators is an interesting direction for future work. We will clarify the intended scope of our
> “architecture-agnostic” claim (victim-side, with supporting evidence of generator-side robustness via U-Net) and
> discuss diffusion-based extensions explicitly in the revised version.

---

> ### Author Response · Authors · 2025-11-21
> **Author Response to Comments (2/2)**
>
> >[Q1] Broadened scope to targeted attacks.
>
> We thank the reviewer for this suggestion. Our main focus in this work is on *untargeted* black-box transfer,
> since this setting naturally extends across models, domains, and downstream tasks (e.g., detection, segmentation),
> while most targeted attacks (including [1]) are typically formulated and evaluated primarily on image classification.
> That said, the proposed semantic consistency regularizer is orthogonal to the attack objective and can also be
> applied to targeted transfer attacks. We will follow up with additional results on targeted attacks.
>
> >[Q2] Spectral energy definition in Table 6.
>
> We appreciate the reviewer’s interest in our spectral analysis. Due to space limitations, the detailed definition
> was deferred to Eq. S1 in the Supplementary, but we briefly summarize it here. For an intermediate feature map at layer $\ell$,
> denoted $\mathcal{F}_{\ell}$, we compute channel-wise 2D FFT and take the squared norm to obtain the power spectrum. We then normalize spatial frequencies to a radial frequency $\rho(u, v)$ ∈ [0, 1] and partition the spectrum into bands, using a cutoff at $\rho$ = 0.2 to separate low-band frequency component by masking. We designate high-band frequencies as
> the region beyond the cutoff (for direct low/high band-wise energy ratio comparison, thus totaling 1, which **allows to measure the spectral energy shift**). The spectral energy of the low/high band is defined as the proportion of total
> energy contained in that band, thus the two sum up to one. Table 6 of the main paper reports these band-wise
> energy proportions, averaged over all $\ell$ intermediate layers, which allows us to characterize how different attacks
> allocate their perturbation energy across frequency bands.
>
> >[Q3] Comparison against the strongest black-box attacks without generator.
>
> We agree that comparing against strong non-generator-based black-box transfer attacks is important for
> positioning our method. Our main experiments focus on generator-based attacks, as our contribution is a
> generator-side semantic regularizer designed to enhance such pipelines. Nevertheless, following the reviewer’s
> suggestion, we additionally compared our method against a strong non-generative transfer attack, Admix [2]
> (ϵ = 10/255, step size=4/255, num_copies=5, num_iter=10) in the table below. In our experiments, **our generator-based attack with the proposed regularizer achieves consistently higher transfer rates than Admix [2] across multiple domains
> and victim models**, while remaining competitive in terms of computational cost (Admix takes ≈1.2 s/forward for
> each pass vs. ≈10.7 ms). We also note that such white-box attacks are mainly restricted to image classification,
> whereas ours can be easily extended to cross-task settings. We will include these results and a discussion in the
> revision to clarify how our method compares to state-of-the-art non-generative black-box transfer attacks.
>
> **Table: Comparison with non-generator-based white-box attack (Acc↓/ASR↑/FR↑/ACR↓).**
> | Method | Cross-Domain | Cross-Model  |
> | :--- | :--- | :--- |
> | Admix | 87.18 / 5.60 / 8.84 / 15.63 | 51.18 / 32.99 / 38.08 / 5.24 |
> | Ours | 47.10 / 49.02 / 51.66 / 9.66 | 44.13 / 44.02 / 50.66 / 8.32 |

---

> > ### Comment · Reviewer_acvB · 2025-11-23
> >
> > I thank the authors for their efforts in addressing my concerns. I also notice that another reviewer suggested including ICCV2025Boosting as a baseline. Could the authors clarify how that method compares with the proposed approach if adding it to the experimental baselines?
> >
> >
> > [ICCV2025Boosting] Boosting Generative Adversarial Transferability with Self-supervised Vision Transformer Features, ICCV, 2025

---

> ### Author Response · Authors · 2025-11-24
> **Follow-up Author Response to Comment by Reviewer acvB**
>
> **Comparison with dSVA (ICCV 2025).**  We thank Reviewer acvB for pointing out the recent ICCV 2025 work (dSVA) [ICCV2025Boosting]. *This concurrent paper is closely relevant yet largely orthogonal to our focus.* For clarity, we summarize apparent similarities and key distinctions below:
>
> - **Common ground.** Both dSVA and our method adopt a generator-based framework for transferable attacks and use feature-aware objectives rather than pure label-space losses. Both are evaluated in black-box transfer settings across multiple target architectures.
>
> - **Where the novelty lies.** dSVA primarily innovates on the *surrogate side*: it replaces standard CNN surrogates with dual self-supervised ViTs (DINO + MAE) and carefully selects ViT facets and layers to supervise a ResNet generator, showing that CL+MIM features are strong surrogates for classification-oriented transfer. **In contrast**, we deliberately *fix* a conventional CNN-surrogate setup aligned with BIA/CDA/GAMA/FACL/PDCL and instead investigate how the perturbation *generator itself* should be regularized via early-layer semantic anchoring and local structural consistency to improve cross-domain, cross-model, and cross-task transfer (classification, detection, and segmentation).
>
> - **Location of feature guidance.** dSVA’s losses are defined on surrogate ViT representations (facet-level features and self-attention maps from DINO/MAE) computed using *both benign and adversarial inputs*, and its analysis centers on which ViT layers/facets to use as supervisory signals. **In contrast**, our method keeps the surrogate objective identical to prior generative attacks and introduces a feature-consistency loss on the *intermediate generator blocks*, using the *benign input image only* as the driving signal. In other words, dSVA optimizes *which surrogate ViT feature(s)* to attack, whereas we orthogonally optimize *how the generator’s internal features* evolve when crafting the perturbation.
>
> - **Surrogate family and protocol.** dSVA assumes a dual self-supervised ViT surrogate ensemble and omits classical CNN surrogates such as VGG-16, while re-implementing earlier baselines (including BIA) in this ViT-centric regime (though how BIA's mid-level *CNN surrogate* feature divergence is instantiated for *ViT surrogates* that have *different layer structures* is not fully specified in the paper). **In contrast**, our work follows the established generative-attack protocol based on CNN surrogates and their mid-level features, which are standard in the literature and directly comparable across BIA/CDA/GAMA/FACL/PDCL. A strictly fair numerical comparison would therefore require either re-deriving dSVA under our CNN-surrogate setting or, conversely, re-designing our entire pipeline around dual ViT surrogates, which is beyond the scope of this work.
>
> - **Scope of evaluation.** dSVA focuses mainly on image-classification transfer (including robust ImageNet and defense models) under its ViT-based surrogate design. **In contrast**, our experiments emphasize not only cross-domain/model classification, but also cross-task transfer to localization-heavy tasks (semantic segmentation and object detection), where early-block structural consistency is particularly beneficial.
>
> - **Timeline and complementarity.** dSVA is a concurrent ICCV 2025 work that is developed under a different surrogate protocol, where the generator is trained with *a dual self-supervised ViT surrogate ensemble rather than the standard CNN surrogates used in the BIA/CDA/GAMA/FACL/PDCL line that we follow*. In the revision, we will explicitly cite and position dSVA as concurrent work and clarify that its self-supervised ViT surrogate design can serve to complement our generator-side regularization. In principle, SCGA could also be combined with dSVA’s generator in future work for an attack that benefits from synergy between both ViT surrogate ensemble (of dSVA) and early-block semantic anchoring (Ours). Upon release of their code, we would love to examine this synergy in attack performance.

---

> > ### Author Response · Authors · 2025-12-01
> > **Final Author Response**
> >
> > [Q1] **Additional experiments on targeted attacks.**
> >
> > Due to the limited rebuttal timeframe, we conducted the state-of-the-art targeted attack experiments (M3D [A] and CGNC [B]) only for a single epoch on ImageNet using the same VGG family as the surrogate. Even under this constraint, our method yields substantial gains, supporting the effectiveness of our **generator-internal semantic consistency**. Higher TSR indicates a stronger targeted attack. We report the average over the targeted classes \{24,99,245,344,471,555,661,701,802,919\} for M3D and \{150, 426, 843, 715, 952, 507, 590, 62\} (normal mode) for CGNC, with $\epsilon$=16/255 as default.
> >
> > **Table 1: Target success rate (TSR, %) for CLIP-guided CGNC attack (surrogate: VGG19).**
> >
> > | Method   | VGG16 | GoogLeNet | Inc-v3 | Res152 | Dense121 | Inc-v4 | IncRes-v2 |  Avg   |
> > |:--------|:-----:|:---------:|:------:|:------:|:--------:|:------:|:---------:|:------:|
> > | CGNC    | 14.71 |   2.03    |  2.77  |  2.68  |  8.31    |  2.41  |   0.96    |  4.84  |
> > | **w/ Ours** | 47.50 |   7.90    | 10.96  | 12.24  | 31.29    | 13.36  |   3.98    | **18.18** |
> >
> > **Key observations:**
> > - **Large average gain.** Our method improves the average TSR from **4.84% → 18.18%**, a **3.7× relative increase**, despite the short training.
> > - **Consistent gains across victim architectures.** Every victim model benefits, with particularly strong improvements on harder transfers such as **DenseNet121** (+22.98%) and **Inception-v4** (+10.95%), indicating that generator-internal semantic consistency remains effective even under CLIP guidance.
> >
> > **Table 2: Target success rate (TSR, %) for M3D attack (surrogate: VGG19).**
> >
> > | Victim   | Method    |  24  |  99  | 245  | 344  | 471  | 555  | 661  | 701  | 802  | 919  |  Avg  |
> > |:---------|:----------|:----:|:----:|:----:|:----:|:----:|:----:|:----:|:----:|:----:|:----:|:-----:|
> > | Dense121 | M3D       | 35.7 | 65.9 | 37.1 | 59.3 | 43.5 | 45.0 | 23.1 | 20.9 | 30.9 | 11.2 | 37.3  |
> > |          | **w/ Ours** | 53.6 | 70.9 | 33.3 | 72.2 | 50.0 | 39.7 | 30.2 | 68.4 | 43.7 | 56.9 | **51.9** |
> > | ResNet50 | M3D       | 30.4 | 63.3 | 24.2 | 50.6 | 45.5 | 53.0 | 18.5 | 33.0 | 27.9 | 39.7 | 38.6  |
> > |          | **w/ Ours** | 50.7 | 72.3 | 39.1 | 71.3 | 53.6 | 37.0 | 27.0 | 81.3 | 42.3 | 60.5 | **53.5** |
> > | ResNet152| M3D       | 16.9 | 48.2 | 21.1 | 42.7 | 22.7 | 33.1 | 10.6 | 29.5 | 20.7 | 21.1 | 26.7  |
> > |          | **w/ Ours** | 30.4 | 39.9 | 28.7 | 51.7 | 29.1 | 21.2 | 27.0 | 59.3 | 29.1 | 54.1 | **37.1** |
> > | WRN-50-2 | M3D       | 34.6 | 40.8 | 26.5 | 33.6 | 28.7 | 31.1 | 13.1 | 16.9 | 18.2 | 18.7 | 26.2  |
> > |          | **w/ Ours** | 50.0 | 51.1 | 27.4 | 52.4 | 34.7 | 17.8 | 20.3 | 68.2 | 28.4 | 52.3 | **40.3** |
> >
> > **Key observations:**
> > - **Consistent gains on a strong targeted attack.** The overall average TSR across all four victims increases from **24.64% → 31.80%**, with absolute gains of **+14.6%** (DenseNet121), **+14.9%** (ResNet50), **+10.4%** (ResNet152), and **+14.1%** (WRN-50-2).
> > - **Across-target robustness.** Improvements are observed across a randomized set of 10 target classes, not just a few “easy” ones. For example, on **ResNet50**, class 701 improves from **33.0% → 81.3%**, illustrating that the generator-centric regularization effectively steers features toward diverse target semantics even under a tight training schedule.
> >
> > ---
> >
> > References
> >
> > [A] Zhao, Anqi, et al. “Minimizing maximum model discrepancy for transferable black box targeted attacks.” CVPR 2023. (M3D)
> >
> > [B] Fang, Hao, et al. “Clip guided generative networks for transferable targeted adversarial attacks.” ECCV 2024. (CGNC)

---

### Official Review · Reviewer_y2kS · 2025-11-01

**Soundness:** 2
**Presentation:** 3
**Contribution:** 2
**Rating:** 2
**Confidence:** 4

**Summary:**

This paper focuses on the intermediate features within the perturbation generator that are often overlooked in previous generative adversarial attacks.  The authors first note that a stronger attack better preserves the coarse shape from the early layers in the generator. Based on the observation, the authors introduce a lightweight EMA teacher to the early blocks during generator training, which regulates the features to maintain object contours and shapes. Extensive experiments show that the proposed strategy can serve as a plug-and-play technique to improve existing attack methods.

**Strengths:**

1. The paper is well-written and easy to follow.
2. The proposed method is simple and intuitive.
3. The authors provide sufficient experiments across different models and data domains to prove the effectiveness of their method.
4. Experiments about the intermediate block-level analysis are interesting and provide evidence of preserving image contours and shapes.

**Weaknesses:**

1. The designed method is simple and intuitive, which somewhat lacks novelty as feature-level guidance has been widely investigated in various existing studies.
2. As shown in Table 2, the proposed method only brings marginal improvements for powerful attacks such as PDCL, raising concerns about its necessity and effectiveness.
3. Lack experiments against input-processing-based defense methods.
4. In line 69, a missing full stop after "generator intermediate blocks".

**Questions:**

See weaknesses.

---

> ### Author Response · Authors · 2025-11-21
> **Author Response to Comments (1/2)**
>
> We thank Reviewer `y2kS` for a careful review, and appreciate that the reviewer found our paper to be **well-written, easy-to-follow, intuitive, and interesting.** In response to your constructive comments, we respectfully provide clarifications on weaknesses [W] below.
> > [W1] The method lacks novelty as a feature-level guidance work.
>
> **On novelty and relation to feature-level guidance.**
> We appreciate the reviewer’s comment and agree that feature-level guidance has been widely explored. Prior work includes intermediate feature guidance such as ILA (Huang et al., 2019), ILA++ (Li et al., 2020), and ILPD (Li et al., 2023), as discussed in *Supp. Sec.C* (“Distinctions of Our Method”), as well as attacks that exploit deep content or semantic structure, e.g., bounded adversarial attacks on deep content features [1] (Xu et al., CVPR 2022), ColorFool [2] (Shamsabadi et al., CVPR 2020), and feature-space style perturbations [3] (Peng et al., AAAI 2021). **Our approach complements these works and differs in three key aspects.**
>
> - (i) Where the feature guidance is applied.
> > Most existing feature-guided attacks operate directly in the feature
> space of a white-box classifier, using intermediate activations as the primary optimization space and crafting
> perturbations via gradient-based updates on these classifier features. Style-perturbation attacks follow a similar
> paradigm, additionally requiring a trained decoder and pre-defined style prototypes to map perturbed features
> back to pixels, restricting their use as generic plug-ins. In contrast, our method does not optimize in the surrogate
> classifier’s feature space; rather, we apply feature-level regularization within the intermediate blocks of the
> perturbation generator with only benign images as input (as a reference signal). The surrogate is kept frozen and
> used as in existing generator-based transfer attacks, but our semantic consistency loss is applied to the generator
> intermediate features rather than to classifier activations. To our knowledge, prior work has not used internal
> generator representations to provide semantic guidance for black-box transfer.
>
> - (ii) How semantic information is preserved during perturbation generation.
> > Our block-wise visualization in Fig.4 shows how the perturbation emerges with block progression and intuitively motivates enforcing semantic consistency specifically at early generator blocks, where coarse object- and shape-level structures still closely follow the benign input. This is conceptually aligned with works such as ColorFool [2] and content-feature
> attacks [1], which emphasize low-frequency or semantically meaningful changes, but those methods either
> manipulate classifier features in a fully white-box setting or rely on specialized decoders and style prototypes. By
> contrast, we constrain the evolution of internal generator features during perturbation synthesis, using benign
> input features as a semantic anchor without introducing an additional decoder or handcrafted style prototypes,
> which makes our regularizer easily integrable into existing generator-based attacks.
>
> - (iii) Threat model and cross-setting transferability.
> > Many deep content- or style-based feature-guidance methods assume full white-box access and are primarily evaluated on image classification tasks. Current generative-model based attacks are capable of strong black-box transfer. However, their cross-setting efficacy (across architectures and downstream tasks such as detection and segmentation) has been less systematically analyzed. Our method builds on these strong generative attacks and introduces a principled early-block semantic regularization that improves robustness across cross-model and cross-task settings under black-box settings. In this sense, we do not propose another classifier-feature space attack but a generator-centric mechanism that enhances state-of-the-art generative attacks for black-box transfer, in a way that is both modular and empirically validated across diverse
> transfer scenarios.
>
> Altogether, (a) shifting feature guidance from classifier activations to internal perturbation-generator representations, (b) explicitly targeting early-block semantic preservation during perturbation synthesis, and (c) designing
> and evaluating this mechanism in the black-box transfer regime with cross-setting benchmarks certainly go
> beyond a straightforward extension of existing feature-level guidance and constitute a distinct and practically
> meaningful contribution.
>
>
>
> ---References---
>
> [1] Q. Xu, G. Tao, S. Cheng, and X. Zhang, “Towards feature space adversarial attack by style perturbation,” AAAI 2021.
>
> [2] A. S. Shamsabadi, R. Sanchez-Matilla, and A. Cavallaro, “Colorfool: Semantic adversarial colorization,” CVPR 2020.
>
> [3] D. Peng, Z. Zheng, and X. Zhang, “Structure-preserving transformation: Generating diverse and transferable
> adversarial examples,”  AAAI-2019 Workshop AICS.

---

> ### Author Response · Authors · 2025-11-21
> **Author Response to Comments (2/2)**
>
> > [W2] Marginal improvements for the add-on to PDCL.
>
> We appreciate the reviewer’s concern that gains over powerful CLIP-based attacks such as PDCL in Table 2
> appear marginal in some settings. While we acknowledge this modest improvement, our method's intention is not to replace such methods, but to **provide a complementary, generator-centric regularization that can be seamlessly integrated into state-of-the-art generative attacks.** CLIP-based generative methods like PDCL primarily exploit the *global* image-text alignment in the CLIP feature space, and already constitute a very strong baseline in the black-box transfer regime. In contrast, our feature
> consistency loss operates on the *early generator intermediate features*, encouraging the perturbation generator to preserve
> coarse object- and region-level semantics inherited from the benign input. As a result, when combined with
> PDCL, our regularizer complements the CLIP-driven objective: PDCL steers global alignment, while our method
> helps maintain structured local semantics that can be more effectively corrupted downstream.
> This division of roles is reflected in the empirical trends: while improvements on image classification (largely
> driven by global semantics) are modest when added to some strong baselines, we observe **more pronounced
> gains in *localization-heavy* cross-task settings** such as semantic segmentation and object detection, where the
> quality of spatially localized perturbations is critical. Importantly, our method generally yields non-trivial improvements, especially across tasks. We therefore view the proposed semantic regularizer as a necessary and
> effective enhancement to strong generative attacks in the black-box transfer setting, rather than as a competing
> standalone attack. We respectfully refer to the response to Reviewer 7rta's [W4] comments for further understanding of the interplay.
>
> > [W3] Lacks experiments on input-processing-based defenses.
>
> We thank the reviewer for emphasizing the importance of evaluating against input-processing-based defenses.
> In the current submission, we already included experiments with widely used input transformations such as
> JPEG compression, bit-depth reduction, and randomization-based preprocessing, which serve as canonical inputprocessing defenses. Moreover, we evaluate against NRP purification defense in Table S10 in the Supplementary
> Materials. Our results uniformly indicate that our proposed method maintains its advantage even under these
> transformations. We will follow up with additional results on other input processing defenses.
>
> > [W4] Missing a ’period (.)’ in L69.
>
> We appreciate the reviewer’s careful reading. We will add the missing full stop after “generator intermediate
> blocks” in line 69 and perform additional proofreading to correct similar minor typographical issues.

---

> > ### Comment · Reviewer_y2kS · 2025-11-24
> >
> > I appreciate the author's efforts in solving my concerns, which help me better understand the motivation and contribution of this work.
> >
> > For `[W3]`, could the authors supplement experiments on diffusion-based purification defenses (e.g., DiffPure) ?

---

> ### Author Response · Authors · 2025-12-01
> **Final Author Response (1/3)**
>
> We thank the reviewer for suggesting an evaluation with diffusion based purification. Running DiffPure on ImageNet is computationally intensive and the cost grows with the number of samples, so within the rebuttal timeframe we conducted **preliminary experiments on a random subset of 1k validation images**. We plan to extend this study to the full 50k set in a later revision given sufficient time.
>
> Our framework is designed to improve **adversarial transferability on undefended models**, following standard protocols in CDA, LTP, BIA, GAMA, FACL, and PDCL, rather than to construct an adaptive attack specifically tailored to circumvent purification defenses.
>
> **Results and analysis.**  Under DiffPure with $\epsilon$ = 10/255 and $t$ = 150, incorporating our method yields **robust accuracies that remain comparable** to the baselines on both victims. Fluctuations are small in magnitude and are plausibly explained by the stochastic diffusion process and the limited 1k sample size, rather than by a systematic loss of robustness.
>
> **Table: Preliminary robustness evaluations against DiffPure ($\epsilon$ = 10/255, $t$ = 150) on 1k samples (Cls. Acc., %).**
>
> | Victim   | CDA  | w/ Ours | LTP  | w/ Ours | BIA  | w/ Ours | GAMA | w/ Ours | FACL | w/ Ours | PDCL | w/ Ours |
> |:---------|:----:|:-------:|:----:|:-------:|:----:|:-------:|:----:|:-------:|:----:|:-------:|:----:|:-------:|
> | Res152   | 67.7 | 66.3    | 66.1 | 68.0    | 66.1 | 66.8    | 65.8 | 66.6    | 65.4 | 66.2    | 65.3 | 65.2    |
> | Dense121 | 62.0 | 60.5    | 62.9 | 64.0    | 62.1 | 61.4    | 62.0 | 61.6    | 61.8 | 62.1    | 61.5 | 61.4    |
>
> **Key observations.**
> - On **ResNet152**, accuracies with Ours stay close to the baselines, with modest increases (e.g., LTP, BIA, GAMA, FACL) and modest decreases (e.g., CDA), all within a narrow band.
> - On **DenseNet121**, deviations are similarly small, and several pairs (e.g., LTP, FACL) show slight improvements.
> - Overall, these preliminary results indicate that our method remains **largely compatible with strong diffusion based purification**. The semantic consistency enforced during generation improves transfer on standard models without causing a significant loss of robustness under DiffPure compared to the original baselines.

---

> > ### Author Response · Authors · 2025-12-01
> > **Final Author Response (2/3)**
> >
> > [W3] **Additional input processing defense evaluations.**
> >
> > - **Rotation (deg):** Small random rotations sampled from a bounded angle range to preserve semantic content while perturbing pixel-level alignment [A,B]
> >
> > - **Smoothing:** standard spatial smoothing filters (Gaussian, median, and mean) to attenuate small high-freq adversarial perturbations while largely preserving coarse structure, at the cost of some local blurring [A,C]
> >
> > - **Total variation minimization (TVM):** TV-based denoising by approximately minimizing an $\ell_2$ reconstruction loss with a total-variation regularizer, yielding a piecewise-smooth reconstruction that preserves major edges while suppressing small oscillatory perturbations [A,D]
> >
> > - **Pixel deflection (PD):** Randomly selecting a subset of pixels and replacing each with the value of a randomly chosen neighbor within a local window to stochastically disrupt finely tuned adversarial patterns without destroying global semantics [E]
> >
> >
> > Taken together, JPEG-style compression (JPEG), bit-depth reduction (BDR), random resizing–padding (R&P), and smoothing, as well as those in the additional Table below, correspond to the **standard family of input-processing defenses** widely used in prior work on transferable attacks, including our baselines and recent studies such as TransferAttackEval [F] and CGNC [G]. TVM and pixel deflection can be regarded as stronger, yet conceptually similar, input transformations in this family. Evaluating under this common protocol makes our results directly comparable to existing attacks, and across these defenses we observe that adding our generator-side regularizer consistently lowers accuracy and increases FR/ASR relative to each underlying baseline, indicating that **our method *enhances* attack effectiveness against the standard input pre-processing defenses.**
> >
> >
> > ---References---
> >
> > [A] Guo et al., Countering Adversarial Images using Input Transformations, ICLR 2018
> >
> > [B] Xie et al., Mitigating Adversarial Effects Through Randomization, ICLR 2018
> >
> > [C] Xu et al., Feature Squeezing: Detecting Adversarial Examples in Deep Neural Networks, NDSS 2018
> >
> > [D] Rudin et al., Nonlinear Total Variation Based Noise Removal Algorithms, Physica D 1992;
> >
> > [E] Prakash et al., Deflecting Adversarial Attacks With Pixel Deflection, CVPR 2018
> >
> > [F] Zhao, Zhengyu, et al. "Revisiting Transferable Adversarial Images: Systemization, Evaluation, and New Insights." TPAMI 2025.
> >
> > [G] Fang, Hao, et al. "Clip-guided generative networks for transferable targeted adversarial attacks." ECCV 2024.

---

> > > ### Author Response · Authors · 2025-12-01
> > > **Final Author Response (3/3)**
> > >
> > > **Table: Additional input processing defenses.**
> > > |  |  |  | **Random**  | **Rotation**| **(deg)** | | **Smoothing** | **(Kernel)**| | **Total Var. Min.** | **Random Pixel** |
> > > | :- | :- | :- | :- | :- | :- | :- | :- | :- | :- | :- | :- |
> > > |  |  | **Avg** | **30** | **50** | **70** | **90** | **Gaussian** | **Median** | **Mean** | **TVM** | **PD** |
> > > | **BIA** | Acc. (%) $\downarrow$ | 37.08 | 33.59 | 29.20 | 26.27 | 24.92 | 58.75 | 54.27 | 59.06 | 41.17 | 41.62 |
> > > | | ASR (%) $\uparrow$ | 53.95 | 58.38 | 63.86 | 67.52 | 69.26 | 26.66 | 32.44 | 26.15 | 49.05 | 48.48 |
> > > | | FR (%) $\uparrow$ | 59.56 | 63.89 | 68.66 | 71.81 | 73.34 | 34.81 | 40.36 | 34.46 | 55.38 | 54.98 |
> > > | | ACR (%) $\downarrow$ | 8.36 | 7.91 | 6.97 | 6.40 | 6.31 | 11.94 | 11.77 | 11.76 | 9.87 | 9.96 |
> > > | **BIA_Ours** | Acc. (%) $\downarrow$ | 34.05 | 28.61 | 28.43 | 22.27 | 21.18 | 55.55 | 49.61 | 57.20 | 37.42 | 38.15 |
> > > | | ASR (%) $\uparrow$ | 57.75 | 64.52 | 64.79 | 72.44 | 73.86 | 30.84 | 38.44 | 28.63 | 53.61 | 52.77 |
> > > | | FR (%) $\uparrow$ | 63.01 | 69.32 | 69.50 | 76.14 | 77.43 | 38.88 | 45.82 | 36.70 | 59.60 | 58.86 |
> > > | | ACR (%) $\downarrow$ | 7.80 | 6.62 | 6.76 | 5.36 | 5.34 | 12.01 | 11.37 | 11.88 | 8.76 | 9.09 |
> > > | **GAMA** | Acc. (%) $\downarrow$ | 31.66 | 26.10 | 26.25 | 20.18 | 19.10 | 52.47 | 45.53 | 54.05 | 35.08 | 35.78 |
> > > | | ASR (%) $\uparrow$ | 60.67 | 67.59 | 67.48 | 75.03 | 76.46 | 34.61 | 43.45 | 32.55 | 56.41 | 55.52 |
> > > | | FR (%) $\uparrow$ | 65.60 | 71.93 | 71.88 | 78.35 | 79.57 | 42.41 | 50.40 | 40.47 | 62.00 | 61.17 |
> > > | | ACR (%) $\downarrow$ | 7.12 | 5.90 | 6.19 | 4.88 | 4.88 | 11.15 | 10.32 | 11.18 | 7.88 | 7.97 |
> > > | **GAMA_Ours** | Acc. (%) $\downarrow$ | 30.30 | 25.31 | 21.75 | 19.41 | 18.41 | 52.09 | 43.89 | 53.91 | 32.60 | 33.61 |
> > > | | ASR (%) $\uparrow$ | 62.37 | 68.63 | 73.06 | 76.01 | 77.29 | 35.09 | 45.46 | 32.76 | 59.55 | 58.28 |
> > > | | FR (%) $\uparrow$ | 67.08 | 72.92 | 76.65 | 79.26 | 80.33 | 42.71 | 52.09 | 40.62 | 64.77 | 63.66 |
> > > | | ACR (%) $\downarrow$ | 6.86 | 5.93 | 5.16 | 4.77 | 4.65 | 11.09 | 9.80 | 11.25 | 7.48 | 7.68 |
> > > | **FACL** | Acc. (%) $\downarrow$ | 27.12 | 22.00 | 18.64 | 16.68 | 15.84 | 45.91 | 36.29 | 48.38 | 32.40 | 33.11 |
> > > | | ASR (%) $\uparrow$ | 66.41 | 72.75 | 77.08 | 79.49 | 80.59 | 42.91 | 55.01 | 39.73 | 59.88 | 59.00 |
> > > | | FR (%) $\uparrow$ | 70.66 | 76.42 | 80.21 | 82.20 | 83.13 | 49.88 | 60.58 | 46.97 | 64.60 | 64.31 |
> > > | | ACR (%) $\downarrow$ | 6.44 | 5.22 | 4.95 | 4.43 | 4.42 | 10.14 | 8.46 | 10.34 | 7.74 | 7.89 |
> > > | **FACL_Ours** | Acc. (%) $\downarrow$ | 25.30 | 19.68 | 16.91 | 15.20 | 14.61 | 43.08 | 33.43 | 46.18 | 30.56 | 31.32 |
> > > | | ASR (%) $\uparrow$ | 68.69 | 75.78 | 79.19 | 81.35 | 82.13 | 46.35 | 58.54 | 42.50 | 62.15 | 61.28 |
> > > | | FR (%) $\uparrow$ | 72.68 | 79.02 | 81.99 | 83.83 | 84.49 | 52.95 | 63.78 | 49.47 | 67.02 | 66.33 |
> > > | | ACR (%) $\downarrow$ | 6.08 | 5.15 | 4.47 | 4.15 | 4.16 | 9.29 | 7.77 | 9.97 | 7.25 | 7.66 |
> > > | **PDCL** | Acc. (%) $\downarrow$ | 31.34|26.04|22.50|21.84|19.02|52.68|45.87|56.05|32.97|34.40|
> > > | | ASR (%) $\uparrow$|60.87|67.76|72.16|73.00|76.50|31.79|43.04|29.95|59.06|57.78|
> > > | | FR (%) $\uparrow$|65.75|72.10|75.91|76.60|79.65|39.75|50.02|38.04|64.29|63.17|
> > > | | ACR (%) $\downarrow$|7.09|6.20|5.39|5.35|4.67|11.42|10.41|11.30|7.46|7.87|
> > > | **PDCL_Ours** | Acc. (%) $\downarrow$|30.80|25.20|21.84|19.58|18.63|53.77|45.34|55.29|32.64|33.71|
> > > | | ASR (%) $\uparrow$|61.76|68.80|73.00|75.85|77.04|32.87|43.68|30.98|59.54|58.26|
> > > | | FR (%) $\uparrow$ | 66.54|73.03|76.60|79.06|80.16|40.66|50.63|38.92|64.34|63.74|
> > > | | ACR (%) $\downarrow$|7.02|6.00|5.35|4.97|4.80|11.03|10.22|11.39|7.63|8.06|

---

### Author Response · Authors · 2025-12-01
**Final Remarks**

We thank the reviewers and ACs for the entire rebuttal process.
In the following, we summarize highlights where the reviews align and how our rebuttal addresses the questions.

### 1. Consensus on strengths
- *Generator-centric attack framework*
  Reviewers `acvB`, `7rta`, and `GNru` emphasize the generator block analysis, and `y2kS` finds the intermediate block study meaningful. Together, the reviews recognize **generator-centric semantic consistency inside the generator** as a novel and effective handle on black-box transferability.

- *Sharper and reliable evaluation via `ACR`*
  Reviewers `7rta` and `GNru` note that **ACR** separates genuine adversarial success from **inadvertent turnovers to correct labels**, filling a gap in standard accuracy and fooling metrics and providing a useful diagnostic for attacks and defenses.

- *Breadth and clarity*
  Reviewers `y2kS`, `acvB`, `7rta`, and `GNru` regard the experiments as broad, and `y2kS` and `foEg` praise the clarity of the writing, figures, and tables.

### 2. Main reviewer questions and our responses
- **Generality beyond the original untargeted setting**
  Reviewers `acvB` and `foEg` ask whether the framework extends beyond the original untargeted classification setup. We added targeted **M3D** and CLIP-guided **CGNC** experiments. For M3D, the target success improves from **24.64%** to **31.80%**, averaged over 10 target classes. For CGNC, the target success rises from **4.8%** to **18.2%** across seven victims. This shows that our generator-centric mechanism is effective for both a strong targeted and a CLIP-guided generative attack.

- **Attack behavior on CNN/ViT victims**
  Reviewers `7rta` and `GNru` ask how to interpret the mixed gains on transformer victims. Our attack uses a ResNet-based generator and CNN surrogates, while the victim pool includes diverse architectures such as CNN and ViT, so heterogeneous choices lead to different transfer patterns. Within this structure, our method **consistently improves each baseline**, with varying magnitude across architectures. We see this architectural mismatch as a limitation of the current generative framework in general and will point it as a natural direction for future cross-architecture transfer work for more aligned generator-surrogate-victim choices for high transfer.

- **Robustness under defenses**
  Reviewers `y2kS`, `7rta`, and `GNru` request a more systematic defense study, including diffusion-based purification. The original submission already evaluated NRP (Table S10). During rebuttal, we added experiments with input preprocessing defenses (randomization, random rotation, smoothing, total variation minimization, pixel deflection) and with DiffPure on two victim architectures. Across these settings, defended performance with our method does not show systematic degradation relative to the baselines mostly. Our method thus **does not trade robustness to defenses for adversarial transfer** and remains compatible with widely used preprocessing and purification defenses, while still targeting improved transfer in the undefended setting.

- **Comparison with other strong attacks and compute**
  Reviewers `acvB`, `foEg`, and `y2kS` ask whether the gains could be attributed to heavier tuning or compute and how the method compares to additional strong attacks. All baselines follow their original formulations, and we add our semantic consistency loss inside the generator on top of Mean Teacher. We compare our method on total training compute cost,and against a strong white-box transfer attack e.g., Admix. Ablations on EMA, hyperparameter sensitivity, and early block selection show stable improvements and indicate that our method is a **robust generator-centric addition rather than a narrow tuning of existing attacks.**

### 3. Scope of Additional Experiments
- **Tasks:** Extended to targeted generative attacks M3D and CLIP-guided CGNC, both showing clear gains.
- **Baselines:** Included Admix (strong white-box transfer) for a stronger comparison.
- **Defenses:** Further evaluated robustness under input preprocessing and purification defenses.
- **Analysis:** Ablations on design choices and compute logs confirm stability and efficiency.

These results support that **controlling generator semantics** reliably improves transfer across diverse settings.

### 4. Overall
- The work establishes a **generator-centric perspective** via **intermediate semantic consistency**, with substantial gains across domains, architectures, and tasks.
- Together with **ACR-based evaluation**, defense studies, and ablations, this provides a rigorous and comprehensive validation of our approach.

In sum, we move beyond surrogate manipulation to focus on generator-internal  semantics, supported by extensive experiments and the ACR metric across cross-model, domain, and task settings, and ensuring more robust and practical black-box generative attacks.
We have revised the manuscript accordingly. Thank you.

---

### Meta-Review · Area_Chair_Hib3 · 2026-01-09

**Summary:**

This paper proposes SCGA, a generative attack that improves transferability by making generator features semantically consistent and by stabilizing training with EMA. The paper also introduces a new metric, ACR.

Initially, the paper received mixed scores from 2 to 8. While several reviewers (acvB, 7rta, GNru) found the idea interesting and potentially useful, others (y2kS, foEg) raised concerns regarding limited novelty, weak cross-architecture transferability, and the unclear significance of the proposed ACR metric.

One of the main concerns is the limited novelty. The method mainly follows the same prior generator-based transfer-attack framework,  such as BIA [1], and common methods for loss function and EMA training. While it is true that the proposed approach employs commonly used components, such as feature consistency losses and EMA-based stabilization, on top of the previous framework, the authors clarify in the rebuttal that their method is the first work to explicitly target and validate a generator-side semantic regularizer for black-box transferability. The AC agrees that the generator-centric perspective represents a meaningful conceptual shift, and the observed performance gains indicate that the contribution is incremental but non-trivial.

Another issue is about cross-architecture transferability. The method shows clear performance gains on CNNs, but shows limited benefit on Transformer-based models. In the rebuttal, the authors are transparent about this limitation and provide plausible explanations related to architectural inductive biases. After considering the rebuttal, the current evidence is considered sufficient to establish the method’s effectiveness in widely used CNN-based black-box settings, although generalization to Transformer-based victims remains limited.

The value of the proposed ACR metric was also discussed. While its necessity was initially unclear, the rebuttal clarifies that ACR is defined to explicitly isolate incorrect-to-correct prediction flips by framing evaluation in terms of pre- and post-attack prediction transitions, thereby capturing a failure mode not distinguished by existing metrics.

In response to reviewer requests regarding evaluation against additional defense methods, the authors added extensive evaluations against input-processing defenses and included DiffPure [2] experiments. Although SCGA is not designed as a defense-adaptive attack, the additional results demonstrate that it does not degrade attack effectiveness under strong purification defenses.

Overall, the paper is technically sound, clearly written, carefully implemented, and empirically well structured. While some concerns remain regarding novelty and cross-architecture generalization, the paper introduces a useful generator-centric perspective on transfer attacks and provides sufficient empirical support for its claims. Therefore, the AC recommends accepting the paper.


[1] Beyond imagenet attack: Towards crafting adversarial examples for black-box domains. ICLR 2022

[2] Diffusion Models for Adversarial Purification, ICML 2022

**Reviewer Concerns:**

addressed: 7rta, acvB, y2kS, GNru
not fully resolved: foEg

**Reviewer Scores:**

y2kS: 2->4
7rta: 6->6
acvB: 6->6
foEg: 2->2 (or 2->0)
GNru: 8->8

---

### Decision · Program_Chairs · 2026-01-26

Accept (Poster)